# UCB-based Algorithms for
# Multinomial Logistic Regression Bandits

**Sanae Amani**
University of California, Los Angeles
samani@ucla.edu

**Christos Thrampoulidis**
University of British Columbia
cthrampo@ece.ubc.ca

## Abstract

Out of the rich family of generalized linear bandits, perhaps the most well studied ones are logistic bandits that are used in problems with binary rewards: for instance, when the learner aims to maximize the profit over a user that can select one of two possible outcomes (e.g., 'click' vs 'no-click'). Despite remarkable recent progress and improved algorithms for logistic bandits, existing works do not address practical situations where the number of outcomes that can be selected by the user is larger than two (e.g., 'click', 'show me later', 'never show again', 'no click'). In this paper, we study such an extension. We use multinomial logit (MNL) to model the probability of each one of $K + 1 \geq 2$ possible outcomes (+1 stands for the 'not click' outcome): we assume that for a learner's action $\mathbf{x}_t$, the user selects one of $K + 1 \geq 2$ outcomes, say outcome $i$, with a MNL probabilistic model with corresponding unknown parameter $\bar{\boldsymbol{\theta}}_{*i}$. Each outcome $i$ is also associated with a revenue parameter $\rho_i$ and the goal is to maximize the expected revenue. For this problem, we present MNL-UCB, an upper confidence bound (UCB)-based algorithm, that achieves regret $\tilde{\mathcal{O}}(dK\sqrt{T})$ with small dependency on problem-dependent constants that can otherwise be arbitrarily large and lead to loose regret bounds. We present numerical simulations that corroborate our theoretical results.

## 1 Introduction

Linear stochastic bandits provide simple, yet commonly encountered, models for a variety of sequential decision-making problems under uncertainty. Specifically, linear bandits generalize the classical multi-armed bandit (MAB) problem of $K$ arms that each yields reward sampled independently from an underlying distribution with unknown parameters, to a setting where the expected reward of each arm is a linear function that depends on the same unknown parameter vector Dani et al. (2008); Abbasi-Yadkori et al. (2011); Rusmevichientong and Tsitsiklis (2010). Linear bandits have been successfully applied over the years in online advertising, recommendation services, resource allocation, etc. Lattimore and Szepesvári (2020). More recently, researchers have explored the potentials of such algorithms in more complex systems, such as in robotics, wireless networks, the power grid, medical trials, e.g., Li et al. (2013); Avner and Mannor (2019); Berkenkamp et al. (2016); Sui et al. (2018). However, linear bandits fail to model a host of other applications. This has called for extensions of linear bandits to a broader range of reward structures beyond linear models. One of the leading lines of work addressing these extensions relies on the Generalized Linear Model (GLM) framework of statistics. In GLMs the expected reward associated with an arm $\mathbf{x}$ is given by $\mu(\bar{\boldsymbol{\theta}}^T \mathbf{x})$, where $\bar{\boldsymbol{\theta}} \in \mathbb{R}^d$ is the system unknown parameter and $\mu$ is a non-linear link function. Specifically, *logistic bandits*, that are appropriate for modeling binary reward structures, are a special case of generalized linear bandits (GLBs) with $\mu(x) = (1 + \exp(-x))^{-1}$. UCB-based algorithms for GLBs were first introduced in Filippi et al. (2010); Li et al. (2017); Faury et al. (2020). The same problem, but with a Thompson Sampling- (TS) strategy was also studied in Abeille et al. (2017);

Russo and Van Roy (2013, 2014); Dong and Van Roy (2018). Beyond GLMs, an even more general framework for modeling reward is the semi-parametric index model (see for example Yang et al. (2017); Gamarnik and Gaudio (2020) for a list of applications in statistics). A semi-parametric index model relates the reward $y \in \mathbb{R}$ and the action/arm $\mathbf{x} \in \mathbb{R}^d$ as $y = \mu(\bar{\boldsymbol{\theta}}_1^T \mathbf{x}, \bar{\boldsymbol{\theta}}_2^T \mathbf{x}, \ldots, \bar{\boldsymbol{\theta}}_K^T \mathbf{x}) + \epsilon$, where $\mu : \mathbb{R}^K \to \mathbb{R}$ and $\bar{\boldsymbol{\theta}}_1, \ldots, \bar{\boldsymbol{\theta}}_K \in \mathbb{R}^d$ are $K$ system's unknown parameters. GLBs are special cases of this for $K = 1$, also known as single-index models (SIM) in statistics. In this paper, we formulate an extension of the problem of binary logistic bandits (i.e., a special case of SIM) to multinomial logit (MNL) bandits, a special case of multi-index models (MIM) to account for settings with more than two possible outcomes on the user choices ($K \geq 1$). For this model, we present an algorithm and corresponding regret bound. Our algorithmic and analytic contribution is in large inspired by recent exciting progress on binary logistic bandits by Faury et al. (2020).

To motivate MNL bandits, consider ad placement. When an ad is shown to a user, the user may have several options to react to the ad. For example, she can choose to 1) click on the ad; 2) click on "show me later"; 3) click on "never show me this ad"; 4) not click at all, etc. The user selects each of these options based on an unknown probability distribution that inherently involves linear combinations of the selected feature vector denoting the ad and unknown parameters denoting the user's preferences about the ad. In this setting, each option is associated with a specific notion of reward. The agent's goal is to determine ads with maximum expected rewards to increase the chance of a successful advertisement.

**Outline.** In Section 1.1, we formally define the problem. In Sections 2.1, 2.2 and 2.4, we elaborate on the challenges that the generalization of the Logistic-UCB-1 by Faury et al. (2020) to the settings with MIM rewards brings to our theoretical analysis. We then summarize our proposed MNL-UCB in Algorithm 1 and provide a regret bound for it in Section 2.5. In Section 3, we present a detailed discussion on the challenges and computation of necessary problem-dependent constants. Finally, we complement our theoretical results with numerical simulations in Section 4.

**Notation.** We use lower-case letters for scalars, lower/upper-case bold letters for vectors/matrices. $\|\mathbf{x}\|_2$ denotes the Euclidean norm and $\mathbf{x}^T \mathbf{y}$ inner product. $\delta_{ij}$ denotes the Kronecker delta, i.e., $\delta_{ij} = 1$ if $i \neq j$ and $\delta_{ij} = 0$ if $i = j$. $\mathbf{A} \otimes \mathbf{B}$ denotes the Kronecker product. For square matrices $\mathbf{A}$ and $\mathbf{B}$, we use $\mathbf{A} \preceq \mathbf{B}$ to denote $\mathbf{B} - \mathbf{A}$ is positive semi-definite. We denote the minimum and maximum eigenvalues of $\mathbf{A}$ by $\lambda_{\min}(\mathbf{A})$ and $\lambda_{\max}(\mathbf{A})$. For $\mathbf{A} \succeq 0$, the weighted 2-norm of $\boldsymbol{\nu}$ with respect to $\mathbf{A}$ is defined by $\|\boldsymbol{\nu}\|_{\mathbf{A}} = \sqrt{\boldsymbol{\nu}^T \mathbf{A} \boldsymbol{\nu}}$. For positive integers $n$ and $m \leq n$, $[n]$ and $[m : n]$ denote the sets $\{1, 2, \ldots, n\}$ and $\{m, \ldots, n\}$, respectively. For any $\boldsymbol{\nu} \in \mathbb{R}^{Kd}$, $\bar{\boldsymbol{\nu}}_i = \boldsymbol{\nu}_{[(i-1)d+1:d]} \in \mathbb{R}^d$ denotes the vector containing the $i$-th set of $d$ entries of $\boldsymbol{\nu}$. We use $\mathbf{1}$ and $\mathbf{e}_i$ to denote the vector of all 1's and the $i$-th standard basis vector, respectively. Finally, we use $\tilde{\mathcal{O}}$ for big-Oh notation that ignores logarithmic factors.

## 1.1 Problem formulation

**Reward Model.** The agent is given a decision set[1] $\mathcal{D} \subset \mathbb{R}^d$. At each round $t$, the agent chooses an action $\mathbf{x}_t \in \mathcal{D}$ and observes the user purchase decision $y_t \in [K] \cup \{0\}$. Here, $\{0\}$ denotes the "outside decision", which means the user did not select any of the presented options. The agent's decision at round $t$ is based on the information gathered until time $t$, which can be formally encoded in the filtration $\mathcal{F}_t := (\mathcal{F}_0, \sigma(\{\mathbf{x}_s, y_s\}_{s=1}^{t-1}))$, where $\mathcal{F}_0$ represents any prior knowledge. Let each option $i \in [K]$ be associated with an unknown vector $\bar{\boldsymbol{\theta}}_{*i} \in \mathbb{R}^d$ and let $\boldsymbol{\theta}_* = [\bar{\boldsymbol{\theta}}_{*1}^T, \bar{\boldsymbol{\theta}}_{*2}^T, \ldots, \bar{\boldsymbol{\theta}}_{*K}^T]^T \in \mathbb{R}^{Kd}$. The user's choice of what to click on is given by a multinomial logit (MNL) choice model. Under this model, the probability distribution of the user purchase decision is given by

$$\mathbb{P}(y_t = i | \mathbf{x}_t, \mathcal{F}_t) := \begin{cases} \frac{1}{1 + \sum_{j=1}^K \exp(\bar{\boldsymbol{\theta}}_{*j}^T \mathbf{x}_t)} & \text{, if } i = 0, \\ \frac{\exp(\bar{\boldsymbol{\theta}}_{*i}^T \mathbf{x}_t)}{1 + \sum_{j=1}^K \exp(\bar{\boldsymbol{\theta}}_{*j}^T \mathbf{x}_t)} & \text{, if } i \in [K]. \end{cases} \tag{1}$$

When the user clicks on the $i$-th option, a corresponding reward $\rho_i$ is revealed to the agent and we set $\rho_0 = 0$. Then, the expected reward observed by the agent when she plays action $\mathbf{x}_t$ is

---

[1]Our results extend easily to time-varying decision sets.

$\mathbb{E}[R_t|\mathbf{x}_t, \mathcal{F}_t] = \frac{\sum_{j=1}^{K} \rho_j \exp(\bar{\boldsymbol{\theta}}_{*j}^T \mathbf{x}_t)}{1 + \sum_{j=1}^{K} \exp(\bar{\boldsymbol{\theta}}_{*j}^T \mathbf{x}_t)} = \boldsymbol{\rho}^T \mathbf{z}(\mathbf{x}_t, \boldsymbol{\theta}_*)$, where $\boldsymbol{\rho} = [\rho_1, \rho_2, \dots, \rho_K]^T$, $\mathbf{z}(\mathbf{x}_t, \boldsymbol{\theta}_*) = [z_1(\mathbf{x}_t, \boldsymbol{\theta}_*), z_2(\mathbf{x}_t, \boldsymbol{\theta}_*), \dots, z_K(\mathbf{x}_t, \boldsymbol{\theta}_*)]^T$, and

$$z_i(\mathbf{x}_t, \boldsymbol{\theta}_*) = \mathbb{P}(y_t = i|\mathbf{x}_t, \mathcal{F}_t), \ \forall i \in [K]. \tag{2}$$

Note that $\mathbb{E}[R_t|\mathbf{x}_t, \mathcal{F}_t] = \mu(\bar{\boldsymbol{\theta}}_{*1}^T \mathbf{x}, \bar{\boldsymbol{\theta}}_{*2}^T \mathbf{x}, \dots, \bar{\boldsymbol{\theta}}_{*K}^T \mathbf{x})$ is *not* directly a generalized linear model, i.e., a function $\mu(\bar{\boldsymbol{\theta}}_*^T \mathbf{x}_t)$, but rather it is a *multi-index model*, where $\mu : \mathbb{R}^K \to \mathbb{R}$.

**Goal.** Let $T$ be the total number of rounds and $\mathbf{x}_*$ be the optimal action that maximizes the reward in expectation, i.e., $\mathbf{x}_* \in \arg\max_{\mathbf{x} \in \mathcal{D}} \boldsymbol{\rho}^T \mathbf{z}(\mathbf{x}, \boldsymbol{\theta}_*)$. The agent's goal is to minimize the *cumulative pseudo-regret* defined by $R_T = \sum_{t=1}^{T} \boldsymbol{\rho}^T \mathbf{z}(\mathbf{x}_*, \boldsymbol{\theta}_*) - \boldsymbol{\rho}^T \mathbf{z}(\mathbf{x}_t, \boldsymbol{\theta}_*)$.

## 1.2 Contributions

We study MNL logistic regression bandits, a generalization of binary logistic bandits, that address applications where the number of outcomes that can be selected by the user is larger than two. The probability of selecting any possible $K + 1 > 2$ options (+1 stands for the 'not click' outcome aka "outside decision") is modeled using a multinomial logit (MNL) model. For this problem:

• We identify a critical parameter $\kappa$, which we interpret as the degree of (non)-smoothness (less smooth for larger values of $\kappa$) of the MNL model over the agent's decision set. We prove that $\kappa$ scales exponentially with the size of the agent's decision set creating a challenge in the design of low-regret algorithms, similar to the special binary case previously studied in the literature.
• We develop a UCB-type algorithm for MNL logistic regression bandits. At every step, the algorithm decides on the inclusion of a K-tuple of parameter vectors $\bar{\boldsymbol{\theta}}_{*1}, \dots, \bar{\boldsymbol{\theta}}_{*K}$ in the confidence region in a way that captures the local smoothness of the MNL model around this K-tuple and past actions. We show that this is critical for the algorithm's favorable regret performance in terms of $\kappa$.
• Specifically, we prove that the regret of our MNL-UCB scales as $\tilde{\mathcal{O}}(dK\sqrt{\kappa}\sqrt{T})$. Instead, we show that a confidence ellipsoid that fails to capture local dependencies described above results in regret that scales linearly with $\kappa$ rather than with $\sqrt{\kappa}$. Moreover, our regret bound scales optimally in terms of the number of options $K$.
• We propose an improved algorithm that achieves a regret bound with problem-dependent constant $\kappa$ being pushed into a second order term that vanishes quickly.
• We complement our theoretical results with numerical simulations and corresponding discussions on the performance of our algorithm.

## 1.3 Related works

**Generalized Linear Bandits.** GLBs were studied in Filippi et al. (2010); Li et al. (2017); Abeille et al. (2017); Russo and Van Roy (2013, 2014); Dong and Van Roy (2018) where the stochastic reward is modeled through an appropriate strictly increasing link function $\mu$. All these works provide regret bounds $\tilde{\mathcal{O}}(\kappa\sqrt{T})$, where the multiplicative factor $\kappa$ is a problem-dependent constant and characterizes the degree of non-linearity of the link function.

**Logistic Bandits.** In Faury et al. (2020), the authors focused on the *logistic bandit* problem as a special case of GLBs. By introducing a novel Bernstein-like self-normalized martingale tail-inequality, they reduced the dependency of the existing GLB algorithms' regret bounds on the constant $\kappa$ by a factor of $\sqrt{\kappa}$ and obtained a $\mathcal{O}(d\sqrt{\kappa T})$ regret for the logistic bandit problem. They further discussed the crucial role of $\kappa$, which can be arbitrarily large as it scales exponentially with the size of the decision set, on the performance of existing algorithms. Motivated by such considerations, with careful algorithmic designs, they achieved to drop entirely the dependence on $\kappa$ leading to a regret of $\tilde{\mathcal{O}}(d\sqrt{T})$.

**Multinomial Logit Bandits.** In a different line of work, Agrawal et al. (2017, 2019); Wang et al. (2018); Oh and Iyengar (2019); Chen et al. (2018); Dong et al. (2020); Agrawal et al. (2020) used the multinomial logit choice model to address the *dynamic assortment selection* problem, which is a combinatorial variant of the bandit problem. In this problem, the agent chooses a so-called assortment which is a subset of a set $\mathcal{S} = [N]$ of $N$ items. At round $t$, feature vectors $\mathbf{x}_{it}$, for every item $i \in \mathcal{S}$, are revealed to the agent, and given this contextual information, the agent selects an assortment $S_t \subset \mathcal{S}$

and observes the user choice $y_t = i$, $i \in S_t \cup \{0\}$ where $\{0\}$ corresponds to the user not selecting any item in $S_t$. The user choice is given by a MNL model with an unknown parameter $\bar{\boldsymbol{\theta}}_* \in \mathbb{R}^d$ such that the probability that the user selects item $i \in S_t$ is $\exp(\bar{\boldsymbol{\theta}}_*^T \mathbf{x}_{it})/(1 + \sum_{i \in S_t} \exp(\bar{\boldsymbol{\theta}}_*^T \mathbf{x}_{it}))$. Furthermore, a revenue parameter denoted by $r_{it}$ for each item $i$ is also revealed at round $t$. The goal of the agent is to offer assortments with size at most $K$ to maximize the expected cumulative revenue or to minimize the cumulative regret $\sum_{t=1}^T R_t(S_t^*, \bar{\boldsymbol{\theta}}_*) - R_t(S_t, \bar{\boldsymbol{\theta}}_*)$, where $R_t(S_t, \bar{\boldsymbol{\theta}}_*) := \sum_{i \subset S_t} r_{it} \exp(\bar{\boldsymbol{\theta}}_*^T \mathbf{x}_{it})/(1 + \sum_{i \in S_t} \exp(\bar{\boldsymbol{\theta}}_*^T \mathbf{x}_{it}))$. Finally, the closely related paper Cheung and Simchi-Levi (2017) studies a problem where at each round, the agent observes a user-specific context based on which, it recommends a set of items to the user. The probability distribution of each one of the items in that set being selected by the user is given by an MNL model. This problem can be categorized as an online assortment optimization problem. Despite similarities in the use of an MNL model, there are certain differences between Cheung and Simchi-Levi (2017) and our paper in terms of problem formulation. In our setting, the user may have multiple reactions (one of $K + 1$ options), to a single selected item. In contrast, in Cheung and Simchi-Levi (2017), the agent must select a set of items to each of which the user reacts by either clicking or not clicking. Also, here the probability distribution of different user reactions remains the same at all rounds, while in Cheung and Simchi-Levi (2017) the response to an item from the recommended set depends on the other items in that set. We defer to future work studying implications of our techniques to the setting of Cheung and Simchi-Levi (2017).

## 2 Multnomial Logit UCB Algorithms

In this section, we introduce two key quantities: (i) $\boldsymbol{\theta}_t$, an estimate of $\boldsymbol{\theta}_*$; (ii) $\epsilon_t(\mathbf{x})$, an exploration bonus for each $\mathbf{x} \in \mathcal{D}$ at each round $t \in [T]$. Based on these, we design a UCB-type algorithm, called MNL-UCB. At each round $t$, the algorithm computes an estimate $\boldsymbol{\theta}_t$ of $\boldsymbol{\theta}_*$, that we present in Section 2.4. For each $\mathbf{x} \in \mathcal{D}$ and $t \in [T]$, let $\epsilon_t(\mathbf{x})$ be such that the following holds with high probability:

$$\Delta(\mathbf{x}, \boldsymbol{\theta}_t) := |\boldsymbol{\rho}^T \mathbf{z}(\mathbf{x}, \boldsymbol{\theta}_*) - \boldsymbol{\rho}^T \mathbf{z}(\mathbf{x}, \boldsymbol{\theta}_t)| \leq \epsilon_t(\mathbf{x}). \tag{3}$$

At round $t$, having knowledge of $\epsilon_t(\mathbf{x})$, the agent computes the following upper bound on the expected reward $\boldsymbol{\rho}^T \mathbf{z}(\mathbf{x}, \boldsymbol{\theta}_*)$ for all $\mathbf{x} \in \mathcal{D}$:

$$\boldsymbol{\rho}^T \mathbf{z}(\mathbf{x}, \boldsymbol{\theta}_*) \leq \boldsymbol{\rho}^T \mathbf{z}(\mathbf{x}, \boldsymbol{\theta}_t) + \epsilon_t(\mathbf{x}). \tag{4}$$

Then, the learner follows a UCB decision rule to select an action $\mathbf{x}_t$ according to the following rule:

$$\mathbf{x}_t := \arg\max_{\mathbf{x} \in \mathcal{D}} \boldsymbol{\rho}^T \mathbf{z}(\mathbf{x}, \boldsymbol{\theta}_t) + \epsilon_t(\mathbf{x}). \tag{5}$$

To see how the UCB decision rule in (5) helps us control the cumulative regret, we show how it controls the instantaneous regret by the following standard argument Abbasi-Yadkori et al. (2011); Faury et al. (2020):

$$r_t = \boldsymbol{\rho}^T \mathbf{z}(\mathbf{x}_*, \boldsymbol{\theta}_*) - \boldsymbol{\rho}^T \mathbf{z}(\mathbf{x}_t, \boldsymbol{\theta}_*) \overset{(4)}{\leq} \boldsymbol{\rho}^T \mathbf{z}(\mathbf{x}_*, \boldsymbol{\theta}_t) + \epsilon_t(\mathbf{x}_*) - \boldsymbol{\rho}^T \mathbf{z}(\mathbf{x}_t, \boldsymbol{\theta}_*)$$

$$\overset{(5)}{\leq} \boldsymbol{\rho}^T \mathbf{z}(\mathbf{x}_t, \boldsymbol{\theta}_t) - \boldsymbol{\rho}^T \mathbf{z}(\mathbf{x}_t, \boldsymbol{\theta}_*) + \epsilon_t(\mathbf{x}_t) \overset{(3)}{\leq} 2\epsilon_t(\mathbf{x}_t). \tag{6}$$

In view of this, our goal is to design an algorithm that appropriately chooses the estimator $\boldsymbol{\theta}_t$ and the exploration bonus $\epsilon_t(\mathbf{x})$ such that its regret is sub-linear.

### 2.1 Maximum likelihood estimate

The problem of estimating $\boldsymbol{\theta}_*$ at round $t$ given $\mathcal{F}_t$ is identical to a multi-class linear classification problem, where $\bar{\boldsymbol{\theta}}_{*i}$ is the "classifier" for class $i$. A natural way to compute the estimator of the unknown parameter $\boldsymbol{\theta}_*$ of the MNL model given $\mathcal{F}_t$ is to use the maximum likelihood principle. At round $t$, the regularized log-likelihood (aka negative cross-entropy loss) with regularizer $\lambda > 0$ writes

$$\mathcal{L}_t^\lambda(\boldsymbol{\theta}) := \sum_{s=1}^{t-1} \sum_{i=0}^K \mathbb{1}\{y_s = i\} \log \left( z_i(\mathbf{x}_s, \boldsymbol{\theta}) \right) - \frac{\lambda}{2} \|\boldsymbol{\theta}\|_2^2. \tag{7}$$

Then, the maximum likelihood estimate of $\boldsymbol{\theta}_*$ is defined as $\hat{\boldsymbol{\theta}}_t := \arg\max_{\boldsymbol{\theta}} \mathcal{L}_t^\lambda(\boldsymbol{\theta})$. Taking the gradient of (7) with respect to $\boldsymbol{\theta}$ we obtain

$$\nabla_{\boldsymbol{\theta}} \mathcal{L}_t^\lambda(\boldsymbol{\theta}) := \sum_{s=1}^{t-1} [\mathbf{m}_s - \mathbf{z}(\mathbf{x}_s, \boldsymbol{\theta})] \otimes \mathbf{x}_s - \lambda \boldsymbol{\theta}, \tag{8}$$

where $\mathbf{m}_s$ is the 'one-hot encoding' vector of the user's selection at round $s$, i.e., $\mathbf{m}_s := \left[\mathbb{1}\{y_s = 1\}, \ldots, \mathbb{1}\{y_s = K\}\right]^T$. It will also be convenient to define the Hessian of $-\mathcal{L}_t^\lambda(\boldsymbol{\theta})$:

$$\mathbf{H}_t(\boldsymbol{\theta}) := \lambda I_{Kd} + \sum_{s=1}^{t-1} \mathbf{A}(\mathbf{x}_s, \boldsymbol{\theta}) \otimes \mathbf{x}_s \mathbf{x}_s^T, \tag{9}$$

where $\mathbf{A}(\mathbf{x}, \boldsymbol{\theta})_{ij} := z_i(\mathbf{x}, \boldsymbol{\theta}) \left(\delta_{ij} - z_j(\mathbf{x}, \boldsymbol{\theta})\right)$ for all $i, j \in [K]$. Equivalently, in matrix form

$$\mathbf{A}(\mathbf{x}, \boldsymbol{\theta}) := \mathrm{diag}(\mathbf{z}(\mathbf{x}, \boldsymbol{\theta})) - \mathbf{z}(\mathbf{x}, \boldsymbol{\theta})\mathbf{z}^\mathrm{T}(\mathbf{x}, \boldsymbol{\theta}). \tag{10}$$

Note here that $\mathbf{A}(\mathbf{x}, \boldsymbol{\theta})$ is a matrix function that depends on $\boldsymbol{\theta}$ and $\mathbf{x}$ via the inner products $\bar{\boldsymbol{\theta}}_1^T \mathbf{x}, \bar{\boldsymbol{\theta}}_2^T \mathbf{x}, \ldots, \bar{\boldsymbol{\theta}}_K^T \mathbf{x}$. Also, the matrix $\mathbf{A}(\mathbf{x}, \boldsymbol{\theta})$ has nice algebraic properties (discussed more in Section 3) that turn our to be critical in the execution and analysis of our algorithm.

Now, we introduce assumptions on the problem structure, under which our theoretical results hold.

**Assumption 1** (Boundedness). *Without loss of generality, $\|\mathbf{x}\|_2 \le 1$ for all $\mathbf{x} \in \mathcal{D}$. Also, $\boldsymbol{\theta}_* \in \Theta := \{\boldsymbol{\theta} \in \mathbb{R}^{Kd} : \|\boldsymbol{\theta}\|_2 \le S\}$ and $\|\boldsymbol{\rho}\|_2 \le R$. Both upper bounds $S$ and $R$ are known to the agent.*

The assumption that $S$ is known is standard in the literature of GLBs. Knowledge of $R$ is also reasonable to assume because $\rho_i$'s represent the revenue parameters that are typically known or set by the system operator.

**Assumption 2** (Problem-dependent constants). *There exist strictly positive constants $0 < L < \infty$ and $0 < \kappa < \infty$ such that $\sup_{\mathbf{x} \in \mathcal{D}, \boldsymbol{\theta} \in \Theta} \lambda_{\max}\left(\mathbf{A}(\mathbf{x}, \boldsymbol{\theta})\right) := L$ and $\inf_{\mathbf{x} \in \mathcal{D}, \boldsymbol{\theta} \in \Theta} \lambda_{\min}\left(\mathbf{A}(\mathbf{x}, \boldsymbol{\theta})\right) := \frac{1}{\kappa}$.*

We comment further on the knowledge of $\kappa$ and $L$ in Section 3. Here, we note that $\kappa$ is reminiscent of the corresponding quantity in binary logistic bandits which is defined accordingly as $\kappa := \sup_{\mathbf{x} \in \mathcal{D}, \|\bar{\boldsymbol{\theta}}\|_2 \le S} 1/\dot{\mu}(\bar{\boldsymbol{\theta}}^T \mathbf{x})$, where $\dot{\mu}$ is the first derivative of the logistic function $\mu(x) = 1/(1 + \exp(-x))$. As Filippi et al. (2010); Li et al. (2017); Faury et al. (2020) have shown, this quantity plays a key role in characterizing the behavior of binary ($K = 1$) logit bandit algorithms. In this paper, we will show that the proper analogue of this quantity to multinomial ($K > 1$) logit bandit algorithms is the parameter $\kappa$ defined in Assumption 2.

## 2.2 Confidence set around $\boldsymbol{\theta}_*$

We introduce a confidence set $\mathcal{C}_t(\delta)$ that will include $\boldsymbol{\theta}_*$ with high probability thus allowing us to upper bound $\Delta(\mathbf{x}, \boldsymbol{\theta}_t)$ in the following subsection. We start by defining the key quantity

$$\mathbf{g}_t(\boldsymbol{\theta}) := \lambda\boldsymbol{\theta} + \sum_{s=1}^{t-1} \mathbf{z}(\mathbf{x}_s, \boldsymbol{\theta}) \otimes \mathbf{x}_s. \tag{11}$$

To see why this is useful, note that by the first-order optimality condition $\nabla_{\boldsymbol{\theta}} \mathcal{L}_t^\lambda(\hat{\boldsymbol{\theta}}_t) = \mathbf{0}$ we have $\boldsymbol{\theta}_*$ satisfying $\mathbf{g}_t(\boldsymbol{\theta}_*) - \mathbf{g}_t(\hat{\boldsymbol{\theta}}_t) = \lambda\boldsymbol{\theta}_* + \mathbf{s}_t$, with $\mathbf{s}_t := \sum_{s=1}^{t-1} \left(\mathbf{z}(\mathbf{x}_s, \boldsymbol{\theta}_*) - \mathbf{m}_s\right) \otimes \mathbf{x}_s$. This in turn motivates us to define a confidence set $\mathcal{C}_t$ at the beginning of each round $t \in [T]$ such that

$$\mathcal{C}_t(\delta) := \{\boldsymbol{\theta} \in \Theta : \left\|\mathbf{g}_t(\boldsymbol{\theta}) - \mathbf{g}_t(\hat{\boldsymbol{\theta}}_t)\right\|_{\mathbf{H}_t^{-1}(\boldsymbol{\theta})} \le \beta_t(\delta)\}, \tag{12}$$

where $\beta_t(\delta)$ is chosen as in Theorem 1 below to guarantee $\boldsymbol{\theta}_* \in \mathcal{C}_t(\delta)$ with high probability $1 - \delta$.

**Theorem 1** (Confidence set). *Let the Assumption 1 hold and for $\delta \in (0, 1)$, define $\beta_t(\delta) := \frac{K^{3/2}d}{\sqrt{\lambda}} \log\left(1 + \frac{t}{d\lambda}\right) + \frac{\sqrt{\lambda/K}}{2} + \frac{2K^{3/2}d}{\sqrt{\lambda}} \log(\frac{2}{\delta}) + \sqrt{\lambda}S$. Then with probability at least $1 - \delta$, for all $t \in [T]$ it holds that $\boldsymbol{\theta}_* \in \mathcal{C}_t(\delta)$.*

Once we have properly identified the key quantities $\mathbf{g}_t(\boldsymbol{\theta}_*)$ and $\mathbf{H}_t(\boldsymbol{\theta})$, the proof of the theorem above rather naturally extends Lemma 1 in Faury et al. (2020). Compared to the special case $K = 1$ studied in Faury et al. (2020), extra care is needed here to properly track the scaling of $\beta_t(\delta)$ with respect to the new parameter $K$ that is of interest for us. The details are deferred to Appendix A. To a great extent, the similarities between the binary and multinomial cases end here. It will turn out that bounding $\Delta(\mathbf{x}, \boldsymbol{\theta}_t)$, for an appropriate choice of $\boldsymbol{\theta}_t$, is significantly more intricate here than in the binary case. Our main technical contribution towards this direction is given in the lemmas presented in Section 2.3 that are used to prove the following key result.

**Lemma 1.** *Let Assumptions 1 and 2 hold. For all* $\mathbf{x} \in \mathcal{D}$, $t \in [T]$ *and* $\boldsymbol{\theta} \in \mathcal{C}_t(\delta)$*, with probability at least* $1 - \delta$*:*

$$\Delta(\mathbf{x}, \boldsymbol{\theta}) \leq 2RL\beta_t(\delta)\sqrt{\kappa\left(1 + 2S\right)}\|\mathbf{x}\|_{\mathbf{V}_t^{-1}}, \tag{13}$$

*where* $\mathbf{V}_t := \kappa\lambda I + \sum_{s=1}^{t-1} \mathbf{x}_s\mathbf{x}_s^T$.

The complete proof is in Appendix B. In the following section, we give a proof sketch.

## 2.3  Proof sketch of Lemma 1

To prove Lemma 1, we will use the high probability confidence set $\mathcal{C}_t(\delta)$ in (12) paired with the problem setting's properties encapsulated in Assumptions 1 and 2. To see the key challenges in establishing (13), consider the following. By definition of $\Delta(\mathbf{x}, \boldsymbol{\theta})$ in (3), Cauchy-Schwarz inequality, and Assumption 1, for any $\mathbf{x} \in \mathcal{D}$, $t \in [T]$, and $\boldsymbol{\theta} \in \mathcal{C}_t(\delta)$:

$$\Delta(\mathbf{x}, \boldsymbol{\theta}) \leq R\|\mathbf{z}(\mathbf{x}, \boldsymbol{\theta}_*) - \mathbf{z}(\mathbf{x}, \boldsymbol{\theta})\|_2. \tag{14}$$

Thus, our goal becomes relating $\|\mathbf{z}(\mathbf{x}, \boldsymbol{\theta}_*) - \mathbf{z}(\mathbf{x}, \boldsymbol{\theta})\|_2$ to $\left\|\mathbf{g}_t(\boldsymbol{\theta}_*) - \mathbf{g}_t(\hat{\boldsymbol{\theta}}_t)\right\|_{\mathbf{H}_t^{-1}(\boldsymbol{\theta}_*)}$ and/or $\left\|\mathbf{g}_t(\hat{\boldsymbol{\theta}}_t) - \mathbf{g}_t(\boldsymbol{\theta})\right\|_{\mathbf{H}_t^{-1}(\boldsymbol{\theta})}$. The reason is that the two latter quantities are both known to be bounded by $\beta_t(\delta)$ with high probability since $\boldsymbol{\theta}_* \in \mathcal{C}_t(\delta)$ with probability at least $1 - \delta$ (cf. Theorem 1). We accomplish our goal in three steps. First, in Lemma 2, we connect $\mathbf{z}(\mathbf{x}, \boldsymbol{\theta}_1) - \mathbf{z}(\mathbf{x}, \boldsymbol{\theta}_2)$ to $\boldsymbol{\theta}_1 - \boldsymbol{\theta}_2$ for any $\mathbf{x} \in \mathbb{R}^d$ and $\boldsymbol{\theta}_1, \boldsymbol{\theta}_2 \in \mathbb{R}^{Kd}$.

**Lemma 2.** *For any* $\mathbf{x} \in \mathbb{R}^d$, $\boldsymbol{\theta}_1, \boldsymbol{\theta}_2 \in \mathbb{R}^{Kd}$*, recall the definition of* $\mathbf{A}(\mathbf{x}, \boldsymbol{\theta})$ *in (10) and define*

$$\mathbf{B}(\mathbf{x}, \boldsymbol{\theta}_1, \boldsymbol{\theta}_2) := \int_0^1 \mathbf{A}(\mathbf{x}, v\boldsymbol{\theta}_1 + (1 - v)\boldsymbol{\theta}_2)dv. \tag{15}$$

*Then, we have* $\mathbf{z}(\mathbf{x}, \boldsymbol{\theta}_1) - \mathbf{z}(\mathbf{x}, \boldsymbol{\theta}_2) = \left[\mathbf{B}(\mathbf{x}, \boldsymbol{\theta}_1, \boldsymbol{\theta}_2) \otimes \mathbf{x}^T\right] (\boldsymbol{\theta}_1 - \boldsymbol{\theta}_2)$.

The proof of the lemma above in Appendix B.1 relies on a proper application of the mean-value Theorem. Next, in Lemma 3 below, we relate $\boldsymbol{\theta}_1 - \boldsymbol{\theta}_2$ to $\mathbf{g}_t(\boldsymbol{\theta}_1) - \mathbf{g}_t(\boldsymbol{\theta}_2)$.

**Lemma 3.** *Let*

$$\mathbf{G}_t(\boldsymbol{\theta}_1, \boldsymbol{\theta}_2) := \lambda I_{Kd} + \sum_{s=1}^{t-1} \mathbf{B}(\mathbf{x}_s, \boldsymbol{\theta}_1, \boldsymbol{\theta}_2) \otimes \mathbf{x}_s\mathbf{x}_s^T. \tag{16}$$

*Then, for any* $\boldsymbol{\theta}_1, \boldsymbol{\theta}_2 \in \mathbb{R}^{Kd}$*, we have*

$$\mathbf{g}_t(\boldsymbol{\theta}_1) - \mathbf{g}_t(\boldsymbol{\theta}_2) = \mathbf{G}_t(\boldsymbol{\theta}_1, \boldsymbol{\theta}_2)(\boldsymbol{\theta}_1 - \boldsymbol{\theta}_2). \tag{17}$$

The proof in Appendix B.1 uses the definition of $\mathbf{g}_t(\boldsymbol{\theta})$, Lemma 2 and a proper application of the mixed-product property of the Kronecker product. Note that $\mathbf{G}_t(\boldsymbol{\theta}_1, \boldsymbol{\theta}_2) \succeq 0$; thus, it is invertible.

Now, combining Lemmas 2 and 3, our new goal becomes bounding $\left\|\left[\mathbf{B}(\mathbf{x}, \boldsymbol{\theta}_*, \boldsymbol{\theta}) \otimes \mathbf{x}^T\right] \mathbf{G}_t^{-1}(\boldsymbol{\theta}_*, \boldsymbol{\theta}) \left(\mathbf{g}_t(\boldsymbol{\theta}_*) - \mathbf{g}_t(\boldsymbol{\theta})\right)\right\|_2$. Our key technical contribution is establishing good bounds for the spectral norm $\left\|\left[\mathbf{B}(\mathbf{x}, \boldsymbol{\theta}_*, \boldsymbol{\theta}) \otimes \mathbf{x}^T\right] \mathbf{G}_t^{-1/2}(\boldsymbol{\theta}_*, \boldsymbol{\theta})\right\|_2$ and the weighted Euclidean norm $\left\|\mathbf{g}_t(\boldsymbol{\theta}_*) - \mathbf{g}_t(\boldsymbol{\theta})\right\|_{\mathbf{G}_t^{-1}(\boldsymbol{\theta}_*, \boldsymbol{\theta})}$. We start by briefly explaining how we bound the spectral norm above; see Appendix B.2 for details. By using the cyclic property of

the maximum eigenvalue and the mixed-product property of the Kronecker product, it suffices to bound $\lambda_{\max}\left(\mathbf{G}_t^{-1/2}\left((\mathbf{B}^T\mathbf{B})\otimes(\mathbf{x}\mathbf{x}^T)\right)\mathbf{G}_t^{-1/2}\right)$, where we denote $\mathbf{B}=\mathbf{B}(\mathbf{x},\boldsymbol{\theta}_*,\boldsymbol{\theta})$ and $\mathbf{G}_t=\mathbf{G}_t(\boldsymbol{\theta}_*,\boldsymbol{\theta})$ for simplicity. There are two essential ideas to do so. First, thanks to our Assumption 2, which upper bounds the eigenvalues of $\mathbf{A}(\mathbf{x},\boldsymbol{\theta})$ by $L$, and by recalling the definition of $\mathbf{B}(\mathbf{x},\boldsymbol{\theta}_1,\boldsymbol{\theta}_2)$, we manage to show that $(\mathbf{B}\mathbf{B}^T)\otimes(\mathbf{x}\mathbf{x}^T)\preceq L^2(I_K\otimes\mathbf{x})(I_K\otimes\mathbf{x}^T)$. Our second idea is to relate the matrix $\mathbf{G}_t$ to the Gram matrix of actions

$$\mathbf{V}_t := \kappa\lambda I_d + \sum_{s=1}^{t-1}\mathbf{x}_s\mathbf{x}_s^T. \tag{18}$$

Specifically, using our Assumption 2 that the minimum eigenvalue of $\mathbf{A}(\mathbf{x},\boldsymbol{\theta})$ is lower bounded by $1/\kappa$ and standard spectral properties of the Kronecker product, we prove in Lemma 12 in the appendix that $\mathbf{G}_t \succeq \frac{1}{\kappa}I_K\otimes\mathbf{V}_t$ or $\mathbf{G}_t^{-1}\preceq\kappa I_K\otimes\mathbf{V}_t^{-1}$. By properly combining the above, we achieve the following convenient upper bound:

$$\left\|[\mathbf{B}(\mathbf{x},\boldsymbol{\theta}_*,\boldsymbol{\theta})\otimes\mathbf{x}^T]\,\mathbf{G}_t^{-1/2}(\boldsymbol{\theta}_*,\boldsymbol{\theta})\right\|_2 \leq L\sqrt{\kappa}\|\mathbf{x}\|_{\mathbf{V}_t^{-1}}. \tag{19}$$

To see why this is useful, note that compared to $\mathbf{H}_t(\boldsymbol{\theta})$ in (9), which is a 'matrix-weighted' version of the Gram matrix, the definition of $\mathbf{V}_t$ in (18) is the same as the definition of the Gram matrix in linear bandits. Thus, we are now able to use standard machinery in Abbasi-Yadkori et al. (2011) to bound $\sum_{t=1}^T\|\mathbf{x}_t\|_{\mathbf{V}_t^{-1}}$.

Finally, we discuss how to control the remaining term $\left\|\mathbf{g}_t(\boldsymbol{\theta}_*)-\mathbf{g}_t(\boldsymbol{\theta})\right\|_{\mathbf{G}_t(\boldsymbol{\theta}_*,\boldsymbol{\theta})^{-1}}$. By adding and subtracting $\mathbf{g}_t(\hat{\boldsymbol{\theta}}_t)$ to and from the argument inside the norm, it suffices to bound

$$\left\|\mathbf{g}_t(\boldsymbol{\theta}_*)-\mathbf{g}_t(\hat{\boldsymbol{\theta}}_t)\right\|_{\mathbf{G}_t(\boldsymbol{\theta}_*,\boldsymbol{\theta})^{-1}} + \left\|\mathbf{g}_t(\hat{\boldsymbol{\theta}}_t)-\mathbf{g}_t(\boldsymbol{\theta})\right\|_{\mathbf{G}_t(\boldsymbol{\theta}_*,\boldsymbol{\theta})^{-1}}. \tag{20}$$

To do this, we exploit the definition of $\mathcal{C}_t(\delta)$ in (12) by first relating the $\mathbf{G}_t^{-1}(\boldsymbol{\theta}_*,\boldsymbol{\theta})$–norms with those in terms of $\mathbf{H}_t^{-1}(\boldsymbol{\theta}_*)$ and $\mathbf{H}_t^{-1}(\boldsymbol{\theta})$. To do this, we rely on the *generalized self concordance* property of the strictly convex *log-sum-exp* (lse : $\mathbb{R}^K \to \mathbb{R}\cup\infty$) function Tran-Dinh et al. (2015) $\mathrm{lse}(\mathbf{s}):=\log(1+\sum_{i=1}^K\exp(\mathbf{s}_i))$. Notice that our $\mathbf{z}(\mathbf{x},\boldsymbol{\theta})$ is the gradient of the lse function at point $[\bar{\boldsymbol{\theta}}_1^T\mathbf{x},\bar{\boldsymbol{\theta}}_2^T\mathbf{x},\dots,\bar{\boldsymbol{\theta}}_K^T\mathbf{x}]^T$, that is $\mathbf{z}(\mathbf{x},\boldsymbol{\theta})=\nabla\mathrm{lse}\left([\bar{\boldsymbol{\theta}}_1^T\mathbf{x},\bar{\boldsymbol{\theta}}_2^T\mathbf{x},\dots,\bar{\boldsymbol{\theta}}_K^T\mathbf{x}]^T\right)$. Thanks to the generalized self-concordance property of the lse, upper bounds on its Hessian matrix (essentially the matrix $\mathbf{H}_t(\boldsymbol{\theta})$) have been developed in Tran-Dinh et al. (2015); Sun and Tran-Dinh (2019). Proper use of such bounds leads to lower bounds on $\mathbf{G}_t(\boldsymbol{\theta}_*,\boldsymbol{\theta})$ as follows.

**Lemma 4** (Generalized self-concordance). *For any $\boldsymbol{\theta}_1,\boldsymbol{\theta}_2\in\Theta$, we have $(1+2S)^{-1}\mathbf{H}_t(\boldsymbol{\theta}_1)\preceq\mathbf{G}_t(\boldsymbol{\theta}_1,\boldsymbol{\theta}_2)$ and $(1+2S)^{-1}\mathbf{H}_t(\boldsymbol{\theta}_2)\preceq\mathbf{G}_t(\boldsymbol{\theta}_1,\boldsymbol{\theta}_2)$.*

The proof is given in Appendix B.1. Finally, plugging the above lower bounds on matrix $\mathbf{G}_t$ into (20) for $\boldsymbol{\theta}_*$ and $\boldsymbol{\theta}$ gives the final bound on $\Delta(\mathbf{x},\boldsymbol{\theta})$ in Lemma 1.

### 2.4 Error bound on $\Delta(\mathbf{x},\boldsymbol{\theta}_t)$

In this section, we specify $\boldsymbol{\theta}_t$ and $\epsilon_t(\mathbf{x})$ for all $t\in[T]$ and $\mathbf{x}\in\mathcal{D}$. In view of Lemma 1, the exploration bonus $\epsilon_t(\mathbf{x})$ can be set equal to the RHS of (13), only if $\boldsymbol{\theta}_t\in\mathcal{C}_t(\delta)$. Recall from the definition of $\mathcal{C}_t(\delta)$ in (12) that for $\boldsymbol{\theta}_t\in\mathcal{C}_t(\delta)$, it must be that $\boldsymbol{\theta}_t\in\Theta$. Since the ML estimator $\hat{\boldsymbol{\theta}}_t$ does not necessarily satisfy $\hat{\boldsymbol{\theta}}_t\in\Theta$, we introduce the following "feasible estimator":

$$\boldsymbol{\theta}_t := \arg\min_{\boldsymbol{\theta}\in\Theta}\left\|\mathbf{g}_t(\boldsymbol{\theta})-\mathbf{g}_t(\hat{\boldsymbol{\theta}}_t)\right\|_{\mathbf{H}_t^{-1}(\boldsymbol{\theta})}, \tag{21}$$

which is guaranteed to be in the confidence set $\mathcal{C}_t(\delta)$ for all $t\in[T]$ since $\|\mathbf{g}_t(\boldsymbol{\theta}_t)-\mathbf{g}_t(\hat{\boldsymbol{\theta}}_t)\|_{\mathbf{H}_t^{-1}(\boldsymbol{\theta})}\leq\|\mathbf{g}_t(\boldsymbol{\theta}_*)-\mathbf{g}_t(\hat{\boldsymbol{\theta}}_t)\|_{\mathbf{H}_t^{-1}(\boldsymbol{\theta})}\leq\beta_t(\delta)$ and $\boldsymbol{\theta}_t\in\Theta$. Thus, we have proved the following.

**Corollary 1** (Exploration bonus). *For all $\mathbf{x}\in\mathcal{D}$ and $t\in[T]$, with probability at least $1-\delta$, we have*

$$\Delta(\mathbf{x},\boldsymbol{\theta}_t)\leq\epsilon_t(\mathbf{x}):=2RL\beta_t(\delta)\sqrt{\kappa(1+2S)}\|\mathbf{x}\|_{\mathbf{V}_t^{-1}}. \tag{22}$$

**Algorithm 1:** MNL-UCB
___
1 **for** $t = 1, \ldots, T$ **do**
2     Compute $\boldsymbol{\theta}_t$ as in (21).
3     Compute $\mathbf{x}_t := \arg\max_{\mathbf{x} \in \mathcal{D}} \boldsymbol{\rho}^T \mathbf{z}(\mathbf{x}, \boldsymbol{\theta}_t) + \epsilon_t(\mathbf{x})$ with $\epsilon_t(\mathbf{x})$ defined in (22).
4     Play $\mathbf{x}_t$ and observe $y_t$.
___

With these, we are now ready to summarize MNL-UCB in Algorithm 1.

**Remark 1.** The projection step in (21) is similar to the ones used in Filippi et al. (2010); Faury et al. (2020) for binary logistic bandits. In particular, this step in Filippi et al. (2010) involves norms with respect to $\mathbf{V}_t$ instead of $\mathbf{H}_t(\boldsymbol{\theta})$ (for $K = 1$). All of these involve non-convex optimization problems. Empirically, we observe that it occurs frequently that $\hat{\boldsymbol{\theta}}_t \in \Theta$. Then, $\boldsymbol{\theta}_t = \hat{\boldsymbol{\theta}}_t$ and these complicated projection steps do not need to be implemented.

## 2.5 Regret bound of MNL-UCB

In the following theorem, we state the regret bound of MNL-UCB as our main result.

**Theorem 2** (Regret of MNL-UCB). *Fix $\delta \in (0, 1)$. Let Assumptions 1 and 2 hold. Then, with probability at least $1 - \delta$, it holds that $R_T \leq 4RL\beta_T(\delta)\sqrt{2\max(1, \frac{1}{\lambda\kappa})\kappa\,(1 + 2S)\,dT\log(1 + \frac{T}{\kappa\lambda d})}$. In particular, choosing $\lambda = Kd\log(T)$ yields $R_T \leq \mathcal{O}\left(RLKd\log(T)\sqrt{\kappa T}\right)$.*

Now we comment on the regret order with respect to the key problem parameters $T, d, K$ and $\sqrt{\kappa}$. With respect to these the theorem shows a scaling of the regret of Algorithm 1 as $\mathcal{O}(Kd\log(T)\sqrt{\kappa T})$. Specifically, for $K = 1$ we retrieve the exact same scaling as in Theorem 2 of Faury et al. (2020) for the binary case in terms of $T, d$ and $\kappa$. In particular, the bound is optimal with respect to the action-space dimension $d$ as $\mathcal{O}(d)$ is the optimal order in the simpler setting of linear bandits Dani et al. (2008). Of course, compared to Faury et al. (2020) our result applies for general $K \geq 1$ and implies a linear scaling with the number $K$ of possible outcomes that can be selected by the user. In fact, our bound suggests our algorithm on a $K$-multinomial problem has performance of same order as the performance of the algorithm of Faury et al. (2020) for a binary problem with dimension $Kd$ instead of $d$. On the one hand, this is intuitive since the MNL reward model indeed involves $Kd$ unknown parameters. Thus, we cannot expect regret better than $\mathcal{O}(Kd)$ for $Kd$ unknown parameters. On the other hand, the MNL is a special case of multi-index models rather than a GLM. Thus, it is a-priori unclear whether it differs from a binary logistic model with $Kd$ parameters in terms of regret performance. In fact, our proof does *not* treat the MNL reward model as a GLM. Despite that, it results to the linear order $\mathcal{O}(K)$ same as the optimal order of a binary logistic setting with $Kd$ unknown parameters. Finally, as previously mentioned, the parameter $\kappa$ defined in Assumption 2 generalizes the corresponding parameter for binary bandits. As in Theorem 2 of Faury et al. (2020) our bound for the multinomial case scales with $\sqrt{\kappa}$. Next, we show how this scaling is non-trivial and improves upon standard approaches.

**Remark 2.** Using same tools as for GLM-UCB in Filippi et al. (2010) for single-index reward models ($K = 1$), we can define the following alternative confidence set around the parameter $\boldsymbol{\theta}_*$ of the MNL model:

$$\mathcal{E}_t(\delta) := \left\{\boldsymbol{\theta} \in \Theta : \|\boldsymbol{\theta} - \tilde{\boldsymbol{\theta}}_t\|_{\tilde{\mathbf{V}}_t^{-1}} \leq \kappa\gamma_t(\delta)\right\}, \tag{23}$$

where $\gamma_t(\delta)$ is a slowly increasing function of $t$ with similar order as $\beta_t(\delta)$ (see Lemma 14 in appendix), $\tilde{\mathbf{V}}_t = I_K \otimes \mathbf{V}_t$, and $\tilde{\boldsymbol{\theta}}_t := \arg\min_{\boldsymbol{\theta} \in \Theta} \|\mathbf{g}_t(\boldsymbol{\theta}) - \mathbf{g}_t(\hat{\boldsymbol{\theta}}_t)\|_{\tilde{\mathbf{V}}_t^{-1}}$. Due to the appearance of an extra $\kappa$ factor above compared to (12), relying on $\mathcal{E}_t(\delta)$ (23) in our analysis would lead to the following error bound (see Appendix C.2):

$$\Delta(\mathbf{x}, \tilde{\boldsymbol{\theta}}_t) \leq \tilde{\epsilon}_t(\mathbf{x}) := 2RL\kappa\gamma_t(\delta)\|\mathbf{x}\|_{\mathbf{V}_t^{-1}}. \tag{24}$$

This bound is significantly looser compared to our bound in (22) since the parameter $\kappa$ can become arbitrarily large depending on the size of set $\mathcal{D} \times \Theta$.

## 2.6 Improved MNL-UCB

The regret of MNL-UCB scales as $\mathcal{O}(\sqrt{\kappa})$ with respect to $\kappa$. A more careful treatment of $\mathcal{C}_t(\delta)$ in (12) leads to Improved MNL-UCB, a modification of MNL-UCB, with a newly defined estimator and improved exploration bonus. Compared to MNL-UCB, the new algorithm is computationally intractable. Yet, it is interesting from a theory perspective as we show a regret bound where the $\kappa$-dependence is pushed into a quickly-vanishing second order term, trading-off a slight increase in the regret's dependence on $K$. In particular, we prove improved regret of order $\mathcal{O}(K^{1.5}d\sqrt{T}\log(T) + \kappa K^2 d\log(T))$. Due to space limitations, we defer a detailed description (see Algorithm 2) and analysis (see Theorem 3) of Improved MNL-UCB in Appendix D.

## 3 Discussion on $\kappa$ and $L$

Knowledge of the problem-dependent constants $\kappa$ and $L$ is required to implement MNL-UCB as they appear in the definition of $\epsilon_t(\mathbf{x})$ in (22). While specifying their true values (defined in Assumption 2) requires solving non-convex optimization problems in general, here we present computationally efficient ways to obtain simple upper bounds. Also, we show that $\kappa$ can indeed scale poorly with the size of $\mathcal{D} \times \Theta$ by deriving an appropriate lower bound for it. The upper bound on $L$ is rather straightforward and it can be easily checked that $L \leq \max_{\mathbf{x} \in \mathcal{D}} \frac{e^{S\|\mathbf{x}\|_2}}{1+e^{S\|\mathbf{x}\|_2}+(K-1)e^{-S\|\mathbf{x}\|_2}}$. Next, we focus on upper/lower bounding $\kappa$, which is more interesting.

In (26), we will show that $\kappa$ scales unfavorably with the size of the set $\mathcal{D} \times \Theta$. In our regret analysis in the previous sections, it was useful to assume in Assumption 1 that $\|\mathbf{x}\|_2 \leq 1$. This assumption was without loss of generality because of the following. Suppose a general setting with $X = \max_{\mathbf{x} \in \mathcal{D}} \|\mathbf{x}\|_2 > 1$. We can then define an equivalent MNL model with actions $\tilde{\mathbf{x}} = \mathbf{x}/X$ and new parameter vector $\tilde{\boldsymbol{\theta}} = X\boldsymbol{\theta}$. The new problem satisfies the unit norm constraint on the radius of $\mathcal{D}$ and has a new parameter norm bound $\tilde{S} = SX$. For clarity with regards to the goal of this section, it is convenient to keep track of the radius of $\mathcal{D}$ (rather than push it in $S$). Thus, we let $X := \max_{\mathbf{x} \in \mathcal{D}} \|\mathbf{x}\|_2$ (possibly large). We will prove that $\kappa$ grows at least exponentially in $SX$, thus it can be very large if the action decision set is large.

In order to bound $\kappa$, we identify and take advantage of the following key property of $\mathbf{A}(\mathbf{x}, \boldsymbol{\theta})$ stated as Lemma 5. Recall that a matrix $\mathbf{A} \in \mathbb{R}^{K \times K}$ is *strictly diagonally dominant* if each of its diagonal entries is greater than the sum of absolute values of all other entries in the corresponding row/column. We also need the definition of an $M$-*matrix*: A matrix $\mathbf{A}$ is an $M$-matrix if all its off-diagonal entries are non-positive and the real parts of its eigenvalues are non-negative Berman and Plemmons (1994).

**Lemma 5.** *For any* $\mathbf{x} \in \mathbb{R}^d$ *and* $\boldsymbol{\theta} \in \mathbb{R}^{Kd}$*, the matrix* $\mathbf{A}(\mathbf{x}, \boldsymbol{\theta})$ *in* (10) *is a strictly diagonally dominant* $M$-*matrix.*

This key observation (see Appendix C for the proof) allows us to use Theorem 1.1 in Tian and Huang (2010) that provides upper and lower bounds on the minimum eigenvalue of a strictly diagonally dominant $M$-matrix. Specifically, we find that for all all $\mathbf{x}$ and $\boldsymbol{\theta}$:

$$\min_{i \in [K]} \sum_{j=1}^{K} \mathbf{A}(\mathbf{x}, \boldsymbol{\theta})_{ij} \leq \lambda_{\min}(\mathbf{A}(\mathbf{x}, \boldsymbol{\theta})) \leq \max_{i \in [K]} \sum_{j=1}^{K} \mathbf{A}(\mathbf{x}, \boldsymbol{\theta})_{ij}. \tag{25}$$

Starting from this and setting $X := \max_{\mathbf{x} \in \mathcal{D}} \|\mathbf{x}\|_2$ we show the following bounds (Appendix C.4):

$$e^{\frac{SX}{\sqrt{K}}}(1 + Ke^{-\frac{SX}{\sqrt{K}}})^2 \leq \kappa \leq e^{SX}(1 + Ke^{SX})^2. \tag{26}$$

## 4 Experiments

We present numerical simulations to complement and confirm our theoretical findings. In all experiments, we used the upper bound on $\kappa$ in (26) to compute the exploration bonus $\epsilon_t(\mathbf{x})$.

We evaluate the performance of MNL-UCB on synthetic data. All the results shown depict averages over 20 realizations, for which we have chosen $\delta = 0.01$, $d = 2$, and $T = 1000$. We considered time-independent decision sets $\mathcal{D}$ of 20 arms in $\mathbb{R}^2$ and the reward vector $\boldsymbol{\rho} = [1, \dots, K]^T$. Moreover,

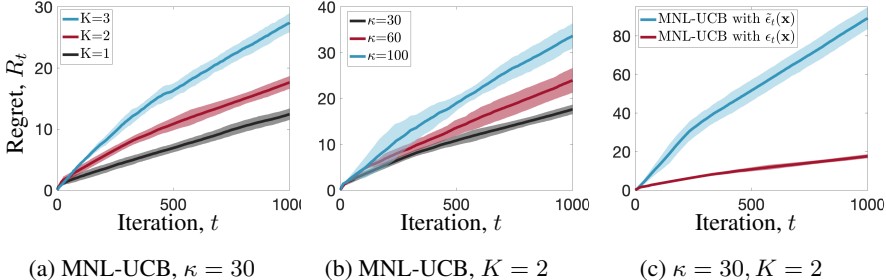

| (a) MNL-UCB, $\kappa = 30$ | (b) MNL-UCB, $K = 2$ | (c) $\kappa = 30, K = 2$ |

Figure 1: The shaded regions show standard deviation around the average over 20 problem realizations. See text for detailed description.

the arms and $\bar{\boldsymbol{\theta}}_{*i}$ are drawn from $\mathcal{N}(0, I_d)$ and $\mathcal{N}(0, I_d/K)$, respectively. The normalization of the latter by $K$ is so that it guarantees that the problem's signal-to-noise ratio $\|\boldsymbol{\theta}_*\|_2$ does not change with varying $K$. Figure 1a depicts the average regret of MNL-UCB for problem settings with different values of $K = 1, 2, 3$. The plot verifies that larger $K$ leads to larger regret and seems to match the proved scaling of the regret bound of MNL-UCB as $\mathcal{O}(K)$ with respect to $K$. Figure 1b showcases the average regret of MNL-UCB for problem settings with fixed $K = 2$ and different values of $\kappa = 30, 60, 100$ (upper bounds computed using (26)). Observe that larger $\kappa$ leads to larger regret. This is consistent with our theoretical findings on the impacts of $\kappa$ on the algorithm's performance. Finally, Figure 1c emphasizes the value of using the exploration bonus $\epsilon_t(\mathbf{x})$ in (22) compared to $\tilde{\epsilon}_t(\mathbf{x})$ introduced in (24) in the UCB decision making step. In this figure, we fixed $K = 2$ and the average regret curves are associated with a problem setting with $\kappa = 30$. A comparison between regret curves further confirms the worse regret performance of MNL-UCB when it exploits $\tilde{\boldsymbol{\theta}}_t$ and $\tilde{\epsilon}_t(\mathbf{x})$ rather than $\boldsymbol{\theta}_t$ and $\epsilon_t(\mathbf{x})$ in the UCB decision rule at Line 3 of the Algorithm 1.

## 5    Conclusion

For the MNL regression bandit problem, we developed MNL-UCB and showed a regret $\tilde{\mathcal{O}}(Kd\sqrt{\kappa}\sqrt{T})$ that scales favorably with the critical problem-dependent parameter $\kappa$ and optimally with respect to the number of options $K$. We further proposed Improved MNL-UCB that achieves a regret bound with problem-dependent constant $\kappa$ being pushed into a second order logarithmic term, trading-off a slight increase in the regret's dependence on $K$. After this work was completed, we became aware of Abeille et al. (2021) that improves Faury et al. (2020) in the binary case. Our work shows that extension of Faury et al. (2020) to MNL model is non-trivial and requires several careful analysis adjustments. It is an exciting future direction to investigate whether the algorithmic improvements of Abeille et al. (2021) can be properly adjusted in the multiclass case of interest here. In light of our discussion after Theorem 2 in Section 2.5, another promising future direction is to determine a lower bound that formally justifies the optimal dependence is $\mathcal{O}(K)$ or determine an improved algorithm/analysis that lowers the dependency on $K$. It is also interesting to study the efficacy of Thompson sampling-based algorithms for this new problem. Also, extending our results to other multi-index models is yet another important future direction.

### Acknowledgments and Disclosure of Funding

This work is supported in part by the National Science Foundation under Grant Number (1934641). Part of this work was completed while the second author was affiliated with the University of California, Santa Barbara.

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
