## Notation

Before presenting the proofs of our results, we recall some key notation for the reader's convenience.

First, recall that $\boldsymbol{\theta} = \begin{bmatrix} \bar{\boldsymbol{\theta}}_1 \\ \bar{\boldsymbol{\theta}}_2 \\ \vdots \\ \bar{\boldsymbol{\theta}}_K \end{bmatrix} \in \mathbb{R}^{Kd}$, where $\bar{\boldsymbol{\theta}}_i \in \mathbb{R}^d$.

**MNL model.**

$$\mathrm{lse}(\mathbf{s}) := \log(1 + \sum_{i=1}^{K} \exp(\mathbf{s}_i)).$$

$$\mathbf{z}(\mathbf{x}, \boldsymbol{\theta}) = \nabla \mathrm{lse}\left([\bar{\boldsymbol{\theta}}_1^T \mathbf{x}, \bar{\boldsymbol{\theta}}_2^T \mathbf{x}, \ldots, \bar{\boldsymbol{\theta}}_K^T \mathbf{x}]^T\right) = \frac{1}{1 + \sum_{j=1}^{K} \exp(\bar{\boldsymbol{\theta}}_j^T \mathbf{x})} \begin{bmatrix} \exp(\bar{\boldsymbol{\theta}}_1^T \mathbf{x}) \\ \exp(\bar{\boldsymbol{\theta}}_2^T \mathbf{x}) \\ \vdots \\ \exp(\bar{\boldsymbol{\theta}}_K^T \mathbf{x}) \end{bmatrix}.$$

$$\mathbf{A}(\mathbf{x}, \boldsymbol{\theta}) = \nabla^2 \mathrm{lse}\left([\bar{\boldsymbol{\theta}}_1^T \mathbf{x}, \bar{\boldsymbol{\theta}}_2^T \mathbf{x}, \ldots, \bar{\boldsymbol{\theta}}_K^T \mathbf{x}]^T\right) = \mathrm{diag}(\mathbf{z}(\mathbf{x}, \boldsymbol{\theta})) - \mathbf{z}(\mathbf{x}, \boldsymbol{\theta})\mathbf{z}(\mathbf{x}, \boldsymbol{\theta})^{\mathrm{T}}.$$

**Confidence set.**

$$\mathbf{H}_t(\boldsymbol{\theta}) := \lambda I_{Kd} + \sum_{s=1}^{t-1} \mathbf{A}(\mathbf{x}_s, \boldsymbol{\theta}) \otimes \mathbf{x}_s \mathbf{x}_s^T$$

$$\mathbf{g}_t(\boldsymbol{\theta}) := \lambda \boldsymbol{\theta} + \sum_{s=1}^{t-1} \mathbf{z}(\mathbf{x}_s, \boldsymbol{\theta}) \otimes \mathbf{x}_s.$$

Recall the definition of $\nabla_{\boldsymbol{\theta}} \mathcal{L}_t^\lambda(\boldsymbol{\theta})$ in (8), which we repeat here for the reader's convenience:

$$\nabla_{\boldsymbol{\theta}} \mathcal{L}_t^\lambda(\boldsymbol{\theta}) := \sum_{s=1}^{t-1} [\mathbf{m}_s - \mathbf{z}(\mathbf{x}_s, \boldsymbol{\theta})] \otimes \mathbf{x}_s - \lambda \boldsymbol{\theta}.$$

By definition, $\hat{\boldsymbol{\theta}}_t$ satisfies $\nabla_{\boldsymbol{\theta}} \mathcal{L}_t^\lambda(\hat{\boldsymbol{\theta}}_t) = \mathbf{0}$.

$$\mathcal{C}_t(\delta) := \left\{ \boldsymbol{\theta} \in \Theta : \left\| \mathbf{g}_t(\boldsymbol{\theta}) - \mathbf{g}_t(\hat{\boldsymbol{\theta}}_t) \right\|_{\mathbf{H}_t^{-1}(\boldsymbol{\theta})} \leq \beta_t(\delta) \right\}.$$

## A  Proof of Theorem 1

In this section, we present the proof of Theorem 1.

To see what is necessary to prove Theorem 1, we start with the following standard argument to analyze $\left\| \mathbf{g}_t(\boldsymbol{\theta}_*) - \mathbf{g}_t(\hat{\boldsymbol{\theta}}_t) \right\|_{\mathbf{H}_t^{-1}(\boldsymbol{\theta}^*)}$. By definition, $\hat{\boldsymbol{\theta}}_t$ satisfies $\nabla_{\boldsymbol{\theta}} \mathcal{L}_t^\lambda(\hat{\boldsymbol{\theta}}_t) = \mathbf{0}$. Consequently, it holds that

$$\mathbf{g}_t(\boldsymbol{\theta}_*) - \mathbf{g}_t(\hat{\boldsymbol{\theta}}_t) = \lambda \boldsymbol{\theta}_* + \mathbf{s}_t, \tag{27}$$

where we defined

$$\boldsymbol{\epsilon}_s := \mathbf{z}(\mathbf{x}_s, \boldsymbol{\theta}_*) - \mathbf{m}_s \quad \text{and} \quad \mathbf{s}_t := \sum_{s=1}^{t-1} \boldsymbol{\epsilon}_s \otimes \mathbf{x}_s. \tag{28}$$

Assumption 1 and simple linear algebra then imply that

$$\left\| \mathbf{g}_t(\boldsymbol{\theta}_*) - \mathbf{g}_t(\hat{\boldsymbol{\theta}}_t) \right\|_{\mathbf{H}_t^{-1}(\boldsymbol{\theta}^*)} \leq \|\mathbf{s}_t\|_{\mathbf{H}_t^{-1}(\boldsymbol{\theta}^*)} + \sqrt{\lambda}S. \tag{29}$$

Therefore, establishing an appropriate confidence set $\mathcal{C}_t(\delta)$ requires us controlling $\|\mathbf{s}_t\|_{\mathbf{H}_t^{-1}(\boldsymbol{\theta}^*)}$. We do this in several steps organized in a series of key lemmas presented next. 1.

## A.1 Necessary lemmas

The following is a matrix version of Lemma 7 in Faury et al. (2020) that uses a rather standard argument to bound the moment generating function of a random vector with bounded components in terms of its covariance matrix.

**Lemma 6.** *Let $\boldsymbol{\epsilon} \in \mathbb{R}^K$ be a zero-mean random vector with covariance matrix $\mathbf{C}$. Then, for any vector $\boldsymbol{\eta} \in \mathbb{R}^K$ such that $|\boldsymbol{\epsilon}^T\boldsymbol{\eta}| \leq 1$, we have*

$$\mathbb{E}\left[\exp(\boldsymbol{\epsilon}^T\boldsymbol{\eta})\right] \leq \exp\left(\boldsymbol{\eta}^T\mathbf{C}\boldsymbol{\eta}\right). \tag{30}$$

*Proof.* Starting with Taylor expansion of the exponential, we have the following chain of inequalities

$$\mathbb{E}\left[\exp(\boldsymbol{\epsilon}^T\boldsymbol{\eta})\right] = 1 + \mathbb{E}\left[\boldsymbol{\epsilon}^T\boldsymbol{\eta}\right] + \sum_{k=2}^{\infty} \frac{\mathbb{E}\left[(\boldsymbol{\epsilon}^T\boldsymbol{\eta})^k\right]}{k!}$$

$$= 1 + \sum_{k=2}^{\infty} \frac{\mathbb{E}\left[(\boldsymbol{\epsilon}^T\boldsymbol{\eta})^k\right]}{k!} \qquad (\mathbb{E}[\boldsymbol{\epsilon}] = \mathbf{0})$$

$$\leq 1 + \sum_{k=2}^{\infty} \frac{\mathbb{E}\left[(\boldsymbol{\epsilon}^T\boldsymbol{\eta})^2\right]}{k!} \qquad (|\boldsymbol{\epsilon}^T\boldsymbol{\eta}| \leq 1)$$

$$= 1 + \left(\boldsymbol{\eta}^T\mathbf{C}\boldsymbol{\eta}\right)\sum_{k=2}^{\infty} \frac{1}{k!} \leq 1 + \boldsymbol{\eta}^T\mathbf{C}\boldsymbol{\eta},$$

$$\leq \exp\left(\boldsymbol{\eta}^T\mathbf{C}\boldsymbol{\eta}\right). \qquad (1 + x \leq \exp(x), \forall x \in \mathbb{R})$$

This completes the proof. $\qquad\square$

Hereafter, we will denote the $p$-dimensional unit ball by $\mathcal{B}_2(p) := \{\boldsymbol{\nu} \in \mathbb{R}^p : \|\boldsymbol{\nu}\|_2 \leq 1\}$.

**Lemma 7.** *For all $\boldsymbol{\xi} \in \frac{1}{\sqrt{K}}\mathcal{B}_2(Kd)$, let $M_0(\boldsymbol{\xi}) = 1$ and for $t > 1$*

$$M_t(\boldsymbol{\xi}) = \exp(\boldsymbol{\xi}^T\mathbf{s}_t - \|\boldsymbol{\xi}\|_{\bar{\mathbf{H}}_t(\boldsymbol{\theta}_*)}^2), \tag{31}$$

*where $\bar{\mathbf{H}}_t(\boldsymbol{\theta}_*) := \sum_{s=1}^{t-1} \mathbf{A}(\mathbf{x}_s, \boldsymbol{\theta}_*) \otimes \mathbf{x}_s\mathbf{x}_s^T$. Then $\{M_t(\boldsymbol{\xi})\}_{t=1}^{\infty}$ is a non-negative super-martingale.*

*Proof.* For $t > 1$, let $\boldsymbol{\eta}_t(\boldsymbol{\xi}) = [\bar{\boldsymbol{\xi}}_1^T\mathbf{x}_t, \ldots, \bar{\boldsymbol{\xi}}_K^T\mathbf{x}_t]$. For all $\boldsymbol{\xi} \in \frac{1}{\sqrt{K}}\mathcal{B}_2(Kd)$ and $t > 1$, we have

$$\mathbb{E}\left[M_t(\boldsymbol{\xi})|\mathcal{F}_{t-1}\right] = \mathbb{E}\left[\exp(\boldsymbol{\xi}^T\mathbf{s}_t)|\mathcal{F}_{t-1}\right]\exp\left(-\|\boldsymbol{\xi}\|_{\bar{\mathbf{H}}_t(\boldsymbol{\theta}_*)}^2\right)$$

$$= \mathbb{E}\left[\exp(\boldsymbol{\xi}^T[\boldsymbol{\epsilon}_{t-1} \otimes \mathbf{x}_{t-1}])|\mathcal{F}_{t-1}\right]\exp\left(\boldsymbol{\xi}^T\mathbf{s}_{t-1} - \|\boldsymbol{\xi}\|_{\bar{\mathbf{H}}_t(\boldsymbol{\theta}_*)}^2\right)$$

$$= \mathbb{E}\left[\exp\left(\boldsymbol{\epsilon}_{t-1}^T\boldsymbol{\eta}_{t-1}(\boldsymbol{\xi})\right)|\mathcal{F}_{t-1}\right]\exp\left(\boldsymbol{\xi}^T\mathbf{s}_{t-1} - \|\boldsymbol{\xi}\|_{\bar{\mathbf{H}}_t(\boldsymbol{\theta}_*)}^2\right). \tag{32}$$

Note that $\mathbb{E}\left[\boldsymbol{\epsilon}_t|\mathcal{F}_{t-1}\right] = \mathbf{0}$ and $\mathbb{E}\left[\boldsymbol{\epsilon}_t\boldsymbol{\epsilon}_t^T|\mathcal{F}_{t-1}\right] = \mathbf{A}(\mathbf{x}_t,\boldsymbol{\theta}_*)$ for all $t > 1$. Furthermore, $|\boldsymbol{\epsilon}_{t-1}^T\boldsymbol{\eta}_{t-1}(\boldsymbol{\xi})|\leq 1$ because

$$|\boldsymbol{\epsilon}_{t-1}^T\boldsymbol{\eta}_{t-1}(\boldsymbol{\xi})| \leq \|\boldsymbol{\epsilon}_{t-1}\|_2\|\boldsymbol{\eta}_{t-1}\|_2 \qquad \text{(Cauchy-Schwarz)}$$

$$\leq \sqrt{K}\sqrt{\sum_{i=1}^{K}\left(\bar{\boldsymbol{\xi}}_i^T\mathbf{x}_t\right)^2} \qquad (|[\boldsymbol{\epsilon}_{t-1}]_i|\leq 1)$$

$$= \sqrt{K}\sqrt{\mathbf{x}_t^T\left(\sum_{i=1}^{K}\bar{\boldsymbol{\xi}}_i\bar{\boldsymbol{\xi}}_i^T\right)\mathbf{x}_t}$$

$$\leq \|\mathbf{x}_t\|_2\sqrt{K\lambda_{\max}\left(\sum_{i=1}^{K}\bar{\boldsymbol{\xi}}_i\bar{\boldsymbol{\xi}}_i^T\right)}$$

$$\leq \sqrt{K\mathrm{Tr}\left(\sum_{i=1}^{K}\bar{\boldsymbol{\xi}}_i\bar{\boldsymbol{\xi}}_i^T\right)} \qquad \text{(Assumption 1)}$$

$$= \sqrt{K\sum_{i=1}^{K}\|\bar{\boldsymbol{\xi}}_i\|_2^2}$$

$$= \sqrt{K\|\boldsymbol{\xi}\|_2^2}$$

$$\leq 1. \qquad (\|\boldsymbol{\xi}\|_2 \leq \tfrac{1}{\sqrt{K}})$$

Thus, using Lemma 6, we conclude that

$$\mathbb{E}\left[\exp\left(\boldsymbol{\epsilon}_{t-1}^T\boldsymbol{\eta}_{t-1}(\boldsymbol{\xi})\right)|\mathcal{F}_{t-1}\right] \leq 1 + \boldsymbol{\eta}_{t-1}^T(\boldsymbol{\xi})\mathbf{A}(\mathbf{x}_{t-1},\boldsymbol{\theta}_*)\boldsymbol{\eta}_{t-1}(\boldsymbol{\xi})$$

$$\leq \exp\left(\boldsymbol{\xi}^T\left(\mathbf{A}(\mathbf{x}_{t-1},\boldsymbol{\theta}_*)\otimes\left[\mathbf{x}_{t-1}\mathbf{x}_{t-1}^T\right]\right)\boldsymbol{\xi}\right), \qquad (33)$$

Combining (32) and (33), we have

$$\mathbb{E}\left[M_t(\boldsymbol{\xi})|\mathcal{F}_{t-1}\right] \leq \exp\left(\boldsymbol{\xi}^T\mathbf{s}_{t-1} - \|\boldsymbol{\xi}\|_{\bar{\mathbf{H}}_t(\boldsymbol{\theta}_*)}^2 + \boldsymbol{\xi}^T\left(\mathbf{A}(\mathbf{x}_{t-1},\boldsymbol{\theta}_*)\otimes\left[\mathbf{x}_{t-1}\mathbf{x}_{t-1}^T\right]\right)\boldsymbol{\xi}\right) = M_{t-1}(\boldsymbol{\xi}),$$
$$(34)$$

as desired. $\qquad\square$

Now, let $h(\boldsymbol{\xi})$ be a probability measure with support on $\frac{1}{\sqrt{K}}\mathcal{B}_2(Kd)$ and define

$$\bar{M}_t = \int_{\boldsymbol{\xi}\in\frac{1}{\sqrt{K}}\mathcal{B}_2(Kd)} M_t(\boldsymbol{\xi})dh(\boldsymbol{\xi}) = \int_{\boldsymbol{\xi}\in\frac{1}{\sqrt{K}}\mathcal{B}_2(Kd)} \exp(\boldsymbol{\xi}^T\mathbf{s}_t - \|\boldsymbol{\xi}\|_{\bar{\mathbf{H}}_t(\boldsymbol{\theta}_*)}^2)\,dh(\boldsymbol{\xi}). \qquad (35)$$

By Lemma 20.3 in Lattimore and Szepesvári (2020) $\bar{M}_t$ is also a non-negative super-martingale and $\mathbb{E}\left[\bar{M}_0\right] = 1$. Let $\tau$ be the stopping point with respect to the filtration $\{\mathcal{F}_t\}_{t=0}^{\infty}$. By Lemma 8 Abbasi-Yadkori et al. (2011), $\bar{M}_\tau$ is well-defined and $\mathbb{E}\left[\bar{M}_\tau\right] \leq 1$. Thanks to Markov inequality, for any $\delta \in (0, 1)$, we have

$$\mathbb{P}(\bar{M}_\tau \geq \frac{1}{\delta}) \leq \delta\mathbb{E}[\bar{M}_\tau] \leq \delta. \qquad (36)$$

**Lemma 8.** *Let $h(\boldsymbol{\xi})$ be the density of an isotropic normal distribution with precision matrix (i.e. inverse covariance matrix) $2\lambda I_{Kd}$ that is truncated on $\frac{1}{\sqrt{K}}\mathcal{B}_2(Kd)$ and $g(\boldsymbol{\xi})$ be the density of the*

*normal distribution with precision matrix* $2\mathbf{H}_t(\boldsymbol{\theta}_*)$ *that is truncated on the ball* $\frac{1}{2\sqrt{K}}\mathcal{B}_2(Kd)$. *Denote the normalization constant of* $h$ *and* $g$ *by* $Z(h)$ *and* $Z(g)$, *respectively. Then*

$$\mathbb{P}\left(\|\mathbf{s}_t\|_{\mathbf{H}_t^{-1}(\boldsymbol{\theta}_*)} \geq \frac{\sqrt{\lambda/K}}{2} + 2\sqrt{K/\lambda}\log\left(\frac{Z(h)}{\delta Z(g)}\right)\right) \leq \delta. \tag{37}$$

*Proof.* We prove this lemma using similar techniques employed in Abbasi-Yadkori et al. (2011); Faury et al. (2020). We first specify $Z(h)$ and $Z(g)$ for completeness as follows:

$$Z(h) := \int_{\left\{\boldsymbol{\xi}\in\mathbb{R}^{Kd}:\|\boldsymbol{\xi}\|_2 \leq \sqrt{\frac{1}{K}}\right\}} \exp\left(-\lambda\|\boldsymbol{\xi}\|_2^2\right) d\boldsymbol{\xi},$$

$$Z(g) := \int_{\left\{\boldsymbol{\xi}\in\mathbb{R}^{Kd}:\|\boldsymbol{\xi}\|_2 \leq \frac{1}{2\sqrt{K}}\right\}} \exp\left(-\frac{1}{2}\|\boldsymbol{\xi}\|_{2\mathbf{H}_t(\boldsymbol{\theta}_*)}^2\right) d\boldsymbol{\xi}.$$

By the definition of $\bar{M}_t$ in (35) and recalling $\mathbf{H}_t(\boldsymbol{\theta}_*) = \lambda\mathbf{I}_{Kd} + \bar{\mathbf{H}}_t(\boldsymbol{\theta}_*)$, we have

$$\bar{M}_t = \frac{1}{Z(h)} \int_{\boldsymbol{\xi}\in\frac{1}{\sqrt{K}}\mathcal{B}_2(Kd)} \exp(\boldsymbol{\xi}^T\mathbf{s}_t - \|\boldsymbol{\xi}\|_{\mathbf{H}_t(\boldsymbol{\theta}_*)}^2) d\boldsymbol{\xi}. \tag{38}$$

Now, let $f(\boldsymbol{\xi}) = \boldsymbol{\xi}^T\mathbf{s}_t - \|\boldsymbol{\xi}\|_{\mathbf{H}_t(\boldsymbol{\theta}_*)}^2$ and $\boldsymbol{\xi}_* = \arg\max_{\|\boldsymbol{\xi}\|_2 \leq \frac{1}{2\sqrt{K}}} f(\boldsymbol{\xi})$. Since

$$f(\boldsymbol{\xi}) = f(\boldsymbol{\xi}_*) + (\boldsymbol{\xi} - \boldsymbol{\xi}_*)^T\nabla f(\boldsymbol{\xi}_*) - (\boldsymbol{\xi} - \boldsymbol{\xi}_*)^T\mathbf{H}_t(\boldsymbol{\theta}_*)(\boldsymbol{\xi} - \boldsymbol{\xi}_*), \tag{39}$$

we have

$$\bar{M}_t = \frac{\exp(f(\boldsymbol{\xi}_*))}{Z(h)} \int_{\mathbb{R}^{Kd}} \mathbb{1}\left\{\|\boldsymbol{\xi}\|_2 \leq \frac{1}{\sqrt{K}}\right\} \exp\left((\boldsymbol{\xi} - \boldsymbol{\xi}_*)^T\nabla f(\boldsymbol{\xi}_*) - \|\boldsymbol{\xi} - \boldsymbol{\xi}_*\|_{\mathbf{H}_t(\boldsymbol{\theta}_*)}^2\right) d\boldsymbol{\xi}$$

$$= \frac{\exp(f(\boldsymbol{\xi}_*))}{Z(h)} \int_{\mathbb{R}^{Kd}} \mathbb{1}\left\{\|\boldsymbol{\xi} + \boldsymbol{\xi}_*\|_2 \leq \frac{1}{\sqrt{K}}\right\} \exp\left(\boldsymbol{\xi}^T\nabla f(\boldsymbol{\xi}_*) - \|\boldsymbol{\xi}\|_{\mathbf{H}_t(\boldsymbol{\theta}_*)}^2\right) d\boldsymbol{\xi}$$

$$\text{(change of variable)}$$

$$\geq \frac{\exp(f(\boldsymbol{\xi}_*))}{Z(h)} \int_{\mathbb{R}^{Kd}} \mathbb{1}\left\{\|\boldsymbol{\xi}\|_2 \leq \frac{1}{2\sqrt{K}}\right\} \exp\left(\boldsymbol{\xi}^T\nabla f(\boldsymbol{\xi}_*) - \|\boldsymbol{\xi}\|_{\mathbf{H}_t(\boldsymbol{\theta}_*)}^2\right) d\boldsymbol{\xi} \quad (\|\boldsymbol{\xi}_*\|_2 \leq \frac{1}{2\sqrt{K}})$$

$$= \exp(f(\boldsymbol{\xi}_*))\frac{Z(g)}{Z(h)}\mathbb{E}_g\left[\exp\left(\boldsymbol{\xi}^T\nabla f(\boldsymbol{\xi}_*)\right)\right] \qquad\qquad \text{(definition of } g(\boldsymbol{\xi}))$$

$$\geq \exp(f(\boldsymbol{\xi}_*))\frac{Z(g)}{Z(h)}\exp\left(\mathbb{E}_g\left[\boldsymbol{\xi}^T\nabla f(\boldsymbol{\xi}_*)\right]\right) \qquad\qquad \text{(Jensen's inequality)}$$

$$= \exp(f(\boldsymbol{\xi}_*))\frac{Z(g)}{Z(h)}, \tag{40}$$

where in the last equality we used the fact that $\mathbb{E}_g[\boldsymbol{\xi}] = \mathbf{0}$. Now, let $\boldsymbol{\xi}_0 = \frac{\mathbf{H}_t^{-1}(\boldsymbol{\theta}_*)\mathbf{s}_t}{\|\mathbf{s}_t\|_{\mathbf{H}_t^{-1}(\boldsymbol{\theta}_*)}} \times \frac{\sqrt{\lambda/K}}{2}$. Hence

$$\|\boldsymbol{\xi}_0\|_2 = \frac{\sqrt{\lambda/K}}{2} \cdot \frac{\|\mathbf{s}_t\|_{\mathbf{H}_t^{-2}(\boldsymbol{\theta}_*)}}{\|\mathbf{s}_t\|_{\mathbf{H}_t^{-1}(\boldsymbol{\theta}_*)}} \leq \frac{\sqrt{\lambda/K}}{2} \cdot \frac{1}{\sqrt{\lambda}} = \frac{1}{2\sqrt{K}}. \tag{41}$$

Combining (36) and (40), we conclude that

$$\mathbb{P}\left(\boldsymbol{\xi}_0^T\mathbf{s}_t - \|\boldsymbol{\xi}_0\|_{\mathbf{H}_t(\boldsymbol{\theta}_*)}^2 \geq \log(\frac{1}{\delta}) + \log\left(\frac{Z(h)}{Z(g)}\right)\right) \leq \mathbb{P}\left(\max_{\|\boldsymbol{\xi}\|_2 \leq \frac{1}{2\sqrt{K}}} \boldsymbol{\xi}^T\mathbf{s}_t - \|\boldsymbol{\xi}\|_{\mathbf{H}_t(\boldsymbol{\theta}_*)}^2 \geq \log(\frac{1}{\delta}) + \log\left(\frac{Z(h)}{Z(g)}\right)\right)$$

$$= \mathbb{P}\left(\exp\left(f(\boldsymbol{\xi}_*)\right)\frac{Z(g)}{Z(h)} \geq \frac{1}{\delta}\right)$$

$$\leq \mathbb{P}\left(\bar{M}_t \geq \frac{1}{\delta}\right) \leq \delta. \tag{42}$$

Thus, by the definition of $\boldsymbol{\xi}_0$, we have

$$\mathbb{P}\left(\frac{\sqrt{\lambda/K}}{2}\|\mathbf{s}_t\|_{\mathbf{H}_t^{-1}(\boldsymbol{\theta}_*)} \geq \frac{\lambda}{4K} + \log\left(\frac{Z(h)}{\delta Z(g)}\right)\right) = \mathbb{P}\left(\|\mathbf{s}_t\|_{\mathbf{H}_t^{-1}(\boldsymbol{\theta}_*)} \geq \frac{\sqrt{\lambda/K}}{2} + 2\sqrt{K/\lambda}\log\left(\frac{Z(h)}{\delta Z(g)}\right)\right) \leq \delta,$$
(43)

as desired. □

We use the following lemma borrowed from Faury et al. (2020) to complete the proof of Theorem 1.

**Lemma 9** (Lemma 6 in Faury et al. (2020)). *It holds that*

$$\log\left(\frac{Z(h)}{\delta Z(g)}\right) \leq \log\left(\frac{\det\left(\mathbf{H}_t^{1/2}(\boldsymbol{\theta}_*)\right)}{\lambda^{Kd/2}}\right) + Kd\log(2/\delta)$$
(44)

## A.2 Completing the proof of Theorem 1

We are now ready to complete the proof of Theorem 1 combining the key results presented above as follows:

$$1 - \delta \leq \mathbb{P}\left(\|\mathbf{s}_t\|_{\mathbf{H}_t^{-1}(\boldsymbol{\theta}^*)} + \sqrt{\lambda}S \leq \frac{\sqrt{\lambda/K}}{2} + 2\sqrt{K/\lambda}\log\left(\frac{Z(h)}{\delta Z(g)}\right) + \sqrt{\lambda}S\right) \qquad \text{(Lemma 8)}$$

$$\leq \mathbb{P}\left(\|\mathbf{s}_t\|_{\mathbf{H}_t^{-1}(\boldsymbol{\theta}^*)} + \sqrt{\lambda}S \leq \frac{\sqrt{\lambda/K}}{2} + 2\sqrt{K/\lambda}\log\left(\frac{\det\left(\mathbf{H}_t^{1/2}(\boldsymbol{\theta}_*)\right)}{\lambda^{Kd/2}}\right) + \frac{2K^{3/2}d}{\sqrt{\lambda}}\log(2/\delta) + \sqrt{\lambda}S\right)$$
(Lemma 9)

$$\leq \mathbb{P}\left(\left\|\mathbf{g}_t(\boldsymbol{\theta}_*) - \mathbf{g}_t(\hat{\boldsymbol{\theta}}_t)\right\|_{\mathbf{H}_t^{-1}(\boldsymbol{\theta}^*)} \leq \frac{\sqrt{\lambda/K}}{2} + 2\sqrt{K/\lambda}\log\left(\frac{\det\left(\mathbf{H}_t^{1/2}(\boldsymbol{\theta}_*)\right)}{\lambda^{Kd/2}}\right) + \frac{2K^{3/2}d}{\sqrt{\lambda}}\log(2/\delta) + \sqrt{\lambda}S\right)$$
(Eqn. (29))

$$\leq \mathbb{P}\left(\left\|\mathbf{g}_t(\boldsymbol{\theta}_*) - \mathbf{g}_t(\hat{\boldsymbol{\theta}}_t)\right\|_{\mathbf{H}_t^{-1}(\boldsymbol{\theta}^*)} \leq \frac{K^{3/2}d}{\sqrt{\lambda}}\log\left(1 + \frac{t}{d\lambda}\right) + \frac{\sqrt{\lambda/K}}{2} + \frac{2K^{3/2}d}{\sqrt{\lambda}}\log(2/\delta) + \sqrt{\lambda}S\right),$$
(45)

where the last inequality follows from

$$\det\left(\mathbf{H}_t(\boldsymbol{\theta}_*)\right) = \prod_{i=1}^{Kd}\lambda_i(\mathbf{H}_t(\boldsymbol{\theta}_*)) \leq \left(\text{trace}\left(\mathbf{H}_t(\boldsymbol{\theta}_*)\right)/Kd\right)^{Kd},$$
(46)

and

$$\text{trace}\left(\mathbf{H}_t(\boldsymbol{\theta}_*)\right) \leq \lambda Kd + \sum_{s=1}^{t-1}\sum_{i=1}^{K}\mathbf{A}(\mathbf{x}_s, \boldsymbol{\theta}_*)_{ii}\|\mathbf{x}_s\|_2^2 \leq \lambda Kd + Kt. \qquad \text{(Assumption 1)}$$

# B  Proofs of Lemma 1 and Theorem 2

As mentioned in Section 2.3, we organize the proof of Lemma 1 in several key lemmas which we prove below

**Conditioning on $\boldsymbol{\theta}_* \in \mathcal{C}_t(\delta)$, $\forall t \in [T]$.**  Consider the event

$$\mathcal{E} := \{\boldsymbol{\theta}_* \in \mathcal{C}_t(\delta), \ \forall t \in [T]\},$$
(47)

that $\boldsymbol{\theta}_*$ is inside the confidence sets for all rounds $t \in [T]$. By Theorem 1 the event holds with probability at least $1 - \delta$. Onwards, we condition on this event, and make repeated use of the fact that $\boldsymbol{\theta}_* \in \mathcal{C}_t$ for all $t \in [T]$, without further explicit reference.

### B.1 Necessary lemmas

Here, we prove Lemmas 2, 3, and 4 of Section 2.3. For the reader's convenience, we repeat their statements as Lemmas 10, 11, and 13 below.

**Lemma 10.** *Recall the definition of the* lse *function:* $\mathrm{lse}(\mathbf{s}) := \log(1 + \sum_{i=1}^{K} \exp(\mathbf{s}_i))$ *and let* $\boldsymbol{\mu} : \mathbb{R}^K \to \mathbb{R}^K$ *be defined as*

$$\boldsymbol{\mu}(\mathbf{s}) := \nabla \mathrm{lse}(\mathbf{s}) = \frac{1}{1 + \sum_{i=1}^{K} \exp(\mathbf{s}_i)} \begin{bmatrix} \exp(\mathbf{s}_1) \\ \vdots \\ \exp(\mathbf{s}_K) \end{bmatrix}. \tag{48}$$

*Moreover, for* $\mathbf{x} \in \mathbb{R}^d$ *and* $\boldsymbol{\theta}_1, \boldsymbol{\theta}_2 \in \mathbb{R}^{Kd}$ *define*

$$\mathbf{B}(\mathbf{x}, \boldsymbol{\theta}_1, \boldsymbol{\theta}_2) := \int_0^1 \nabla \boldsymbol{\mu} \left( \begin{bmatrix} v\bar{\boldsymbol{\theta}}_{11}^T \mathbf{x} + (1-v)\bar{\boldsymbol{\theta}}_{21}^T \mathbf{x} \\ v\bar{\boldsymbol{\theta}}_{12}^T \mathbf{x} + (1-v)\bar{\boldsymbol{\theta}}_{22}^T \mathbf{x} \\ \vdots \\ v\bar{\boldsymbol{\theta}}_{1K}^T \mathbf{x} + (1-v)\bar{\boldsymbol{\theta}}_{2K}^T \mathbf{x} \end{bmatrix} \right) dv = \int_0^1 \mathbf{A}(\mathbf{x}, v\boldsymbol{\theta}_1 + (1-v)\boldsymbol{\theta}_2) dv, \tag{49}$$

*where the matrix* $\mathbf{A}(\mathbf{x}, \boldsymbol{\theta})$ *is given in* (10). *Then, it holds that*

$$\mathbf{z}(\mathbf{x}, \boldsymbol{\theta}_1) - \mathbf{z}(\mathbf{x}, \boldsymbol{\theta}_2) = \left[ \mathbf{B}(\mathbf{x}, \boldsymbol{\theta}_1, \boldsymbol{\theta}_2) \otimes \mathbf{x}^T \right] (\boldsymbol{\theta}_1 - \boldsymbol{\theta}_2). \tag{50}$$

**Remark 3** (Notation). In the notation of the lemma, notice that our vector function $\mathbf{z}(\mathbf{x}, \boldsymbol{\theta})$ defined in (2) can be expressed in terms of $\boldsymbol{\mu}$ as:

$$\mathbf{z}(\mathbf{x}, \boldsymbol{\theta}) = \boldsymbol{\mu}([\bar{\boldsymbol{\theta}}_1^T \mathbf{x}, \dots, \bar{\boldsymbol{\theta}}_K^T \mathbf{x}]^T).$$

Similarly,

$$\mathbf{A}(\mathbf{x}, \boldsymbol{\theta}) = \nabla \boldsymbol{\mu}([\bar{\boldsymbol{\theta}}_1^T \mathbf{x}, \dots, \bar{\boldsymbol{\theta}}_K^T \mathbf{x}]^T).$$

*Proof.* According to *mean-value Theorem*, for a differentiable function $\mathbf{f} : \mathbb{R}^P \to \mathbb{R}^Q$ and differentiable function $\mathbf{q} : \mathbb{R} \to \mathbb{R}^P$, we have

$$\int_a^b \nabla \mathbf{f}^T(\mathbf{q}(v)) \mathbf{q}'(v) dv = \mathbf{f}(\mathbf{q}(b)) - \mathbf{f}(\mathbf{q}(a)). \tag{51}$$

Now, let $\mathbf{f}(\boldsymbol{\theta}) = \mathbf{z}(\mathbf{x}, \boldsymbol{\theta})$, $a = 0$, $b = 1$ and $\mathbf{q}(v) = v\boldsymbol{\theta}_1 + (1-v)\boldsymbol{\theta}_2$. By the mean-value theorem, we have

$$\mathbf{z}(\mathbf{x}, \boldsymbol{\theta}_1) - \mathbf{z}(\mathbf{x}, \boldsymbol{\theta}_2) = \left[ \mathbf{B}(\mathbf{x}, \boldsymbol{\theta}_1, \boldsymbol{\theta}_2) \otimes \mathbf{x} \right]^T (\boldsymbol{\theta}_1 - \boldsymbol{\theta}_2) = \left[ \mathbf{B}(\mathbf{x}, \boldsymbol{\theta}_1, \boldsymbol{\theta}_2) \otimes \mathbf{x}^T \right] (\boldsymbol{\theta}_1 - \boldsymbol{\theta}_2), \tag{52}$$

where the last equality follows from the fact that $(\mathbf{A} \otimes \mathbf{B})^T = \mathbf{A}^T \otimes \mathbf{B}^T$ for any matrices $\mathbf{A}$ and $\mathbf{B}$ and matrix $\mathbf{B}(\mathbf{x}, \boldsymbol{\theta}_1, \boldsymbol{\theta}_2) \in \mathbb{R}^{K \times K}$ being a symmetric matrix. $\square$

**Lemma 11.** *Define*

$$\mathbf{G}_t(\boldsymbol{\theta}_1, \boldsymbol{\theta}_2) := \lambda I_{Kd} + \sum_{s=1}^{t-1} \mathbf{B}(\mathbf{x}_s, \boldsymbol{\theta}_1, \boldsymbol{\theta}_2) \otimes \mathbf{x}_s \mathbf{x}_s^T. \tag{53}$$

*Then, for any* $\boldsymbol{\theta}_1, \boldsymbol{\theta}_2 \in \mathbb{R}^{Kd}$, *we have*

$$\mathbf{g}_t(\boldsymbol{\theta}_1) - \mathbf{g}_t(\boldsymbol{\theta}_2) = \mathbf{G}_t(\boldsymbol{\theta}_1, \boldsymbol{\theta}_2)(\boldsymbol{\theta}_1 - \boldsymbol{\theta}_2). \tag{54}$$

*Proof.* By the definition of $\mathbf{g}_t(\boldsymbol{\theta})$ in (11) we have

$$\mathbf{g}_t(\boldsymbol{\theta}_1) - \mathbf{g}_t(\boldsymbol{\theta}_2) = \lambda(\boldsymbol{\theta}_1 - \boldsymbol{\theta}_2) + \sum_{s=1}^{t-1}[\mathbf{z}(\mathbf{x}_s, \boldsymbol{\theta}_1) - \mathbf{z}(\mathbf{x}_s, \boldsymbol{\theta}_2)] \otimes \mathbf{x}_s$$

$$= \lambda(\boldsymbol{\theta}_1 - \boldsymbol{\theta}_2) + \sum_{s=1}^{t-1}\left[\left[\mathbf{B}(\mathbf{x}_s, \boldsymbol{\theta}_1, \boldsymbol{\theta}_2) \otimes \mathbf{x}_s^T\right](\boldsymbol{\theta}_1 - \boldsymbol{\theta}_2)\right] \otimes \mathbf{x}_s \qquad \text{(Eqn. (50))}$$

$$= \lambda(\boldsymbol{\theta}_1 - \boldsymbol{\theta}_2) + \sum_{s=1}^{t-1}\left[\left[\mathbf{B}(\mathbf{x}_s, \boldsymbol{\theta}_1, \boldsymbol{\theta}_2) \otimes \mathbf{x}_s^T\right] \otimes \mathbf{x}_s\right](\boldsymbol{\theta}_1 - \boldsymbol{\theta}_2)$$

$$\text{(mixed-product property)}$$

$$= \lambda(\boldsymbol{\theta}_1 - \boldsymbol{\theta}_2) + \sum_{s=1}^{t-1}\left[\mathbf{B}(\mathbf{x}_s, \boldsymbol{\theta}_1, \boldsymbol{\theta}_2) \otimes \left[\mathbf{x}_s^T \otimes \mathbf{x}_s\right]\right](\boldsymbol{\theta}_1 - \boldsymbol{\theta}_2)$$

$$= \lambda(\boldsymbol{\theta}_1 - \boldsymbol{\theta}_2) + \sum_{s=1}^{t-1}\left[\mathbf{B}(\mathbf{x}_s, \boldsymbol{\theta}_1, \boldsymbol{\theta}_2) \otimes \mathbf{x}_s\mathbf{x}_s^T\right](\boldsymbol{\theta}_1 - \boldsymbol{\theta}_2) = \mathbf{G}_t(\boldsymbol{\theta}_1, \boldsymbol{\theta}_2)(\boldsymbol{\theta}_1 - \boldsymbol{\theta}_2),$$

as desired. $\qquad\qquad\qquad\qquad\qquad\qquad\qquad\qquad\qquad\qquad\qquad\qquad\qquad\qquad\square$

**Lemma 12.** *Define*

$$\tilde{\mathbf{V}}_t := \kappa\lambda I_{Kd} + \sum_{s=1}^{t-1}\sum_{i=1}^{K}(\mathbf{e}_i \otimes \mathbf{x}_s)(\mathbf{e}_i \otimes \mathbf{x}_s)^T = I_K \otimes \mathbf{V}_t. \qquad (55)$$

*Then, for any $\boldsymbol{\theta}_1, \boldsymbol{\theta}_2 \in \Theta$, we have*

$$\frac{1}{\kappa}\tilde{\mathbf{V}}_t \preceq \mathbf{G}_t(\boldsymbol{\theta}_1, \boldsymbol{\theta}_2).$$

*Proof.* By definition of $\mathbf{G}_t(\boldsymbol{\theta}_1, \boldsymbol{\theta}_2)$ for any $\boldsymbol{\theta}_1, \boldsymbol{\theta}_2 \in \Theta$ in (53), we have

$$\mathbf{G}_t(\boldsymbol{\theta}_1, \boldsymbol{\theta}_2) = \lambda I_{Kd} + \sum_{s=1}^{t-1}\mathbf{B}(\mathbf{x}_s, \boldsymbol{\theta}_1, \boldsymbol{\theta}_2) \otimes \mathbf{x}_s\mathbf{x}_s^T$$

$$\succeq \frac{1}{\kappa}\left[\kappa\lambda I_{Kd} + \sum_{s=1}^{t-1}I_K \otimes \mathbf{x}_s\mathbf{x}_s^T\right] \qquad \text{(Assumption 2)}$$

$$= \frac{1}{\kappa}\tilde{\mathbf{V}}_t. \qquad (56)$$

We derived the inequality above as follows. Assumption 2 and definition of $\mathbf{B}(\mathbf{x}, \boldsymbol{\theta}_1, \boldsymbol{\theta}_2)$ in (49) give $\mathbf{B}(\mathbf{x}, \boldsymbol{\theta}_1, \boldsymbol{\theta}_2) \succeq \frac{1}{\kappa}I_K$. Now, from standard spectral properties of the Kronecker product, the eigenvalues of the matrix $\mathbf{B}(\mathbf{x}, \boldsymbol{\theta}_1, \boldsymbol{\theta}_2) \otimes \mathbf{x}_s\mathbf{x}_s^T - \frac{1}{\kappa}I_K \otimes \mathbf{x}_s\mathbf{x}_s^T = (\mathbf{B}(\mathbf{x}, \boldsymbol{\theta}_1, \boldsymbol{\theta}_2) - \frac{1}{\kappa}I_K) \otimes \mathbf{x}_s\mathbf{x}_s^T$ are the pairwise products of the individual eigenvalues of $\mathbf{B}(\mathbf{x}, \boldsymbol{\theta}_1, \boldsymbol{\theta}_2) - \frac{1}{\kappa}I_K$ and of $\mathbf{x}_s\mathbf{x}_s^T$. The desired inequality then follows since both of the latter two matrices are positive semi-definite. $\quad\square$

**Lemma 13** (Generalized self-concordance)**.** *For any $\boldsymbol{\theta}_1, \boldsymbol{\theta}_2 \in \Theta$, we have*

$$(1 + 2S)^{-1}\mathbf{H}_t(\boldsymbol{\theta}_1) \preceq \mathbf{G}_t(\boldsymbol{\theta}_1, \boldsymbol{\theta}_2) \quad \text{and} \quad (1 + 2S)^{-1}\mathbf{H}_t(\boldsymbol{\theta}_2) \preceq \mathbf{G}_t(\boldsymbol{\theta}_1, \boldsymbol{\theta}_2). \qquad (57)$$

*Proof.* Recall the function $\mathrm{lse} : \mathbb{R}^K \to \mathbb{R}$

$$\mathrm{lse}(\mathbf{s}) := \log\left(1 + \sum_{i=1}^{K}\exp(\mathbf{s}_i)\right). \qquad (58)$$

Then its gradient and Hessian are written as

$$\nabla\mathrm{lse}(\mathbf{s}) = \boldsymbol{\mu}(\mathbf{s}) := \frac{1}{1 + \sum_{i=1}^{K}\exp(\mathbf{s}_i)}\begin{bmatrix} \exp(\mathbf{s}_1) \\ \vdots \\ \exp(\mathbf{s}_K) \end{bmatrix} \quad \text{and} \quad \nabla^2\mathrm{lse}(\mathbf{s}) = \mathrm{diag}(\mathbf{s}) - \mathbf{s}\mathbf{s}^T. \qquad (59)$$

As shown in Lemma 4 in Tran-Dinh et al. (2015), the function lse is $(M_{\text{lse}}, \nu)$-generalized self-concordant with $\nu = 1$ and $M_{\text{lse}} = \sqrt{6}$. Therefore, according to Corollary 2 in Sun and Tran-Dinh (2019), for any $\mathbf{x} \in \mathcal{D}$ and $\boldsymbol{\theta}_1, \boldsymbol{\theta}_2 \in \Theta$, we have

$$\frac{1 - \exp\left(-d(\mathbf{x}, \boldsymbol{\theta}_1, \boldsymbol{\theta}_2)\right)}{d(\mathbf{x}, \boldsymbol{\theta}_1, \boldsymbol{\theta}_2)} \nabla^2 f \left( \begin{bmatrix} \bar{\boldsymbol{\theta}}_{21}^T \mathbf{x} \\ \bar{\boldsymbol{\theta}}_{22}^T \mathbf{x} \\ \vdots \\ \bar{\boldsymbol{\theta}}_{2K}^T \mathbf{x} \end{bmatrix} \right) \preceq \int_0^1 \nabla^2 f \left( \begin{bmatrix} v\bar{\boldsymbol{\theta}}_{11}^T \mathbf{x} + (1-v)\bar{\boldsymbol{\theta}}_{21}^T \mathbf{x} \\ v\bar{\boldsymbol{\theta}}_{12}^T \mathbf{x} + (1-v)\bar{\boldsymbol{\theta}}_{22}^T \mathbf{x} \\ \vdots \\ v\bar{\boldsymbol{\theta}}_{1K}^T \mathbf{x} + (1-v)\bar{\boldsymbol{\theta}}_{2K}^T \mathbf{x} \end{bmatrix} \right) dv, \quad (60)$$

where $d(\mathbf{x}, \boldsymbol{\theta}_1, \boldsymbol{\theta}_2) = \left\| \left[ \bar{\boldsymbol{\theta}}_{11}^T \mathbf{x} - \bar{\boldsymbol{\theta}}_{21}^T \mathbf{x}, \ldots, \bar{\boldsymbol{\theta}}_{1K}^T \mathbf{x} - \bar{\boldsymbol{\theta}}_{2K}^T \mathbf{x} \right]^T \right\|_2$. Now, by the definition of $\mathbf{A}(\mathbf{x}, \boldsymbol{\theta})$ and $\mathbf{B}(\mathbf{x}, \boldsymbol{\theta}_1, \boldsymbol{\theta}_2)$ in (10) and (49), we conclude from (60) that

$$\mathbf{B}(\mathbf{x}, \boldsymbol{\theta}_1, \boldsymbol{\theta}_2) \succeq \left(1 + d(\mathbf{x}, \boldsymbol{\theta}_1, \boldsymbol{\theta}_2)\right)^{-1} \mathbf{A}(\mathbf{x}, \boldsymbol{\theta}_2) \qquad (\tfrac{1-\exp(-t)}{t} \geq (1+t)^{-1})$$

$$\succeq (1 + 2S)^{-1} \mathbf{A}(\mathbf{x}, \boldsymbol{\theta}_2), \qquad (61)$$

where the last inequality follows from Assumption 1 and the fact that both $\boldsymbol{\theta}_1, \boldsymbol{\theta}_2 \in \Theta$. Combining (61) and the definition of $\mathbf{G}_t(\boldsymbol{\theta}_1, \boldsymbol{\theta}_2)$ in (53) for any $\boldsymbol{\theta}_1, \boldsymbol{\theta}_2 \in \Theta$, we have

$$\mathbf{G}_t(\boldsymbol{\theta}_1, \boldsymbol{\theta}_2) = \lambda I_{Kd} + \sum_{s=1}^{t-1} \mathbf{B}(\mathbf{x}_s, \boldsymbol{\theta}_1, \boldsymbol{\theta}_2) \otimes \mathbf{x}_s \mathbf{x}_s^T$$

$$\succeq (1 + 2S)^{-1} \left[ \lambda I_{Kd} + \sum_{s=1}^{t-1} \mathbf{A}(\mathbf{x}_s, \boldsymbol{\theta}_2) \otimes \mathbf{x}_s \mathbf{x}_s^T \right]$$

$$= (1 + 2S)^{-1} \mathbf{H}_t(\boldsymbol{\theta}_2). \qquad \text{(by definition of } \mathbf{H}_t(\boldsymbol{\theta}_2) \text{ in Eqn. (9))}$$

By symmetric roles of $\boldsymbol{\theta}_1$ and $\boldsymbol{\theta}_2$ in the definition of $\mathbf{B}(\mathbf{x}, \boldsymbol{\theta}_1, \boldsymbol{\theta}_2)$, we can similarly prove that $(1 + 2S)^{-1} \mathbf{H}_t(\boldsymbol{\theta}_1) \preceq \mathbf{G}_t(\boldsymbol{\theta}_1, \boldsymbol{\theta}_2)$. $\qquad\square$

## B.2 Completing the proof of Lemma 1

In this section, we complete the proof of Lemma 1 using Lemmas 10, 11, 12 and 13.

By the definition of $\Delta(\mathbf{x}, \boldsymbol{\theta})$, for all $\mathbf{x} \in \mathcal{D}$, $t \in [T]$ and $\boldsymbol{\theta} \in \mathcal{C}_t(\delta)$, we have the following chain of inequalities

$$
\begin{aligned}
\Delta(\mathbf{x}, \boldsymbol{\theta}) &= |\boldsymbol{\rho}^T \mathbf{z}(\mathbf{x}, \boldsymbol{\theta}_*) - \boldsymbol{\rho}^T \mathbf{z}(\mathbf{x}, \boldsymbol{\theta})| \\
&\leq R \big\| \mathbf{z}(\mathbf{x}, \boldsymbol{\theta}_*) - \mathbf{z}(\mathbf{x}, \boldsymbol{\theta}) \big\|_2 \qquad \text{(Assumption 1 and Cauchy-Schwarz inequality)} \\
&= R \Big\| \big[ \mathbf{B}(\mathbf{x}, \boldsymbol{\theta}_*, \boldsymbol{\theta}) \otimes \mathbf{x}^T \big] (\boldsymbol{\theta}_* - \boldsymbol{\theta}) \Big\|_2 \qquad \text{(Lemma 10)} \\
&= R \Big\| \big[ \mathbf{B}(\mathbf{x}, \boldsymbol{\theta}_*, \boldsymbol{\theta}) \otimes \mathbf{x}^T \big] \mathbf{G}_t^{-1/2}(\boldsymbol{\theta}_*, \boldsymbol{\theta}) \mathbf{G}_t^{1/2}(\boldsymbol{\theta}_*, \boldsymbol{\theta})(\boldsymbol{\theta}_* - \boldsymbol{\theta}) \Big\|_2 \\
&\leq R \Big\| \big[ \mathbf{B}(\mathbf{x}, \boldsymbol{\theta}_*, \boldsymbol{\theta}) \otimes \mathbf{x}^T \big] \mathbf{G}_t^{-1/2}(\boldsymbol{\theta}_*, \boldsymbol{\theta}) \Big\|_2 \| \boldsymbol{\theta}_* - \boldsymbol{\theta} \|_{\mathbf{G}_t(\boldsymbol{\theta}_*, \boldsymbol{\theta})} \\
&\qquad\qquad\qquad\qquad\qquad\qquad\qquad\qquad\qquad\qquad\qquad \text{(Cauchy-Schwarz inequality)} \\
&= R \sqrt{\lambda_{\max}\Big( \big[ \mathbf{B}(\mathbf{x}, \boldsymbol{\theta}_*, \boldsymbol{\theta}) \otimes \mathbf{x}^T \big] \mathbf{G}_t^{-1}(\boldsymbol{\theta}_*, \boldsymbol{\theta}) \big[ \mathbf{B}^T(\mathbf{x}, \boldsymbol{\theta}_*, \boldsymbol{\theta}) \otimes \mathbf{x} \big] \Big)} \| \mathbf{g}_t(\boldsymbol{\theta}_*) - \mathbf{g}_t(\boldsymbol{\theta}) \|_{\mathbf{G}_t^{-1}(\boldsymbol{\theta}_*, \boldsymbol{\theta})} \\
&\qquad\qquad\qquad\qquad\qquad\qquad\qquad\qquad\qquad\qquad\qquad\qquad\qquad \text{(Lemma 11)} \\
&= R \sqrt{\lambda_{\max}\Big( \mathbf{G}_t^{-1/2}(\boldsymbol{\theta}_*, \boldsymbol{\theta}) \big[ \mathbf{B}^T(\mathbf{x}, \boldsymbol{\theta}_*, \boldsymbol{\theta}) \otimes \mathbf{x} \big] \big[ \mathbf{B}(\mathbf{x}, \boldsymbol{\theta}_*, \boldsymbol{\theta}) \otimes \mathbf{x}^T \big] \mathbf{G}_t^{-1/2}(\boldsymbol{\theta}_*, \boldsymbol{\theta}) \Big)} \| \mathbf{g}_t(\boldsymbol{\theta}_*) - \mathbf{g}_t(\boldsymbol{\theta}) \|_{\mathbf{G}_t^{-1}(\boldsymbol{\theta}_*, \boldsymbol{\theta})} \\
&\qquad\qquad\qquad\qquad\qquad\qquad\qquad\qquad\qquad\qquad\qquad\qquad \text{(cyclic property of } \lambda_{\max}) \\
&= R \sqrt{\lambda_{\max}\Big( \mathbf{G}_t^{-1/2}(\boldsymbol{\theta}_*, \boldsymbol{\theta}) \big[ \mathbf{B}^T(\mathbf{x}, \boldsymbol{\theta}_*, \boldsymbol{\theta}) \mathbf{B}(\mathbf{x}, \boldsymbol{\theta}_*, \boldsymbol{\theta}) \otimes \mathbf{x}\mathbf{x}^T \big] \mathbf{G}_t^{-1/2}(\boldsymbol{\theta}_*, \boldsymbol{\theta}) \Big)} \| \mathbf{g}_t(\boldsymbol{\theta}_*) - \mathbf{g}_t(\boldsymbol{\theta}) \|_{\mathbf{G}_t^{-1}(\boldsymbol{\theta}_*, \boldsymbol{\theta})} \\
&\qquad\qquad\qquad\qquad\qquad\qquad\qquad\qquad\qquad\qquad\qquad\qquad \text{(mixed-product property)} \\
&\leq RL \sqrt{\lambda_{\max}\Big( \mathbf{G}_t^{-1}(\boldsymbol{\theta}_*, \boldsymbol{\theta}) \big[ I_K \otimes \mathbf{x}\mathbf{x}^T \big] \Big)} \| \mathbf{g}_t(\boldsymbol{\theta}_*) - \mathbf{g}_t(\boldsymbol{\theta}) \|_{\mathbf{G}_t^{-1}(\boldsymbol{\theta}_*, \boldsymbol{\theta})} \\
&\qquad\qquad\qquad\qquad\qquad\qquad\qquad\qquad\qquad\qquad\qquad\qquad \text{(Assumption 2 \ (*))} \\
&= RL \sqrt{\lambda_{\max}\Big( \mathbf{G}_t^{-1}(\boldsymbol{\theta}_*, \boldsymbol{\theta}) \big[ I_K \otimes \mathbf{x} \big] \big[ I_K \otimes \mathbf{x}^T \big] \Big)} \| \mathbf{g}_t(\boldsymbol{\theta}_*) - \mathbf{g}_t(\boldsymbol{\theta}) \|_{\mathbf{G}_t^{-1}(\boldsymbol{\theta}_*, \boldsymbol{\theta})} \\
&\qquad\qquad\qquad\qquad\qquad\qquad\qquad\qquad\qquad\qquad\qquad\qquad \text{(mixed-product property)} \\
&= RL \sqrt{\lambda_{\max}\Big( \big[ I_K \otimes \mathbf{x}^T \big] \mathbf{G}_t^{-1}(\boldsymbol{\theta}_*, \boldsymbol{\theta}) \big[ I_K \otimes \mathbf{x} \big] \Big)} \| \mathbf{g}_t(\boldsymbol{\theta}_*) - \mathbf{g}_t(\boldsymbol{\theta}) \|_{\mathbf{G}_t^{-1}(\boldsymbol{\theta}_*, \boldsymbol{\theta})} \\
&\qquad\qquad\qquad\qquad\qquad\qquad\qquad\qquad\qquad\qquad\qquad\qquad \text{(cyclic property of } \lambda_{\max}) \\
&\leq RL \sqrt{\kappa \lambda_{\max}\Big( \big[ I_K \otimes \mathbf{x}^T \big] \big[ I_K \otimes \mathbf{V}_t^{-1} \big] \big[ I_K \otimes \mathbf{x} \big] \Big)} \| \mathbf{g}_t(\boldsymbol{\theta}_*) - \mathbf{g}_t(\boldsymbol{\theta}) \|_{\mathbf{G}_t^{-1}(\boldsymbol{\theta}_*, \boldsymbol{\theta})} \\
&\qquad\qquad\qquad\qquad\qquad\qquad\qquad\qquad\qquad\qquad\qquad\qquad \text{(Lemma 12 \ (**))} \\
&= RL \sqrt{\kappa \lambda_{\max}\Big( I_K \otimes \|\mathbf{x}\|_{\mathbf{V}_t^{-1}}^2 \Big)} \| \mathbf{g}_t(\boldsymbol{\theta}_*) - \mathbf{g}_t(\boldsymbol{\theta}) \|_{\mathbf{G}_t^{-1}(\boldsymbol{\theta}_*, \boldsymbol{\theta})} \quad \text{(mixed-product property)} \\
&= RL \sqrt{\kappa} \|\mathbf{x}\|_{\mathbf{V}_t^{-1}} \Big\| \mathbf{g}_t(\boldsymbol{\theta}_*) - \mathbf{g}_t(\hat{\boldsymbol{\theta}}_t) + \mathbf{g}_t(\hat{\boldsymbol{\theta}}_t) - \mathbf{g}_t(\boldsymbol{\theta}) \Big\|_{\mathbf{G}_t^{-1}(\boldsymbol{\theta}_*, \boldsymbol{\theta})} \\
&\leq RL \sqrt{\kappa} \|\mathbf{x}\|_{\mathbf{V}_t^{-1}} \Big[ \big\| \mathbf{g}_t(\boldsymbol{\theta}_*) - \mathbf{g}_t(\hat{\boldsymbol{\theta}}_t) \big\|_{\mathbf{G}_t^{-1}(\boldsymbol{\theta}_*, \boldsymbol{\theta})} + \big\| \mathbf{g}_t(\hat{\boldsymbol{\theta}}_t) - \mathbf{g}_t(\boldsymbol{\theta}) \big\|_{\mathbf{G}_t^{-1}(\boldsymbol{\theta}_*, \boldsymbol{\theta})} \Big] \\
&\qquad\qquad\qquad\qquad\qquad\qquad\qquad\qquad\qquad\qquad\qquad \big( (I_K \otimes \mathbf{V}_t)^{-1} = I_K \otimes \mathbf{V}_t^{-1} \big) \\
&\leq RL \sqrt{\kappa(1+2S)} \|\mathbf{x}\|_{\mathbf{V}_t^{-1}} \Big[ \big\| \mathbf{g}_t(\boldsymbol{\theta}_*) - \mathbf{g}_t(\hat{\boldsymbol{\theta}}_t) \big\|_{\mathbf{H}_t^{-1}(\boldsymbol{\theta}_*)} + \big\| \mathbf{g}_t(\hat{\boldsymbol{\theta}}_t) - \mathbf{g}_t(\boldsymbol{\theta}) \big\|_{\mathbf{H}_t^{-1}(\boldsymbol{\theta})} \Big] \\
&\qquad\qquad\qquad\qquad\qquad\qquad\qquad\qquad\qquad\qquad\qquad\qquad\qquad \text{(Lemma 13)} \\
&\leq 2RL\beta_t(\delta) \sqrt{\kappa(1+2S)} \|\mathbf{x}\|_{\mathbf{V}_t^{-1}} , \qquad\qquad\qquad \text{(Theorem 1)}
\end{aligned}
$$

as desired. Above, *cyclic property of* $\lambda_{\max}$ boils down to the fact that for two matrices $\mathbf{M}_1, \mathbf{M}_2 \in \mathbb{R}^{d \times d}$ the eigenvalues of $\mathbf{M}_1 \mathbf{M}_2$ are the same as the eigenvalues of $\mathbf{M}_2 \mathbf{M}_1$ [2]; thus, the same is true

---

[2] To see why this is true: let $\lambda, \mathbf{v}$ be an eigenvalue-eigenvector pair of $\mathbf{M}_1\mathbf{M}_2$, i.e. $\mathbf{M}_1\mathbf{M}_2\mathbf{v} = \lambda\mathbf{v}$. Then, $(\mathbf{M}_2\mathbf{M}_1)\mathbf{M}_2\mathbf{v} = \lambda\mathbf{M}_2\mathbf{v}$. Thus, $\lambda$ is also an eigenvalue of $\mathbf{M}_2\mathbf{M}_1$.

of the maximum eigenvalue. For completeness, we also detail the derivation of the third in line inequality marked with (*) above. First, from an argument identical to the proof of Lemma 12, we have that $\mathbf{B}^T(\mathbf{x}, \boldsymbol{\theta}_*, \boldsymbol{\theta})\mathbf{B}(\mathbf{x}, \boldsymbol{\theta}_*, \boldsymbol{\theta}) \otimes \mathbf{x}\mathbf{x}^T \preceq L^2(I_K \otimes \mathbf{x}\mathbf{x}^T)$. From this and Theorem 7.7.2 in Horn and Johnson (2012) it follows that $\mathbf{G}_t^{-1/2}(\boldsymbol{\theta}_*, \boldsymbol{\theta})[\mathbf{B}^T(\mathbf{x}, \boldsymbol{\theta}_*, \boldsymbol{\theta})\mathbf{B}(\mathbf{x}, \boldsymbol{\theta}_*, \boldsymbol{\theta}) \otimes \mathbf{x}\mathbf{x}^T]\mathbf{G}_t^{-1/2}(\boldsymbol{\theta}_*, \boldsymbol{\theta}) \preceq L^2\mathbf{G}_t^{-1/2}(\boldsymbol{\theta}_*, \boldsymbol{\theta})[I_K \otimes \mathbf{x}\mathbf{x}^T]\mathbf{G}_t^{-1/2}(\boldsymbol{\theta}_*, \boldsymbol{\theta})$. The argument then follows directly by the applying the cyclic property of $\lambda_{\max}$.

### B.3 Completing the proof of Theorem 2

Consider the standard instantaneous regret decomposition as follows:

$$
\begin{aligned}
r_t &= |\boldsymbol{\rho}^T\mathbf{z}(\mathbf{x}_*, \boldsymbol{\theta}_*) - \boldsymbol{\rho}^T\mathbf{z}(\mathbf{x}_t, \boldsymbol{\theta}_*)| \\
&\leq |\boldsymbol{\rho}^T\mathbf{z}(\mathbf{x}_*, \boldsymbol{\theta}_t) + \epsilon_t(\mathbf{x}_*) - \boldsymbol{\rho}^T\mathbf{z}(\mathbf{x}_t, \boldsymbol{\theta}_*)| \\
&\leq |\boldsymbol{\rho}^T\mathbf{z}(\mathbf{x}_t, \boldsymbol{\theta}_t) - \boldsymbol{\rho}^T\mathbf{z}(\mathbf{x}_t, \boldsymbol{\theta}_*) + \epsilon_t(\mathbf{x}_t)| \\
&\leq 2\epsilon_t(\mathbf{x}_t).
\end{aligned}
\tag{62}
$$

Therefore

$$
\begin{aligned}
R_T &\leq 2\sum_{t=1}^{T} \epsilon_t(\mathbf{x}_t) \\
&= 4RL\beta_T(\delta)\sqrt{\kappa(1+2S)}\sum_{t=1}^{T}\|\mathbf{x}_t\|_{\mathbf{V}_t^{-1}} && \text{(Corollary 1)} \\
&\leq 4RL\beta_T(\delta)\sqrt{\kappa(1+2S)}\sqrt{T\sum_{t=1}^{T}\|\mathbf{x}_t\|_{\mathbf{V}_t^{-1}}^2} && \text{(Cauchy-Schwarz inequality)} \\
&\leq 4RL\beta_T(\delta)\sqrt{2\max(1, \frac{1}{\lambda\kappa})\kappa(1+2S)\,dT\log\left(1+\frac{T}{\kappa\lambda d}\right)}.
\end{aligned}
\tag{63}
$$

In the last inequality, we used the standard argument in regret analysis of linear bandit algorithm stated in the following Abbasi-Yadkori et al. (2011) (Lemma 11):

$$
\sum_{t=1}^{n}\min\left(\|\mathbf{y}_t\|_{\mathbf{V}_t^{-1}}^2, 1\right) \leq 2\log\frac{\det \mathbf{A}_{n+1}}{\det \mathbf{A}_1} \quad \text{where} \quad \mathbf{A}_n = \mathbf{A}_1 + \sum_{t=1}^{n-1}\mathbf{y}_t\mathbf{y}_t^T.
\tag{64}
$$

## C   Miscellaneous Useful results

### C.1   A looser confidence set

**Lemma 14.** *Let* $\gamma_t(\delta) := 2\left(\sqrt{\lambda}S + 2\sqrt{\log(1/\delta) + Kd\log\left(1+\frac{t}{\kappa\lambda d}\right)}\right)$. *Then, it holds that* $\boldsymbol{\theta}_* \in \mathcal{E}_t(\delta)$ *with probability at least* $1 - \delta$.

*Proof.*

$$\left\|\boldsymbol{\theta}_* - \tilde{\boldsymbol{\theta}}_t\right\|_{\tilde{\mathbf{V}}_t} \le \sqrt{\kappa}\left\|\boldsymbol{\theta}_* - \tilde{\boldsymbol{\theta}}_t\right\|_{\mathbf{G}_t(\boldsymbol{\theta}_*, \tilde{\boldsymbol{\theta}}_t)} \qquad \text{(Lemma 12)}$$

$$= \sqrt{\kappa}\left\|\mathbf{g}_t(\boldsymbol{\theta}_*) - \mathbf{g}_t(\tilde{\boldsymbol{\theta}}_t)\right\|_{\mathbf{G}_t^{-1}(\boldsymbol{\theta}_*, \tilde{\boldsymbol{\theta}}_t)} \qquad \text{(Lemma 3)}$$

$$\le \sqrt{\kappa}\left(\left\|\mathbf{g}_t(\boldsymbol{\theta}_*) - \mathbf{g}_t(\hat{\boldsymbol{\theta}}_t)\right\|_{\mathbf{G}_t^{-1}(\boldsymbol{\theta}_*, \tilde{\boldsymbol{\theta}}_t)} + \left\|\mathbf{g}_t(\hat{\boldsymbol{\theta}}_t) - \mathbf{g}_t(\tilde{\boldsymbol{\theta}}_t)\right\|_{\mathbf{G}_t^{-1}(\boldsymbol{\theta}_*, \tilde{\boldsymbol{\theta}}_t)}\right)$$

$$\le \kappa\left(\left\|\mathbf{g}_t(\boldsymbol{\theta}_*) - \mathbf{g}_t(\hat{\boldsymbol{\theta}}_t)\right\|_{\tilde{\mathbf{V}}_t^{-1}} + \left\|\mathbf{g}_t(\hat{\boldsymbol{\theta}}_t) - \mathbf{g}_t(\tilde{\boldsymbol{\theta}}_t)\right\|_{\tilde{\mathbf{V}}_t^{-1}}\right) \qquad \text{(Lemma 12)}$$

$$\le 2\kappa\left\|\mathbf{g}_t(\boldsymbol{\theta}_*) - \mathbf{g}_t(\hat{\boldsymbol{\theta}}_t)\right\|_{\tilde{\mathbf{V}}_t^{-1}} \qquad \text{(Definition of } \tilde{\boldsymbol{\theta}}_t \text{ and } \boldsymbol{\theta}_* \in \Theta)$$

$$\le 2\kappa\left(\sqrt{\lambda}S + \|\mathbf{s}_t\|_{\tilde{\mathbf{V}}_t^{-1}}\right) \qquad (65)$$

$$\le 2\kappa\left(\sqrt{\lambda}S + 2\sqrt{\log(1/\delta) + Kd\log\left(1 + \frac{t}{\kappa\lambda d}\right)}\right).$$

$$\text{(Theorem 1 in Abbasi-Yadkori et al. (2011))}$$

$\square$

**Lemma 15.** *For all $t \in [T]$, it holds that $\mathcal{C}_t(\delta) \subseteq \tilde{\mathcal{C}}_t(\delta)$.*

*Proof.* Let $\boldsymbol{\theta} \in \mathcal{C}_t(\delta)$.

$$\|\boldsymbol{\theta} - \boldsymbol{\theta}_t\|_{\mathbf{H}_t(\boldsymbol{\theta})} \le \sqrt{1 + 2S}\|\boldsymbol{\theta} - \boldsymbol{\theta}_t\|_{\mathbf{G}_t(\boldsymbol{\theta}, \boldsymbol{\theta}_t)} \qquad \text{(Lemma 13)}$$

$$= \sqrt{1 + 2S}\left\|\mathbf{g}_t(\boldsymbol{\theta}) - \mathbf{g}_t(\boldsymbol{\theta}_t)\right\|_{\mathbf{G}_t^{-1}(\boldsymbol{\theta}, \boldsymbol{\theta}_t)} \qquad \text{(Lemma 11)}$$

$$\le \sqrt{1 + 2S}\left(\left\|\mathbf{g}_t(\boldsymbol{\theta}) - \mathbf{g}_t(\hat{\boldsymbol{\theta}}_t)\right\|_{\mathbf{G}_t^{-1}(\boldsymbol{\theta}, \boldsymbol{\theta}_t)} + \left\|\mathbf{g}_t(\hat{\boldsymbol{\theta}}_t) - \mathbf{g}_t(\boldsymbol{\theta}_t)\right\|_{\mathbf{G}_t^{-1}(\boldsymbol{\theta}, \boldsymbol{\theta}_t)}\right)$$

$$\le (1 + 2S)\left(\left\|\mathbf{g}_t(\boldsymbol{\theta}) - \mathbf{g}_t(\hat{\boldsymbol{\theta}}_t)\right\|_{\mathbf{H}_t^{-1}(\boldsymbol{\theta})} + \left\|\mathbf{g}_t(\hat{\boldsymbol{\theta}}_t) - \mathbf{g}_t(\boldsymbol{\theta}_t)\right\|_{\mathbf{H}_t^{-1}(\boldsymbol{\theta}_t)}\right)$$

$$\text{(Lemma 13)}$$

$$\le (2 + 4S)\beta_t(\delta). \qquad \text{(Theorem 1)}$$

Thus, $\boldsymbol{\theta} \in \tilde{\mathcal{C}}_t(\delta)$ which completes the proof. $\square$

## C.2 Proof of (24)

Based on this confidence set and under the event $\{\boldsymbol{\theta}_* \in \mathcal{E}_t(\delta), \ \forall t \in [T]\}$, we derive a new upper bound on $\Delta(\mathbf{x}, \boldsymbol{\theta})$ for all $\mathbf{x} \in \mathcal{D}$ and $\boldsymbol{\theta} \in \mathcal{E}_t(\delta)$ as follows:

$$
\begin{aligned}
\Delta(\mathbf{x}, \boldsymbol{\theta}) &= |\boldsymbol{\rho}^T \mathbf{z}(\mathbf{x}, \boldsymbol{\theta}_*) - \boldsymbol{\rho}^T \mathbf{z}(\mathbf{x}, \boldsymbol{\theta})| \\
&\leq R \big\| \mathbf{z}(\mathbf{x}, \boldsymbol{\theta}_*) - \mathbf{z}(\mathbf{x}, \boldsymbol{\theta}) \big\|_2 && \text{(Assumption 1 and Cauchy-Schwarz inequality)} \\
&= R \left\| \left[ \mathbf{B}(\mathbf{x}, \boldsymbol{\theta}_*, \boldsymbol{\theta}) \otimes \mathbf{x}^T \right] (\boldsymbol{\theta}_* - \boldsymbol{\theta}) \right\|_2 && \text{(Lemma 10)} \\
&= R \left\| \left[ \mathbf{B}(\mathbf{x}, \boldsymbol{\theta}_*, \boldsymbol{\theta}) \otimes \mathbf{x}^T \right] \tilde{\mathbf{V}}_t^{-1/2} \tilde{\mathbf{V}}_t^{1/2} (\boldsymbol{\theta}_* - \boldsymbol{\theta}) \right\|_2 \\
&\leq R \left\| \left[ \mathbf{B}(\mathbf{x}, \boldsymbol{\theta}_*, \boldsymbol{\theta}) \otimes \mathbf{x}^T \right] \tilde{\mathbf{V}}_t^{-1/2} \right\|_2 \| \boldsymbol{\theta}_* - \boldsymbol{\theta} \|_{\tilde{\mathbf{V}}_t} && \text{(Cauchy-Schwarz inequality)} \\
&= R \sqrt{ \lambda_{\max} \left( \left[ \mathbf{B}(\mathbf{x}, \boldsymbol{\theta}_*, \boldsymbol{\theta}) \otimes \mathbf{x}^T \right] \tilde{\mathbf{V}}_t^{-1} \left[ \mathbf{B}^T(\mathbf{x}, \boldsymbol{\theta}_*, \boldsymbol{\theta}) \otimes \mathbf{x} \right] \right) } \| \boldsymbol{\theta}_* - \boldsymbol{\theta} \|_{\tilde{\mathbf{V}}_t} \\
&\leq RL \sqrt{ \lambda_{\max} \left( \left[ I_K \otimes \mathbf{V}_t^{-1} \right] \left[ I_K \otimes \mathbf{x}\mathbf{x}^T \right] \right) } \| \boldsymbol{\theta}_* - \boldsymbol{\theta} \|_{\tilde{\mathbf{V}}_t} \\
& && \text{(Assumption 2 and cyclic property of } \lambda_{\max}) \\
&= RL \sqrt{ \lambda_{\max} \left( \left[ I_K \otimes \mathbf{V}_t^{-1} \right] \left[ I_K \otimes \mathbf{x} \right] \left[ I_K \otimes \mathbf{x}^T \right] \right) } \| \boldsymbol{\theta}_* - \boldsymbol{\theta} \|_{\tilde{\mathbf{V}}_t} \\
& && \text{(mixed-product property)} \\
&= RL \sqrt{ \lambda_{\max} \left( \left[ I_K \otimes \mathbf{x}^T \right] \left[ I_K \otimes \mathbf{V}_t^{-1} \right] \left[ I_K \otimes \mathbf{x} \right] \right) } \| \boldsymbol{\theta}_* - \boldsymbol{\theta} \|_{\tilde{\mathbf{V}}_t} \\
& && \text{(cyclic property of } \lambda_{\max}) \\
&= RL \| \mathbf{x} \|_{\mathbf{V}_t^{-1}} \left[ \left\| \boldsymbol{\theta}_* - \tilde{\boldsymbol{\theta}}_t \right\|_{\tilde{\mathbf{V}}_t} + \left\| \tilde{\boldsymbol{\theta}}_t - \boldsymbol{\theta} \right\|_{\tilde{\mathbf{V}}_t} \right] \\
&\leq 2RL\kappa\gamma_t(\delta) \| \mathbf{x} \|_{\mathbf{V}_t^{-1}}. && \text{(Lemma 14)}
\end{aligned}
$$

## C.3 A useful diagonally-dominant $M$-matrix

**Lemma 16.** *For any $\mathbf{x} \in \mathbb{R}^d$ and $\boldsymbol{\theta} \in \mathbb{R}^{Kd}$, matrix $\mathbf{A}(\mathbf{x}, \boldsymbol{\theta})$ defined in (10) is a strictly diagonally dominant $M$-matrix.*

*Proof.* Recall that $\mathbf{A}(\mathbf{x}, \boldsymbol{\theta})_{ij} := z_i(\mathbf{x}, \boldsymbol{\theta}) \left( \delta_{ij} - z_j(\mathbf{x}, \boldsymbol{\theta}) \right)$ for all $i, j \in [K]$. Thanks to the MNL probabilistic model, one can easily observe that for each $i \in [K]$ of matrix $\mathbf{A}(\mathbf{x}, \boldsymbol{\theta})$, we have:

$$
\mathbf{A}(\mathbf{x}, \boldsymbol{\theta})_{ii} = z_0(\mathbf{x}, \boldsymbol{\theta}) z_i(\mathbf{x}, \boldsymbol{\theta}) + \sum_{j \neq i} |\mathbf{A}(\mathbf{x}, \boldsymbol{\theta})_{ij}| > \sum_{j \neq i} |\mathbf{A}(\mathbf{x}, \boldsymbol{\theta})_{ij}|, \tag{66}
$$

$$
\mathbf{A}(\mathbf{x}, \boldsymbol{\theta})_{ii} = z_0(\mathbf{x}, \boldsymbol{\theta}) z_i(\mathbf{x}, \boldsymbol{\theta}) + \sum_{j \neq i} |\mathbf{A}(\mathbf{x}, \boldsymbol{\theta})_{ji}| > \sum_{j \neq i} |\mathbf{A}(\mathbf{x}, \boldsymbol{\theta})_{ji}|. \tag{67}
$$

Therefore, for any $\mathbf{x} \in \mathbb{R}^d$ and $\boldsymbol{\theta} \in \mathbb{R}^{Kd}$, matrix $\mathbf{A}(\mathbf{x}, \boldsymbol{\theta})$ is strictly diagonally dominant. Furthermore, a strictly diagonally dominant matrix which is also *symmetric* with positive diagonal entries is positive definite (Theorem 6.1.10 in Horn and Johnson (2012)). Thus, $\mathbf{A}(\mathbf{x}, \boldsymbol{\theta})$ is a positive definite matrix with positive eigenvalues, which paired with the fact that all its off-diagonal entries are negative proves that it is also an $M$-matrix. □

## C.4 Proof of (26)

For convenience, define $\tilde{S} = S/\sqrt{K}$.

**Algorithm 2:** Improved MNL-UCB

---
1 **for** $t = 1$ **to** $T$ **do**
2     Compute $\boldsymbol{\theta}_t$ as in (73).
3     Compute $\mathbf{x}_t := \arg\max_{\mathbf{x} \in \mathcal{D}} \boldsymbol{\rho}^T \mathbf{z}(\mathbf{x}, \boldsymbol{\theta}_t) + \bar{\epsilon}_t(\mathbf{x}, \boldsymbol{\theta}_t)$ with $\boldsymbol{\theta}_t$ and $\bar{\epsilon}_t(\mathbf{x}, \boldsymbol{\theta}_t)$ defined in (73)
    and (78).
4     Play $\mathbf{x}_t$ and observe $y_t$.

---

An application of Theorem 1.1 in Tian and Huang (2010) that provides upper and lower bounds on the minimum eigenvalue of a strictly diagonally dominant $M$-matrix, together with Lemma 5 give the following for all $\mathbf{x}$ and $\boldsymbol{\theta}$: $\min_{i \in [K]} \sum_{j=1}^{K} \mathbf{A}(\mathbf{x}, \boldsymbol{\theta})_{ij} \leq \lambda_{\min}(\mathbf{A}(\mathbf{x}, \boldsymbol{\theta})) \leq \max_{i \in [K]} \sum_{j=1}^{K} \mathbf{A}(\mathbf{x}, \boldsymbol{\theta})_{ij}$. By the definition of $\kappa$ in Assumption 2, the previous lower bound and the definitions of $\mathbf{A}(\mathbf{x}, \boldsymbol{\theta})$ and of the multinomial logit probability model we have then that

$$\frac{1}{\kappa} := \min_{\mathbf{x} \in \mathcal{D}, \boldsymbol{\theta} \in \Theta} \lambda_{\min}(\mathbf{A}(\mathbf{x}, \boldsymbol{\theta})) \geq \min_{\mathbf{x} \in \mathcal{D}, \boldsymbol{\theta} \in \Theta, i \in [K]} \sum_{j=1}^{K} \mathbf{A}(\mathbf{x}, \boldsymbol{\theta})_{ij}$$

$$= \min_{\mathbf{x} \in \mathcal{D}, \boldsymbol{\theta} \in \Theta, i \in [K]} \frac{e^{\bar{\boldsymbol{\theta}}_i^T \mathbf{x}}}{\left(1 + \sum_{j=1}^{K} e^{\bar{\boldsymbol{\theta}}_j^T \mathbf{x}}\right)^2} = \min_{i \in [n]} \min_{\mathbf{x} \in \mathcal{D}, \boldsymbol{\theta} \in \Theta} \frac{e^{\bar{\boldsymbol{\theta}}_i^T \mathbf{x}}}{\left(1 + \sum_{j=1}^{K} e^{\bar{\boldsymbol{\theta}}_j^T \mathbf{x}}\right)^2}$$

$$\geq \min_{i \in [n]} \frac{e^{-SX}}{(1 + Ke^{SX})^2} = \frac{e^{-SX}}{(1 + Ke^{SX})^2}. \tag{68}$$

The lower bound in the last line follows since for any $\mathbf{x} \in \mathcal{D}$ and $\boldsymbol{\theta} \in \Theta$ by Cauchy-Schwarz: $-SX \leq \boldsymbol{\theta}_i^T \mathbf{x} \leq SX$, for all $i \in [n]$. This proves the advertised upper bound on $\kappa$.

For the lower bound, we proceed similarly to find that

$$\frac{1}{\kappa} := \min_{\mathbf{x} \in \mathcal{D}, \boldsymbol{\theta} \in \Theta} \lambda_{\min}(\mathbf{A}(\mathbf{x}, \boldsymbol{\theta})) \leq \min_{\mathbf{x} \in \mathcal{D}, \boldsymbol{\theta} \in \Theta} \max_{i \in [K]} \sum_{j=1}^{K} \mathbf{A}(\mathbf{x}, \boldsymbol{\theta})_{ij}$$

$$= \min_{\mathbf{x} \in \mathcal{D}, \boldsymbol{\theta} \in \Theta} \max_{i \in [K]} \frac{e^{\bar{\boldsymbol{\theta}}_i^T \mathbf{x}}}{\left(1 + \sum_{j=1}^{K} e^{\bar{\boldsymbol{\theta}}_j^T \mathbf{x}}\right)^2}$$

$$\leq \max_{i \in [K]} \frac{e^{-\tilde{S}X}}{(1 + Ke^{-\tilde{S}X})^2} = \frac{e^{-\tilde{S}X}}{(1 + Ke^{-\tilde{S}X})^2}.$$

The inequality in the last line follows by choosing feasible $\mathbf{x}$ and $\boldsymbol{\theta}$ as follows. Let $\bar{\boldsymbol{\theta}}_1 = \ldots = \bar{\boldsymbol{\theta}}_K = \bar{\boldsymbol{\theta}}$ with $\|\bar{\boldsymbol{\theta}}\|_2 = \tilde{S}$ and $\mathbf{x} = -\frac{X}{\tilde{S}}\bar{\boldsymbol{\theta}}$. The above gives the desired upper bound and concludes the proof.

## D   Improved Multnomial Logit UCB

In this section, we exploit careful algorithmic designs to entirely drop the regret's dependency on $\kappa$ (except in logarithmic terms). First, we introduce the following assumption.

**Assumption 3** (Problem-dependent constants). *There exist $0 < M < \infty$ and $0 < M' < \infty$ such that for all $\mathbf{x} \in \mathcal{D}, \boldsymbol{\theta} \in \Theta$ and $i, j \in [K]$*

$$\left| \lambda_{\max} \left( \nabla^2 z_i(\mathbf{x}, \boldsymbol{\theta}) \right) \right| \leq M \quad and \quad \left\| \nabla \mathbf{A}(\mathbf{x}, \boldsymbol{\theta})_{ij} \right\|_2 \leq M', \tag{69}$$

*where $z_i(\mathbf{x}, \boldsymbol{\theta})$ is the $i$-th entry of vector $\mathbf{z}(\mathbf{x}, \boldsymbol{\theta})$ and $\mathbf{A}(\mathbf{x}, \boldsymbol{\theta})_{ij}$ is the $ij$-th entry of matrix $\mathbf{A}(\mathbf{x}, \boldsymbol{\theta})$.*

A close inspection of the proof presented in the previous subsections reveals that the "extra" $\sqrt{\kappa}$ factor enters the regret in Equation (**) in the display in Section B.2. Specifically this follows when

we replace the matrix $\mathbf{G}_t(\boldsymbol{\theta}_*, \boldsymbol{\theta})$ with the simpler matrix $\tilde{\mathbf{V}}_t$ (cf. Lemma 12). In turn, this is possible by replacing the key matrix $\mathbf{A}(\mathbf{x}, \boldsymbol{\theta})$ —the Jacobian of the MNL model— by the smaller —in the sense of the Loewner order— matrix $\frac{1}{\kappa}I_K$ (cf. Assumption 1). In the binary case, the idea of Faury et al. (2020) to circumvent this step is to introduce a refined "local" lower bound to the derivative of the logistic model (corresponding to our Jacobian above). In the binary case, such a lower bound always exists. However, this is not at all clear in the multinomial case because the Loewner order is only a partial order. This is yet another demonstration that extensions to the multinomial case are challenging on their own right and might require careful treatment. In what follows, we propose a new definition for the estimator $\boldsymbol{\theta}_t$ chosen from a newly defined set $\Theta \cap \mathcal{M}_t(\delta)$ (instead of $\Theta$ in (21)) formed based on the minimal elements of partially Loewner-ordered set $\mathcal{C}_t(\delta) \cap \Theta$. Please note that a minimal element of some partially ordered set $\mathcal{S}$ is defined as an element of $\mathcal{S}$ that is not greater than any other element in $\mathcal{S}$.

At each round $t \in [T]$, let $\left\{ \boldsymbol{\theta}'_{t,i} \in \mathcal{C}_t(\delta) \cap \Theta \right\}_{i \in [N_t]}$ be a set of $N_t$ vectors such that if for each $i \in [N_t]$, we define the sets

$$\mathcal{C}_{t,i}(\delta) := \{\boldsymbol{\theta} \in \mathcal{C}_t(\delta) \cap \Theta : \mathbf{A}(\mathbf{x}_t, \boldsymbol{\theta}) \succeq \mathbf{A}(\mathbf{x}_t, \boldsymbol{\theta}'_{t,i})\}, \tag{70}$$

then $\mathcal{C}_t(\delta) \cap \Theta = \cup_{i=1}^{N_t} \mathcal{C}_{t,i}(\delta)$. In fact, vectors $\boldsymbol{\theta}'_{i,t}$ are the minimal elements of the partially ordered set $\mathcal{C}_t(\delta) \cap \Theta$. Next, we define the set

$$\mathcal{M}_t(\delta) := \left\{ \boldsymbol{\theta} \in \mathbb{R}^{Kd} : \forall s \in [t-1], \exists i(s) \in [N_s] \text{ such that } \mathbf{A}(\mathbf{x}_s, \boldsymbol{\theta}) \succeq \mathbf{A}(\mathbf{x}_s, \boldsymbol{\theta}'_{s,i(s)}) \right\}. \tag{71}$$

Next, we define the following feasible set of estimators from which the new estimator $\boldsymbol{\theta}_t$ is chosen:

$$\mathcal{W}_t(\delta) := \mathcal{C}_t(\delta) \cap \mathcal{M}_t(\delta), \tag{72}$$

Now, we define the estimator $\boldsymbol{\theta}_t$ computed by the agent at round $t$:

$$\boldsymbol{\theta}_t := \underset{\boldsymbol{\theta} \in \Theta \cap \mathcal{M}_t(\delta)}{\arg\min} \left\| \mathbf{g}_t(\boldsymbol{\theta}) - \mathbf{g}_t(\hat{\boldsymbol{\theta}}_t) \right\|_{\mathbf{H}_t^{-1}(\boldsymbol{\theta})}. \tag{73}$$

Under the event $\mathcal{E}$ in (47), $\boldsymbol{\theta}_* \in \mathcal{C}_t(\delta) \cap \Theta$ for all $t > 0$. Thus, for all $t \in [T]$, there exists $i(t) \in [N_t]$ such that $\boldsymbol{\theta}_* \in \mathcal{C}_{t,i(t)}$. This combined with the definition of $\mathcal{M}_t(\delta)$ in (71) guarantees that $\boldsymbol{\theta}_* \in \mathcal{M}_t(\delta)$ and therefore $\boldsymbol{\theta}_* \in \Theta \cap \mathcal{M}_t(\delta)$. This shows that $\boldsymbol{\theta}_t \in \mathcal{W}_t(\delta)$ for all $t \in [T]$ since 1) $\boldsymbol{\theta}_t \in \mathcal{C}_t(\delta)$ as $\left\| \mathbf{g}_t(\boldsymbol{\theta}_t) - \mathbf{g}_t(\hat{\boldsymbol{\theta}}_t) \right\|_{\mathbf{H}_t^{-1}(\boldsymbol{\theta})} \leq \left\| \mathbf{g}_t(\boldsymbol{\theta}_*) - \mathbf{g}_t(\hat{\boldsymbol{\theta}}_t) \right\|_{\mathbf{H}_t^{-1}(\boldsymbol{\theta})} \leq \beta_t(\delta)$; and 2) $\boldsymbol{\theta}_t \in \Theta \cap \mathcal{M}_t(\delta)$.

In the next step towards completing the design of the algorithm, in Lemma 18, we define an improved exploration bonus $\epsilon_t(\mathbf{x}, \boldsymbol{\theta}_t)$ for any $\mathbf{x} \in \mathcal{D}$ and $\boldsymbol{\theta}_t$ defined in (73). First, we prove the following key Lemma 17, whose results will be used in the proof of Lemma 18.

**Lemma 17.** *For all $i \in [K]$, $\mathbf{x} \in \mathcal{D}$, $\boldsymbol{\theta}_1, \boldsymbol{\theta}_2 \in \mathcal{W}_t(\delta)$, and $t \in [T]$, with probability at least $1 - \delta$, it holds that*

$$z_i(\mathbf{x}, \boldsymbol{\theta}_1) - z_i(\mathbf{x}, \boldsymbol{\theta}_2) \leq \left[ \mathbf{x}^T(\bar{\boldsymbol{\theta}}_{11} - \bar{\boldsymbol{\theta}}_{21}), \dots, \mathbf{x}^T(\bar{\boldsymbol{\theta}}_{1K} - \bar{\boldsymbol{\theta}}_{2K}) \right] \mathbf{A}(\mathbf{x}, \boldsymbol{\theta}_2)_i + 4\kappa M \beta_t^2(\delta) \|\mathbf{x}\|_{\mathbf{V}_t^{-1}}^2, \tag{74}$$

*and therefore*

$$\mathbf{z}(\mathbf{x}, \boldsymbol{\theta}_1) - \mathbf{z}(\mathbf{x}, \boldsymbol{\theta}_2) \leq \left[ \mathbf{A}(\mathbf{x}, \boldsymbol{\theta}_2) \otimes \mathbf{x}^T \right] (\boldsymbol{\theta}_1 - \boldsymbol{\theta}_2) + 4\kappa M(1 + 2S)\beta_t^2(\delta)\|\mathbf{x}\|_{\mathbf{V}_t^{-1}}^2 \mathbf{1}, \tag{75}$$

.

*Proof.* According to Taylor's theorem, for a continuous twice differential function $f : \mathbb{R} \to \mathbb{R}$, we have

$$f(b) = f(a) + f'(a)(b-a) + \int_a^b f''(v)(b-v)dv. \tag{76}$$

Let $\mathbf{A}(\mathbf{x}, \boldsymbol{\theta})_i$ be the $i$-th column of $\mathbf{A}(\mathbf{x}, \boldsymbol{\theta})$, $a = 0$, $b = 1$, $f(v) = z_i\left(\mathbf{x}, \boldsymbol{\theta}_2 + v(\boldsymbol{\theta}_1 - \boldsymbol{\theta}_2)\right)$, and $\mathbf{M}_i(\mathbf{x}, \boldsymbol{\theta}_1, \boldsymbol{\theta}_2, v) := \nabla^2 z_i\left(\mathbf{x}, v\boldsymbol{\theta}_1 + (1 - v)\boldsymbol{\theta}_2\right)$ for any $v \in [0, 1]$, $i \in [K]$, $\mathbf{x} \in \mathcal{D}$, and $\boldsymbol{\theta}_1, \boldsymbol{\theta}_2 \in \Theta$. By the Taylor's theorem, for any $i \in [K]$, we have

$$
z_i\left(\mathbf{x}, \boldsymbol{\theta}_1\right) - z_i\left(\mathbf{x}, \boldsymbol{\theta}_2\right) = \left[\mathbf{x}^T(\bar{\boldsymbol{\theta}}_{11} - \bar{\boldsymbol{\theta}}_{21}), \ldots, \mathbf{x}^T(\bar{\boldsymbol{\theta}}_{1K} - \bar{\boldsymbol{\theta}}_{2K})\right] \mathbf{A}(\mathbf{x}, \boldsymbol{\theta}_2)_i
$$
$$
+ \int_{v=0}^{1} (1 - v) \left\| \begin{bmatrix} \mathbf{x}^T(\bar{\boldsymbol{\theta}}_{11} - \bar{\boldsymbol{\theta}}_{21}) \\ \vdots \\ \mathbf{x}^T(\bar{\boldsymbol{\theta}}_{1K} - \bar{\boldsymbol{\theta}}_{2K}) \end{bmatrix} \right\|^2_{\mathbf{M}_i(\mathbf{x}, \boldsymbol{\theta}_1, \boldsymbol{\theta}_2, v)} dv \qquad (77)
$$

Now, we bound the second term for any $\boldsymbol{\theta}_1, \boldsymbol{\theta}_2 \in \mathcal{W}_t(\delta)$ as follows:

$$
\int_{v=0}^{1} (1-v) \left\| \begin{bmatrix} \mathbf{x}^T(\bar{\boldsymbol{\theta}}_{11} - \bar{\boldsymbol{\theta}}_{21}) \\ \vdots \\ \mathbf{x}^T(\bar{\boldsymbol{\theta}}_{1K} - \bar{\boldsymbol{\theta}}_{2K}) \end{bmatrix} \right\|^2_{\mathbf{M}_i(\mathbf{x}, \boldsymbol{\theta}_1, \boldsymbol{\theta}_2, v)} dv \leq M \left\| \begin{bmatrix} \mathbf{x}^T(\bar{\boldsymbol{\theta}}_{11} - \bar{\boldsymbol{\theta}}_{21}) \\ \vdots \\ \mathbf{x}^T(\bar{\boldsymbol{\theta}}_{1K} - \bar{\boldsymbol{\theta}}_{2K}) \end{bmatrix} \right\|^2_{2} \qquad \text{(Assumption 3)}
$$

$$
= M \left\| \left[I_K \otimes \mathbf{x}^T\right] (\boldsymbol{\theta}_1 - \boldsymbol{\theta}_2) \right\|^2_{2}
$$

$$
= M \left\| \left[I_K \otimes \mathbf{x}^T\right] \mathbf{G}_t^{-1/2}(\boldsymbol{\theta}_1, \boldsymbol{\theta}_2) \mathbf{G}_t^{1/2}(\boldsymbol{\theta}_1, \boldsymbol{\theta}_2)(\boldsymbol{\theta}_1 - \boldsymbol{\theta}_2) \right\|^2_{2}
$$

$$
\leq M \left\| \left[I_K \otimes \mathbf{x}^T\right] \mathbf{G}_t^{-1/2}(\boldsymbol{\theta}_1, \boldsymbol{\theta}_2) \right\|^2_{2} \|\boldsymbol{\theta}_1 - \boldsymbol{\theta}_2\|^2_{\mathbf{G}_t(\boldsymbol{\theta}_1, \boldsymbol{\theta}_2)}
$$
$$
\text{(Cauchy-Schwarz inequality)}
$$

$$
= M \lambda_{\max}\left(\mathbf{G}_t^{-1}(\boldsymbol{\theta}_1, \boldsymbol{\theta}_2)\left[I_K \otimes \mathbf{x}\mathbf{x}^T\right]\right) \|\mathbf{g}_t(\boldsymbol{\theta}_1) - \mathbf{g}_t(\boldsymbol{\theta}_2)\|^2_{\mathbf{G}_t^{-1}(\boldsymbol{\theta}_1, \boldsymbol{\theta}_2)}
$$
$$
\text{(mixed-product property, cyclic property of } \lambda_{\max}, \text{ Lemma 11)}
$$

$$
\leq \kappa M(1 + 2S) \lambda_{\max}\left(\left[I_K \otimes \mathbf{V}_t^{-1}\right]\left[I_K \otimes \mathbf{x}\mathbf{x}^T\right]\right) \|\mathbf{g}_t(\boldsymbol{\theta}_1) - \mathbf{g}_t(\boldsymbol{\theta}_2)\|^2_{\mathbf{H}_t^{-1}(\boldsymbol{\theta}_1)}
$$
$$
\text{(Lemmas 12 and 13)}
$$

$$
= \kappa M(1 + 2S)\|\mathbf{x}\|^2_{\mathbf{V}_t^{-1}} \left\| \mathbf{g}_t(\boldsymbol{\theta}_1) - \mathbf{g}_t(\hat{\boldsymbol{\theta}}_t) + \mathbf{g}_t(\hat{\boldsymbol{\theta}}_t) - \mathbf{g}_t(\boldsymbol{\theta}_2) \right\|^2_{\mathbf{H}_t^{-1}(\boldsymbol{\theta}_1)}
$$
$$
\text{(cyclic property of } \lambda_{\max})
$$

$$
\leq 4\kappa M(1 + 2S)\beta_t^2(\delta)\|\mathbf{x}\|^2_{\mathbf{V}_t^{-1}}, \qquad \text{(Theorem 1)}
$$

as desired. $\qquad\qquad\qquad\qquad\qquad\qquad\qquad\qquad\qquad\qquad\qquad\qquad\qquad\qquad\qquad\square$

Now, we are ready to state the improved exploration bonus in the following Lemma

**Lemma 18** (Improved exploration bonus). *For all $\mathbf{x} \in \mathcal{D}$ and $t \in [T]$, with probability at least $1 - \delta$, we have*

$$
\Delta(\mathbf{x}, \boldsymbol{\theta}_t) \leq \bar{\epsilon}_t(\mathbf{x}, \boldsymbol{\theta}_t) := R(2 + 4S)\beta_t(\delta) \left\| \left[\mathbf{A}(\mathbf{x}, \boldsymbol{\theta}_t) \otimes \mathbf{x}^T\right] \mathbf{H}_t^{-1/2}(\boldsymbol{\theta}_t) \right\|_2 + 4\kappa M \left(\sum_{i=1}^{K} \rho_i\right)(1 + 2S)\beta_t^2(\delta)\|\mathbf{x}\|^2_{\mathbf{V}_t^{-1}},
$$
$$
(78)
$$

*where $\boldsymbol{\theta}_t$ is chosen as in (73).*

*Proof.* As a direct conclusion of Lemma 17, for all $\mathbf{x} \in \mathcal{D}$, $\boldsymbol{\theta} \in \mathcal{W}_t(\delta)$ and $t \in [T]$, with probability at least $1 - \delta$ it holds that

$$
\begin{aligned}
\Delta(\mathbf{x}, \boldsymbol{\theta}) &\leq R \left\| \left[ \mathbf{A}(\mathbf{x}, \boldsymbol{\theta}) \otimes \mathbf{x}^T \right] (\boldsymbol{\theta}_* - \boldsymbol{\theta}) \right\|_2 + 4\kappa M \left( \sum_{i=1}^K \rho_i \right) (1 + 2S)\beta_t^2(\delta) \|\mathbf{x}\|_{\mathbf{V}_t^{-1}}^2 \\
&= R \left\| \left[ \mathbf{A}(\mathbf{x}, \boldsymbol{\theta}) \otimes \mathbf{x}^T \right] \mathbf{G}_t^{-1/2}(\boldsymbol{\theta}_*, \boldsymbol{\theta}) \right\|_2 \left\| \mathbf{g}_t(\boldsymbol{\theta}_*) - \mathbf{g}_t(\boldsymbol{\theta}) \right\|_{\mathbf{G}_t^{-1}(\boldsymbol{\theta}_*, \boldsymbol{\theta})} + 4\kappa M \left( \sum_{i=1}^K \rho_i \right) (1 + 2S)\beta_t^2(\delta) \|\mathbf{x}\|_{\mathbf{V}_t^{-1}}^2 \\
&\leq 2R \left( \sqrt{1 + 2S} \right) \beta_t(\delta) \left\| \left[ \mathbf{A}(\mathbf{x}, \boldsymbol{\theta}) \otimes \mathbf{x}^T \right] \mathbf{G}_t^{-1/2}(\boldsymbol{\theta}_*, \boldsymbol{\theta}) \right\|_2 + 4\kappa M \left( \sum_{i=1}^K \rho_i \right) (1 + 2S)\beta_t^2(\delta) \|\mathbf{x}\|_{\mathbf{V}_t^{-1}}^2 \\
&\qquad\qquad\qquad\qquad\qquad\qquad\qquad\qquad\qquad \text{(Lemma 13 and Theorem 1)} \\
&\leq R \left( 2 + 4S \right) \beta_t(\delta) \left\| \left[ \mathbf{A}(\mathbf{x}, \boldsymbol{\theta}) \otimes \mathbf{x}^T \right] \mathbf{H}_t^{-1/2}(\boldsymbol{\theta}) \right\|_2 + 4\kappa M \left( \sum_{i=1}^K \rho_i \right) (1 + 2S)\beta_t^2(\delta) \|\mathbf{x}\|_{\mathbf{V}_t^{-1}}^2, \\
&\qquad\qquad\qquad\qquad\qquad\qquad\qquad\qquad\qquad\qquad\qquad \text{(Lemma 13)}
\end{aligned}
$$

which completes the proof of Lemma 18 as $\boldsymbol{\theta}_t \in \mathcal{W}_t(\delta)$ at all rounds $t \in [T]$. $\qquad\square$

**Theorem 3** (Regret of Improved MNL-UCB). *Fix $\delta \in (0, 1)$. Let Assumptions 1 and 2 hold. Then, with probability at least $1 - \delta$, it holds that*

$$
R_T \leq 12C_1 \sqrt{2 \max(1, \frac{1}{\lambda\kappa}) KTd \log \left( 1 + \frac{T}{d\lambda} \right)} + 4C_2 \max(1, \frac{1}{\lambda\kappa}) d \log \left( 1 + \frac{T}{\kappa\lambda d} \right), \quad (79)
$$

*where*

$$
C_1 = R \left( 2 + 4S \right) \beta_T(\delta) \quad \text{and} \quad C_2 = 2RM' \left( 2 + 4S \right) \kappa K \sqrt{(1 + 2S)} \beta_T^2(\delta) + 4\kappa M \left( \sum_{i=1}^K \rho_i \right) (1 + 2S)\beta_T^2(\delta). \tag{80}
$$

*Furthermore, if $\lambda = Kd \log(T)$, then*

$$
R_T = \mathcal{O} \left( K^{3/2} d \sqrt{T} \log(T) + \kappa \max(M, M') K^2 d \log(T) \right). \tag{81}
$$

*Proof.* Recall the definition of $\boldsymbol{\theta}'_{t,i}$ in the line above (70). A first order Taylor expansion gives that for all $\mathbf{x} \in \mathcal{D}$, $t > 0$, $h \in [N_t]$, and $i, j \in [K]$

$$
\mathbf{A}(\mathbf{x}, \boldsymbol{\theta}_t)_{ij} = \mathbf{A}(\mathbf{x}, \boldsymbol{\theta}'_{t,h})_{ij} + \left[ \mathbf{x}^T(\bar{\boldsymbol{\theta}}_{t1} - \bar{\boldsymbol{\theta}}'_{t,h1}), \dots, \mathbf{x}^T(\bar{\boldsymbol{\theta}}_{tK} - \bar{\boldsymbol{\theta}}'_{t,hK}) \right] \int_{v=0}^1 \nabla \mathbf{A}(\mathbf{x}, v\boldsymbol{\theta}_t + (1 - v)\boldsymbol{\theta}'_{t,h})_{ij} dv.
$$

For any $\mathbf{x} \in \mathbb{R}^d$ and $\boldsymbol{\theta}_1, \boldsymbol{\theta}_2 \in \mathbb{R}^{Kd}$, define $\mathbf{U}(\mathbf{x}, \boldsymbol{\theta}_1, \boldsymbol{\theta}_2) \in \mathbb{R}^{K \times K}$ with $\mathbf{U}(\mathbf{x}, \boldsymbol{\theta}_1, \boldsymbol{\theta}_2)_{ij} = \left[ \mathbf{x}^T(\bar{\boldsymbol{\theta}}_{11} - \bar{\boldsymbol{\theta}}_{21}), \dots, \mathbf{x}^T(\bar{\boldsymbol{\theta}}_{1K} - \bar{\boldsymbol{\theta}}_{2K}) \right] \int_{v=0}^1 \nabla \mathbf{A}(\mathbf{x}, v\boldsymbol{\theta}_1 + (1 - v)\boldsymbol{\theta}_2)_{ij} dv$. Thus, we have

$$
\mathbf{A}(\mathbf{x}, \boldsymbol{\theta}_t) = \mathbf{A}(\mathbf{x}, \boldsymbol{\theta}'_{t,h}) + \mathbf{U}(\mathbf{x}, \boldsymbol{\theta}_t, \boldsymbol{\theta}'_{t,h}). \tag{82}
$$

Now, we upper bound $\lambda_{\max}(\mathbf{U}(\mathbf{x}, \boldsymbol{\theta}_t, \boldsymbol{\theta}'_{t,h}))$ as follows

$$
\begin{aligned}
\lambda_{\max}(\mathbf{U}(\mathbf{x}, \boldsymbol{\theta}_t, \boldsymbol{\theta}'_{t,h}) &= \left\| \mathbf{U}(\mathbf{x}, \boldsymbol{\theta}_t, \boldsymbol{\theta}'_{t,h}) \right\|_2 \\
&\leq \left\| \mathbf{U}(\mathbf{x}, \boldsymbol{\theta}_t, \boldsymbol{\theta}'_{t,h}) \right\|_F \\
&= \sqrt{\sum_{i,j\in[K]} \mathbf{U}(\mathbf{x}, \boldsymbol{\theta}_t, \boldsymbol{\theta}'_{t,h})_{ij}^2} \\
&\leq \sqrt{\sum_{i,j\in[K]} \left\| [I_K \otimes \mathbf{x}]\,(\boldsymbol{\theta}_t - \boldsymbol{\theta}'_{t,h}) \right\|_2^2 \left\| \int_{v=0}^1 \nabla \mathbf{A}(\mathbf{x}, v\boldsymbol{\theta}_t + (1-v)\boldsymbol{\theta}'_{t,h})_{ij}\,dv \right\|_2^2} \\
&\qquad\qquad\qquad\qquad\qquad\qquad \text{(Cauchy-Schwarz inequality)} \\
&\leq M' K \left\| [I_K \otimes \mathbf{x}]\,(\boldsymbol{\theta}_t - \boldsymbol{\theta}'_{t,h}) \right\|_2 \qquad (\boldsymbol{\theta}_t, \boldsymbol{\theta}'_{t,h} \in \Theta \text{ and Assumption 3)} \\
&\leq M' K \left\| [I_K \otimes \mathbf{x}]\,\mathbf{G}_t^{-1/2}(\boldsymbol{\theta}_t, \boldsymbol{\theta}'_{t,h}) \right\|_2 \left\| \mathbf{g}_t(\boldsymbol{\theta}_t) - \mathbf{g}_t(\boldsymbol{\theta}'_{t,h}) \right\|_{\mathbf{G}_t^{-1}(\boldsymbol{\theta}_t, \boldsymbol{\theta}'_{t,h})} \\
&\qquad\qquad\qquad\qquad\qquad\qquad \text{(Lemma 11 and Cauchy-Schwarz inequality)} \\
&\leq 2M' K \sqrt{\kappa(1+2S)}\,\beta_t(\delta)\|\mathbf{x}\|_{\mathbf{V}_t^{-1}}. \qquad \text{(Lemma 12 and } \boldsymbol{\theta}_t, \boldsymbol{\theta}'_{t,h} \in \mathcal{C}_t(\delta))
\end{aligned}
$$

Combining this upper bound on $\lambda_{\max}(\mathbf{U}(\mathbf{x}, \boldsymbol{\theta}_t, \boldsymbol{\theta}'_{t,h}))$ and (82), we have

$$
\begin{aligned}
\left\| \left[ \mathbf{A}(\mathbf{x}, \boldsymbol{\theta}_t) \otimes \mathbf{x}^T \right] \mathbf{H}_t^{-1/2}(\boldsymbol{\theta}_t) \right\|_2 &= \left\| \left[ \left( \mathbf{A}(\mathbf{x}, \boldsymbol{\theta}'_{t,h}) + \mathbf{U}(\mathbf{x}, \boldsymbol{\theta}_t, \boldsymbol{\theta}'_{t,h}) \right) \otimes \mathbf{x}^T \right] \mathbf{H}_t^{-1/2}(\boldsymbol{\theta}_t) \right\|_2 \\
&\leq \left\| \left[ \mathbf{A}(\mathbf{x}, \boldsymbol{\theta}'_{t,h}) \otimes \mathbf{x}^T \right] \mathbf{H}_t^{-1/2}(\boldsymbol{\theta}_t) \right\|_2 + \left\| \left[ \mathbf{U}(\mathbf{x}, \boldsymbol{\theta}_t, \boldsymbol{\theta}'_{t,h}) \otimes \mathbf{x}^T \right] \mathbf{H}_t^{-1/2}(\boldsymbol{\theta}_t) \right\|_2 \\
&\qquad\qquad\qquad\qquad\qquad\qquad \text{(Triangle inequality)} \\
&\leq \left\| \left[ \mathbf{A}(\mathbf{x}, \boldsymbol{\theta}'_{t,h}) \otimes \mathbf{x}^T \right] \mathbf{H}_t^{-1/2}(\boldsymbol{\theta}_t) \right\|_2 + 2M' K \sqrt{\kappa(1+2S)}\,\beta_t(\delta)\|\mathbf{x}\|_{\mathbf{V}_t^{-1}} \left\| \left[ I_K \otimes \mathbf{x}^T \right] \mathbf{H}_t^{-1/2}(\boldsymbol{\theta}_t) \right\|_2 \\
&\leq \left\| \left[ \mathbf{A}(\mathbf{x}, \boldsymbol{\theta}'_{t,h}) \otimes \mathbf{x}^T \right] \mathbf{H}_t^{-1/2}(\boldsymbol{\theta}_t) \right\|_2 + 2M' \kappa K \sqrt{(1+2S)}\,\beta_t(\delta)\|\mathbf{x}\|_{\mathbf{V}_t^{-1}} \left\| \left[ I_K \otimes \mathbf{x}^T \right] \tilde{\mathbf{V}}_t^{-1/2} \right\|_2 \\
&\qquad\qquad\qquad\qquad\qquad\qquad (\mathbf{H}_t(\boldsymbol{\theta}_t) \succeq \kappa^{-1}\tilde{\mathbf{V}}_t) \\
&= \left\| \left[ \mathbf{A}(\mathbf{x}, \boldsymbol{\theta}'_{t,h}) \otimes \mathbf{x}^T \right] \mathbf{H}_t^{-1/2}(\boldsymbol{\theta}_t) \right\|_2 + 2M' \kappa K \sqrt{(1+2S)}\,\beta_t(\delta)\|\mathbf{x}\|_{\mathbf{V}_t^{-1}}^2.
\end{aligned}
\tag{83}
$$

Combining (78) and (83), for all $\mathbf{x} \in \mathcal{D}$, $t > 0$, and $h \in [N_t]$ we have

$$
\bar{\epsilon}_t(\mathbf{x}, \boldsymbol{\theta}_t) \leq C_1 \left\| \left[ \mathbf{A}(\mathbf{x}, \boldsymbol{\theta}'_{t,h}) \otimes \mathbf{x}^T \right] \mathbf{H}_t^{-1/2}(\boldsymbol{\theta}_t) \right\|_2 + C_2 \|\mathbf{x}\|_{\mathbf{V}_t^{-1}}^2,
\tag{84}
$$

where

$$
C_1 = R\,(2+4S)\,\beta_T(\delta) \quad \text{and} \quad C_2 = 2RM'\,(2+4S)\,\kappa K \sqrt{(1+2S)}\beta_T^2(\delta) + 4\kappa M \left( \sum_{i=1}^K \rho_i \right)(1+2S)\beta_T^2(\delta).
\tag{85}
$$

Recall the definition of $\mathcal{M}_t(\delta)$ in (35) and for $s \in [t-1]$, let $i(s) \in [N_s]$ be such that $\mathbf{A}(\mathbf{x}_s, \boldsymbol{\theta}_t) \succeq \mathbf{A}(\mathbf{x}_s, \boldsymbol{\theta}'_{s,i(s)})$. Thus, by the definition of $\mathbf{H}_t(\boldsymbol{\theta})$ in (9), we have

$$
\begin{aligned}
\mathbf{H}_t(\boldsymbol{\theta}_t) &= \lambda I_{Kd} + \sum_{s=1}^{t-1} \mathbf{A}(\mathbf{x}_s, \boldsymbol{\theta}_t) \otimes \mathbf{x}_s \mathbf{x}_s^T \\
&\succeq \lambda I_{Kd} + \sum_{s=1}^{t-1} \mathbf{A}(\mathbf{x}_s, \boldsymbol{\theta}'_{s,i(s)}) \otimes \mathbf{x}_s \mathbf{x}_s^T && \text{(Definition of } \boldsymbol{\theta}_t \text{ in (73))} \\
&\succeq \lambda I_{Kd} + \sum_{s=1}^{t-1} \mathbf{A}^2(\mathbf{x}_s, \boldsymbol{\theta}'_{s,i(s)}) \otimes \mathbf{x}_s \mathbf{x}_s^T && (\lambda_i(\mathbf{A}(\mathbf{x}_s, \boldsymbol{\theta}'_{s,i(s)})) \leq 1 \text{ for all } i \in [K]) \\
&= \lambda I_{Kd} + \sum_{s=1}^{t-1} \left[ \mathbf{A}(\mathbf{x}_s, \boldsymbol{\theta}'_{s,i(s)}) \otimes \mathbf{x}_s \right] \left[ \mathbf{A}(\mathbf{x}_s, \boldsymbol{\theta}'_{s,i(s)}) \otimes \mathbf{x}_s^T \right] && \text{(mixed-product property)} \\
&= \lambda I_{Kd} + \sum_{s=1}^{t-1} \sum_{i=1}^{K} \left[ \mathbf{A}(\mathbf{x}_s, \boldsymbol{\theta}'_{s,i(s)})_i \otimes \mathbf{x}_s \right] \left[ \mathbf{A}(\mathbf{x}_s, \boldsymbol{\theta}'_{s,i(s)})_i^T \otimes \mathbf{x}_s^T \right] \\
&= \lambda I_{Kd} + \sum_{s=1}^{t-1} \sum_{i=1}^{K} \tilde{\mathbf{x}}_{i,s} \tilde{\mathbf{x}}_{i,s}^T := \mathbf{L}_t,
\end{aligned}
\tag{86}
$$

where $\mathbf{A}(\mathbf{x}_s, \boldsymbol{\theta}'_{s,i(s)})_i$ is the $i$-th column of $\mathbf{A}(\mathbf{x}_s, \boldsymbol{\theta}'_{s,i(s)})$ and $\tilde{\mathbf{x}}_{i,s} := \mathbf{A}(\mathbf{x}_s, \boldsymbol{\theta}'_{s,i(s)})_i \otimes \mathbf{x}_s$.

Now, we bound the cumulative sum of the first term in (84) for $h = i(t)$, as follows:

$$
\begin{aligned}
\sum_{t=1}^{T} \left\| \left[ \mathbf{A}(\mathbf{x}_t, \boldsymbol{\theta}'_{t,i(t)}) \otimes \mathbf{x}_t^T \right] \mathbf{H}_t^{-1/2}(\boldsymbol{\theta}_t) \right\|_2 &= \sum_{t=1}^{T} \sqrt{ \sum_{i=1}^{K} \left\| \mathbf{A}(\mathbf{x}_t, \boldsymbol{\theta}'_{t,i(t)})_i \otimes \mathbf{x}_t \right\|_{\mathbf{H}_t^{-1}(\boldsymbol{\theta}_t)}^2 } \\
&\leq \sqrt{ T \sum_{t=1}^{T} \sum_{i=1}^{K} \left\| \tilde{\mathbf{x}}_{i,t} \right\|_{\mathbf{H}_t^{-1}(\boldsymbol{\theta}_t)}^2 } \\
& \qquad\qquad\qquad \text{(Cauchy-Schwarz inequality)} \\
&\leq \sqrt{ T \sum_{t=1}^{T} \sum_{i=1}^{K} \left\| \tilde{\mathbf{x}}_{i,t} \right\|_{\mathbf{L}_t^{-1}}^2 }.
\end{aligned}
\tag{87}
$$

For any positive semi-definite matrices $\mathbf{A}$, $\mathbf{B}$, and $\mathbf{C}$ such that $\mathbf{A} = \mathbf{B} + \mathbf{C}$, and for any $\mathbf{x} \neq 0$, we have ([Lemma 12] in Abbasi-Yadkori et al. (2011))

$$
\frac{\|\mathbf{x}\|_{\mathbf{A}}^2}{\|\mathbf{x}\|_{\mathbf{B}}^2} \leq \frac{\det(\mathbf{A})}{\det(\mathbf{B})} \quad \text{and} \quad \frac{\|\mathbf{x}\|_{\mathbf{B}^{-1}}^2}{\|\mathbf{x}\|_{\mathbf{A}^{-1}}^2} \leq \frac{\det(\mathbf{A})}{\det(\mathbf{B})}.
\tag{88}
$$

Thus

$$
\left\| \tilde{\mathbf{x}}_{i,t} \right\|_{\mathbf{L}_t^{-1}}^2 \leq \left\| \tilde{\mathbf{x}}_{i,t} \right\|_{\mathbf{L}_{t+1}^{-1}}^2 \frac{\det \mathbf{L}_{t+1}}{\det \mathbf{L}_t}
\tag{89}
$$

Note that for any $t \leq 1$, $\det \mathbf{L}_1 = \lambda^{Kd}$ and $\det \mathbf{L}_T \leq (\text{trace}(\mathbf{L}_T)/Kd)^{Kd} \leq \left( \lambda + \frac{T}{d} \right)^{Kd}$, and consequently:

$$
\frac{\det \mathbf{L}_T}{\det \mathbf{L}_1} = \prod_{t=1}^{T-1} \frac{\det \mathbf{L}_{t+1}}{\det \mathbf{L}_t} \leq \left( 1 + \frac{T}{d\lambda} \right)^{Kd}.
\tag{90}
$$

Since $1 \leq \frac{\det \mathbf{L}_{t+1}}{\det \mathbf{L}_t}$ for all $t > 0$, we have $e \leq \frac{\det \mathbf{L}_{t+1}}{\det \mathbf{L}_t}$ for at most $Kd \log \left( 1 + \frac{T}{d\lambda} \right)$ pairs of $(i,t) \in [K] \times [T]$, and therefore

$$
\left\| \tilde{\mathbf{x}}_{i,t} \right\|_{\mathbf{L}_t^{-1}}^2 \leq e \left\| \tilde{\mathbf{x}}_{i,t} \right\|_{\mathbf{L}_{t+1}^{-1}}^2
\tag{91}
$$

is true for all pairs of $(i, t) \in [K] \times [T]$ but at most $\phi(K, d, \lambda, T) := Kd \log \left(1 + \frac{T}{d\lambda}\right)$ of them. We define the set of good pairs by $\mathcal{B}_{\text{good}} := \left\{ (i, t) \in [K] \times [T], \left\|\tilde{\mathbf{x}}_{i,t}\right\|^2_{\mathbf{L}_t^{-1}} \leq e\left\|\tilde{\mathbf{x}}_{i,t}\right\|^2_{\mathbf{L}_{t+1}^{-1}} \right\}$. We have

$$
\begin{aligned}
\sum_{t=1}^{T}\sum_{i=1}^{K}\left\|\tilde{\mathbf{x}}_{i,t}\right\|^2_{\mathbf{L}_t^{-1}} &= \sum_{(i,t) \notin \mathcal{B}_{\text{good}}} \left\|\tilde{\mathbf{x}}_{i,t}\right\|^2_{\mathbf{L}_t^{-1}} + \sum_{(i,t) \in \mathcal{B}_{\text{good}}} \left\|\tilde{\mathbf{x}}_{i,t}\right\|^2_{\mathbf{L}_t^{-1}} \\
&\leq \phi(K, d, \lambda, T)/\lambda + e\sum_{t=1}^{T}\sum_{i=1}^{K}\left\|\tilde{\mathbf{x}}_{i,t}\right\|^2_{\mathbf{L}_{t+1}^{-1}} \\
&\leq \phi(K, d, \lambda, T)/\lambda + 2e\phi(K, d, \lambda, T) \qquad (92)
\end{aligned}
$$

Now, combining (78), (87) and (92), we bound $R_T$ in the following:

$$
\begin{aligned}
R_T &\leq 2\sum_{t=1}^{T}\bar{\boldsymbol{\epsilon}}_t(\mathbf{x}_t, \boldsymbol{\theta}_t) \\
&\leq 2C_1\sqrt{T\sum_{t=1}^{T}\sum_{i=1}^{K}\left\|\tilde{\mathbf{x}}_{i,t}\right\|^2_{\mathbf{L}_t^{-1}}} + 2C_2\sum_{t=1}^{T}\left\|\mathbf{x}_t\right\|^2_{\mathbf{V}_t^{-1}} \\
&\leq 12C_1\sqrt{2\max(1, \frac{1}{\lambda\kappa})dKT \log\left(1 + \frac{T}{d\lambda}\right)} + 4C_2\max(1, \frac{1}{\lambda\kappa})d\log\left(1 + \frac{T}{\kappa\lambda d}\right),
\end{aligned}
$$

as desired. $\qquad\square$

## E    Additional Experiments

**The "local dependence" of the confidence set $\mathcal{C}_t(\delta)$.** We highlight the superiority of using $\epsilon_t(\mathbf{x})$ vs $\tilde{\epsilon}_t(\mathbf{x})$ introduced in (22) and (24), respectively. In Figure 2, we give an illustration of how $\mathcal{C}_t(\delta)$ compares to $\mathcal{E}_t(\delta)$ from which $\epsilon_t(\mathbf{x})$ and $\tilde{\epsilon}_t(\mathbf{x})$ are derived for different values of $\kappa$. In this figure, instead of $\mathcal{C}_t(\delta)$, we considered a slightly *larger* confidence set $\tilde{\mathcal{C}}_t(\delta)$ in the more familiar (from linear bandits) form of

$$
\tilde{\mathcal{C}}_t(\delta) := \left\{\boldsymbol{\theta} \in \Theta : \left\|\boldsymbol{\theta} - \hat{\boldsymbol{\theta}}_t\right\|_{\mathbf{H}_t^{-1}(\boldsymbol{\theta})} \leq (2 + 4S)\beta_t(\delta)\right\},
$$

where $\boldsymbol{\theta}$ is a "good" estimate of $\boldsymbol{\theta}_*$ based on the weighted norm of $\boldsymbol{\theta} - \hat{\boldsymbol{\theta}}_t$ rather than $\mathbf{g}_t(\boldsymbol{\theta}) - \mathbf{g}_t(\hat{\boldsymbol{\theta}}_t)$ in (12). Technical details proving $\mathcal{C}_t(\delta) \subseteq \tilde{\mathcal{C}}_t(\delta)$ at all rounds $t \in [T]$ are deferred to Appendix C. For the sake of visualization, we find it instructive to depict the confidence sets in a 1-dimensional space for a realization with $K = 2$. The curves of $\tilde{\mathcal{C}}_t(\delta)$ effectively capture the local dependence of $\mathbf{H}_t(\boldsymbol{\theta})$ on $\boldsymbol{\theta}$. As a result, unlike $\mathcal{E}_t(\delta)$, the confidence set $\tilde{\mathcal{C}}_t(\delta)$ is not an ellipsoid and estimators in different directions are penalized in different ways. Furthermore, the appearance of $\kappa$ in the radius of the confidence ellipsoid $\mathcal{E}_t(\delta)$ (see (23)), which is caused by the use of the matrix $\tilde{\mathbf{V}}_t$ as a global bound on $\mathbf{G}_t$ (see Appendix C for details), results in a larger confidence set compared to $\tilde{\mathcal{C}}_t(\delta)$ that employs the local metric $\mathbf{H}_t(\boldsymbol{\theta})$. This is also consistent with the observation that as $\kappa$ grows, the difference between the diameters of $\tilde{\mathcal{C}}_t(\delta)$ and $\mathcal{E}_t(\delta)$ increases.

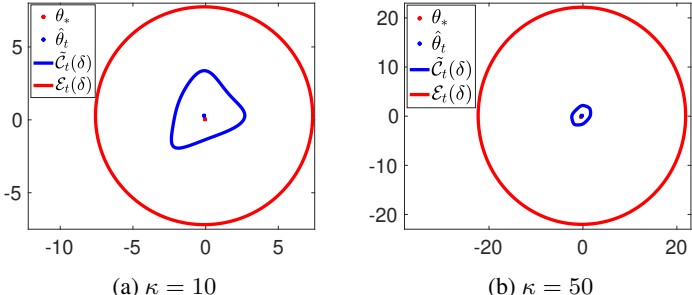

Figure 2: Illustration of $\tilde{\mathcal{C}}_t(\delta)$ and $\mathcal{E}_t(\delta)$ at round $t = 500$ for different values of $\kappa$. Note the "local dependence" of $\tilde{\mathcal{C}}_t(\delta)$ which results in a non-ellipsoidal shape.