# OpenReview forum: "UCB-based Algorithms for Multinomial Logistic Regression Bandits"
_NeurIPS.cc/2021/Conference — NeurIPS 2021 Poster_

### Official Review · Reviewer_SPJz · 2021-07-12

**Rating:** 6
**Confidence:** 4

**Summary:**

This paper considers a version of stochastic contextual bandits where the rewards are drawn from a discrete distribution governed by a multionimal logit model. This setting is motivated by applications in online advertising where the user may give various feedbacks in the face of a recommendation beyond a binary ‘clicked’ or ‘did not click’ – e.g. ‘show me again later’, ‘never show me again’. The authors propose and analyse a UCB algorithm for this problem with the confidence sets being based on the work of Faury et al. (2020, ICML) and the regret analysis being based on that of Abbasi-Yadkori et al. (2011, NeurIPS). The particular advantage of utilising the concentration results of Faury et al is that the regret of the resulting algorithm scales favourably (square root order) with respect to a parameter $\kappa$ characterising the smoothness of the logit function around the true parameter. A further algorithm whose dependence on $\kappa$ is relegated to lower order terms w.r.t. the horizon is also developed – however this approach is computationally intractable. In the related (binary) logistic bandit setting, some popular algorithms have been shown to have exponential order dependence on this parameter and the design of algorithms which are robust to its value is useful. It may be useful to note that this version of a multinomial bandit is different from the combinatorial problem in the dynamic assortment selection literature (e.g. Agrawal et al (2017, 2019)) where the different outcomes represent different items.




**Limitations And Societal Impact:**

yes - this has been considered by the authors.

**Main Review:**

I feel that this is a somewhat borderline paper. The extension of the logistic bandit to the multinomial logit setting is one that would be of genuine interest, and the analyses and proposed algorithm seem to be sound. The issue is that the developments are somewhat marginal. Some detailed theoretical work is required get the correct form of the UCB terms to hold over multiple parameter vectors simultaneously, but it seems that a good deal of the effort lies in combining new concentration theory (in this instance from Faury et al. (2020))  with the regret analyses of Abbasi-Yadkori et al. (2011). This is a credible and often used approach within this literature, but its familiarity does bring with it the question of whether there is sufficient novelty to merit publication at a top venue such as NeurIPS, and not an alternative venue. I have no doubt that there are many venues where this solid piece of research would easily clear the acceptance bar.

ORIGINALITY: The headline results are new in and of themselves, in so far as the literature has not combined the novel concentration results and regret analyses in the context of this variant of an MNL-bandit. As discussed above, a large component of the analysis is focussed on the combining of existing tools – a perfectly useful piece of work but one that perhaps scores lower in the originality aspect.

QUALITY: As far as I can tell, the theory is accurate and the paper is generally written well. The algorithm is not compared to any (even naïve) competitors so it is hard to gauge its efficacy. We can however see that the variation in its performance with respect to particular problem parameters mirrors the implications of the theory. Where I think the paper could be improved through relatively little effort is through making yet more explicit the connections to the preceding work. The following questions occurred to me while reading the manuscript but did not have answers which I could easily discern. I feel the discussion and my understanding would be improved further by addressing them.
1.	Faury et al. (2020) can, according to line 119, drop the dependence on $\kappa$ entirely. Why is this not possible (or perhaps not pursued?) here?
2.	Line 221 asserts the main technical contributions of the paper. Are these ‘main’ in the sense that they are the most challenging, the largest departure from existing techniques, or the most likely to be useful elsewhere? In any of these cases, why not sell this contribution more prominently among your contributions?
3.	In what sense, precisely is the ‘improved algorithm’ computationally intractable? I did not find an explicit discussion of this in the requisite appendix.

CLARITY: I feel the paper is generally clear and well written. As a technical paper with a lot of theoretical angles to introduce in a short page limit it is occasionally more quite concise, but I think it is sufficiently clear for the intended audience.

SIGNIFICANCE: As discussed above, this seems like a pragmatic and realistic extension of logistic bandits to a setting that is likely to arise in online advertising. The analysis is solid and accurate as far as I can tell. This is a nice piece of work, suitably complete for publication (somewhere) and will be of interest to those working in this area.

**Time Spent Reviewing:**

5

---

> ### Author Response · Authors · 2021-08-10
> **Response to Reviewer SPJz**
>
> We thank the reviewer for their valuable time and detailed feedback. We are encouraged by the reviewer’s comments.
>
> Regarding novelty, please see our response to Reviewer yP7L.
>
> Now, we address the reviewer’s questions one by one.
>
> **1.** In Line 119, we refer to Logistic-UCB-2 in Faury et al. (2020) that achieves a regret of $\mathcal{O}(d\sqrt{T}\log(T)+\kappa d^2 \log^2(T))$ or $\tilde{\mathcal{O}}(d\sqrt{T})$, where the problem-dependent constant $\kappa$ is pushed into a second order term that vanishes quickly. Logistic-UCB-2 employs careful algorithmic designs that improve upon Logistic-UCB-1 with a regret of ${\mathcal{O}}(d\sqrt{\kappa T}\log(T))$. That said, unlike Logistic-UCB-1, Logistic-UCB-2 is not computationally tractable. In our paper, we also propose Improved MNL-UCB that improves the regret of MNL-UCB in terms of $\kappa$ from $\mathcal{O}(Kd\sqrt{\kappa T}\log(T))$ to $\mathcal{O}(K^{1.5}d\sqrt{\kappa T}\log(T)+\kappa K^2 d\log(T))$ (see Appendix D and Theorem 3 for details). We discuss this in Lines 104-105, in Section 2.6, and in Appendix D.
>
>
> **2.** With “our key technical contribution” in Line 221, we refer to *both* aspects mentioned by the reviewer. First, bounding these two terms is the most involved and challenging part of our analysis. In establishing these two bounds, it was crucial to carefully combine unique properties of the MNL model (such as the generalized self-concordance property of log-sum-exp function)  with  Due to the constant need for using the Kronecher product and the more complicated structure of the Hessian matrix, MLE and the confidence set introduced in Theorem 1, whose proof involves certain non-trivialities on and of itself, none of the parts in our analysis was a clear extension of its counterpart in binary logistic setting. As a result, these contributions can be significantly useful in the design and improvement of algorithms that involve MNL models. We thank the reviewer for their nice suggestion.  We will include this high-level discussion related to the challenge of bounding the spectral norms of those kronecker products involving the Hessian matrix earlier in the “contributions” section, as suggested by the reviewer.
>
> **3** Similar to Logistic-UCB-2 in Faury et al. (2020), Improved MNL-UCB is computationally intractable as it requires the computation of $\boldsymbol\theta_t$ in Equation (73).This step itself involves a non-convex optimization problem over the set $\Theta\cap\mathcal{M}_t(\delta)$, where it is not clear at all how the construction of the set $\mathcal{M}_t(\delta)$ must be done in an efficient way.

---

> > ### Comment · Reviewer_SPJz · 2021-09-02
> > **Keeping my score of 6**
> >
> > Hi authors,
> >
> > Thank you for your detailed response which clarifies my queries. For you info, I will keep my score at 6 as I still have a positive view of the paper following the review and discussion process.

---

> > > ### Author Response · Authors · 2021-09-03
> > > **Thank you**
> > >
> > > Dear Reviewer,
> > >
> > > Thank you again for your time and for confirming your positive evaluation of our paper.
> > >
> > > Best,
> > > Authors

---

### Official Review · Reviewer_KVKE · 2021-07-12

**Rating:** 6
**Confidence:** 4

**Summary:**

The paper studies multinomial logistic regression bandit problem, where the number of outcomes can be greater than two. This is an extension of the well-studied generalized linear bandit problem with two outcomes (e.g., clicks and no-clicks). The authors first propose the problem setup with real-life motivated scenarios and then propose a UCB based algorithm to minimize cumulative regret. The authors analyze its performance and derive an upper bound of \tilde{O}(dK\sqrt{\kappa T}), where d is the dimension of the action space, (K + 1) is the number of outcomes, T is the number of interactions, and \kappa is the degree of non-smoothness, which is inherent to the multi-nomial problem. At the end, the authors provide a simulation based study which supports their theoretical insights.

**Limitations And Societal Impact:**

No, the authors have not mentioned limitations and societal impacts of their work.

**Main Review:**

Strengths:
1. Relevant and practical extension of an important problem.
2. The theoretical analysis and presentation is unique (some things can be clarified; see detailed comments)

Weaknesses:
1. The paper does not contain experiments with real-datasets and no comparisons with baselines.
2. The are a few concerning questions that confuse the reader; hence presentation can be improved (see detailed comments).

Detailed Feedback:

1. I am a bit confused with the bound that the authors have mentioned in the abstract and in the main text. They seem to be inconsistent. Can you please clarify? Also, when mentioning the bounds, please mention what the parameter stands for, e.g., d, T?

2. Equation 2: I believe it should be \forall i \in [K] instead of \forall i \in [K] \cup {0}.

3. There seems to be some mistake in the way expected reward is defined in line 82. There should be some argument in \mu that should depend on the vector \rho, but I do not see it. Then, the domain of \mu in line 83 should also be defined properly.

4. Why is it necessary to have all the \rho_i >=0? I see an option like "never show me this ad" generating a positive reward to be contradictory? More so, when compared to not-clicking that generates a reward of 0.

5. Equation 6: What is x_* in the definition of instantaneous regret?

6. The analysis largely depends on the regularization parameter $\lambda$. I wonder how the analysis will change if in equation 7, one chooses to work without regularization.
Also, there is notation abuse. \lambda_max/\lambda_min are used for eigen values and \lambda is used for regularization as well.

7. Right at the point where the authors make assumption 2, it is not clear to me why A will have positive eigen values. Can you please clarify?

8. In Lemma 1, I was confused about what V_t stands for. The authors explain its definition much later. It would be better to reorganize that bit.

9. The presentation of analysis is good. However, I would recommend moving the Lemmas which are used to prove other major results to the appendix, and instead add experiments with real-data in the paper. My major reason for giving the current score is because of its weak experiments.

10. The authors provide intuitive reasons of why there should be a linear dependence in K. However, they have not provided any proof in that regard. Thus, the usage of the word 'optimal' is imprecise in my opinion.

11. There has been a blatant use of vspace/vskip in the entire paper. If the paper is accepted, I would recommend avoiding their large usage. It just destroys the aesthetics of the paper.

Minor:
1. Line 30: statistic --> statistics?

**Time Spent Reviewing:**

3

---

> ### Author Response · Authors · 2021-08-10
> **Response to Reviewer KVKE**
>
> We thank the reviewer for their valuable time and detailed feedback. We are encouraged by the reviewer’s comments: “Relevant and a practical extension of an important problem”, and the theoretical analysis and presentation unique.” Below, we address the reviewer’s comments one by one.
>
> **1.** We would like to clarify that the regret bound proven for MNL-UCB is of order $\mathcal{O}(Kd\sqrt{\kappa T})$. This has only a small dependency of $\mathcal{O}(\sqrt{\kappa})$ on the on problem-dependent constant $\kappa$ compared to the regret $\mathcal{O}(Kd\kappa\sqrt{ T})$ achieved by adapting other existing techniques like those used in Filippi et al. (2010). Moreover, we propose an improved algorithm (namely,Improved MNL-UCB in Appendix D) that achieves a regret bound with problem-dependent constant $\kappa$ being pushed into a second order logarithmic term that vanishes quickly.
>
> **2.** Thank you for pointing this out. We will fix it in the revision.
>
> **3.** Here, $\mu:\mathbb{R}^K\rightarrow \mathbb{R}$ is defined as follows: for any $\mathbf{y}\in\mathbb{R}^K$, we have $\mu(\mathbf{y})=\frac{\sum_{j=1}^K \rho_j \exp(\mathbf{y}_j)}{1+\exp(\mathbf{y}_j)}$. Now, if we let $\mathbf{y}_j = {\bar{\boldsymbol{\theta}}}_{\ast j}^T\mathbf{x}$, then $\mathbb{E}[R_t|\mathbf{x}_t,\mathcal{F}_t]=\mu(\mathbf{y})$. In fact, in Line 82, we wanted to distinguish the multi-index model in our paper from a generalized linear model, which binary logistic model is a special case of.
>
> **4.** Thank you for pointing this out. In our analysis we do not rely on $\rho_i$ being non-negative for all $i\in[K]$. All that matters is sorting $\rho_i$ with respect to their corresponding outcome in a reasonable way. In the example mentioned by the reviewer, the revenue of “never show me this ad” is smaller than the revenue of “not clicking”. We will clarify this in the revision.
>
> **5.** $\mathbf{x}_\ast$ in Equation 6 is defined in Line 85 as $\mathbf{x}_\ast \in {\rm argmax}{\mathbf{x}\in\mathcal{D}} \boldsymbol{\rho}^T\mathbf{z}(\mathbf{x},\boldsymbol \theta_\ast)$, the optimal action with maximum expected reward, when we wanted to introduce the cumulative pseudo-regret.
>
>
> **6.** The regularized estimator has been widely used in the literature of linear and generalized linear bandits, e.g., Abbasi-Yadkori et al. (2011), Dani et al. (2008), Filippi et al. (2010),  Faury et al. (2020). Regularization is useful to ensure that the Hessian matrix $\mathbf{H}_t(\boldsymbol\theta)$ and the Gram matrix $\mathbf{V}_t$ used in the implementation of our algorithms are invertible. An alternative approach to make sure that $\mathbf{H}_t(\boldsymbol\theta)$ is invertible without using regularization is selecting actions randomly for a small number of rounds in the beginning of the algorithm, and only then switching to the main phase of the algorithm that uses the UCB decision rule. Moreover, $\lambda$ is the standard notation for the regularization parameter in the literature (e.g., see all the above-stated papers), and is different from $\lambda{\rm min}(\mathbf{A})$ and $\lambda{\rm max}(\mathbf{A})$, which are the minimum and maximum eigenvalues of a matrix $\mathbf{A}$.
>
>
>
> **7.** We explain why $\mathbf{A}(\mathbf{x},\boldsymbol\theta)$ is  positive definite in Line 602-607 in the appendix. In particular, thanks to the MNL probabilistic model, one can easily observe that for any $\mathbf{x}\in\mathbb{R}^d$ and $\boldsymbol\theta\in\mathbb{R}^{Kd}$, the matrix $\mathbf{A}(\mathbf{x},\boldsymbol\theta)$ is strictly diagonally dominant. Now, a strictly diagonally dominant matrix that is also symmetric with positive diagonal entries, as is  $\mathbf{A}(\mathbf{x},\boldsymbol\theta)$, is positive definite (e.g. Theorem 6.1.10 in Hornet and Johnson ,2012). Thus, $\mathbf{A}(\mathbf{x},\boldsymbol\theta)$ is a positive definite matrix with positive eigenvalues. For completeness, we recall the definition of strictly diagonally dominant matrices in Lines 321-322: they are  matrices for which each of their diagonal entries is greater than the sum of absolute values of all other entries in the corresponding row/column.
>
> **8.** Thank you for flagging this, we will address it in the revision.
>
> **9.** The main contribution of our paper is the proof of the regret bound of our proposed algorithm. Since new techniques are being utilized compared to prior work, we have decided to include most of them in the main body.That said, we thank the reviewer for their suggestion, and we will use the additional 1-page given to us (if the paper is accepted) to add more experiments.
>
> **10.** Thank you for raising this. We will address it in the revision.

---

### Official Review · Reviewer_Shft · 2021-07-16

**Rating:** 5
**Confidence:** 5

**Summary:**

This paper presents MNL-UCB Algorithm for contextual MNL bandits problem. At every step, the algorithm estimates a K-tuple of parameter vectors in the confidence region. Specifically, they prove that the regret of MNL-UCB scales as \tilde{O}(dK\sqrt{\kappa}\sqrt{T}) where $\kappa$ is a parameter that captures the degree of (non)-smoothness of the MNL model. It improves the constant from linear in \kappa to \sqrt{\kappa}. They further improved algorithm that achieves a regret bound with problem-dependent constant \kappa being pushed into a second order term.

Compared to the most existing work, a major difference is that each choice is associated with a different parameter so a matrix of parameters needs to be learned.


**Main Review:**

Contextual MNL is a recent hotly studied topic.

My biggest concern is the lack of comparison with an existing work. I want to bring one paper into attention: A Tractable Online Learning Algorithm for the Multinomial Logit Contextual Bandit. The main difference between two papers is whether different choices share the same parameters. However, the analyses are very similar in two settings and the analysis is similar to the generalized linear bandits. Moreover, regarding the regret bound, the improvement is also similar: it improves the constant term from \kappa to \sqrt{\kappa} and it meanwhile pushes the constant term into a second order term. Can you clarify

1) The technical difficulty/contribution of this paper (e.g. how it differs from the existing work);

2) What's the connection between two constants (kappa in your paper and kappa in their paper)?

3) What is the regret lower bound? Is the regret optimal with respect to the order of K? I saw the short comments in line 275-285, but can you provide a rigorous argument for that conclusion?

The writting is good, but the contribution needs to be clearly emphasized.

Minor: 1) The notation \bar{\theta}_* can be simplified to \theta_*;

2) Please define \otimes before formula (8);




**Time Spent Reviewing:**

3

---

> ### Author Response · Authors · 2021-08-10
> **Response to Reviewer Shft**
>
> We thank the reviewer for their valuable time and detailed feedback. We make the following clarifications below.
>
> Regarding the comparison  to Agrawal et al. (2020), as pointed out by both reviewers Shft and yP7L, this work studies the different problem of dynamic assortment selection, where at each round, the agent must select a set of at most $K$ items each of which is selected by the user with a probability that follows a MNL model. As such, the parameter of interest to be learned is a d-dimensional vector, while we need to learn a concatenated vector of dimension $d\times K$. Thus, the main first technical difference is the way confidence sets are constructed around unknown parameters. All in all, we believe that the technical analysis of Agrawal et al. (2020) is a more direct extension of Faury et al. (2020), as they simply treat the MNL model as $K$ different binary logistic models. This is *not* the case in our setting. Having said that, a comparison with their technical contributions is definitely worth mentioning and we will do so in the revision.
>
> Now, we respond to the remaining of the reviewer’s questions one by one:
>
> **1.** Please refer to our response to reviewer yP7L regarding novelty.
>
> **2.** The definition of $\kappa$ in Assumption 2 of Agrawal et al. (2020) is different from our definition of $\kappa$. Unlike ours, their definition is fairly similar to that of Faury et al. (2020), except that they rely on the local information about multiple (i.e., $K$) different binary logistic models. Thus, they define $\kappa$ rather naturally by essentially taking an infimum of $\kappa$’s corresponding to multiple binary logistic models. In contrast, in order to define $\kappa$ in our setting, we rely on the *global* information of the MNL model, comprised of $K$ $d$-dimensional vectors, which is characterized by the minimum eigenvalue of the Hessian matrix $\mathbf{A}(\mathbf{x},\boldsymbol\theta)$. In particular, we used the “strictly diagonally dominant’’ property of $\mathbf{A}(\mathbf{x},\boldsymbol\theta)$ to estimate $\kappa$ by an appropriate lower bound in Equation (26). We needed this new argument, in order to show that our $\kappa$ has exponential dependence on the size of the set $\Theta\times\mathcal{D}$. This new calculation was also necessary to set the value of $\kappa$ in our experiments.
>
> **3.** We believe our regret bound in terms of $K$ is tight. To see this note that the MNL reward model indeed involves $Kd$ unknown parameters and recall that  a binary logistic bandit with $Kd$ parameters is well-known to have a tight *linear* dependence on $K$ (Abbasi-Yadkori et al. 2011). Having said that, we reiterate that since the MNL is a special case of multi-index models rather than a GLM, it is a-priori unclear whether it can achieve the regret of a binary logistic model with $Kd$ parameters. In fact, our proof does not treat the MNL reward model as a GLM and our proof does not treat it as such. Despite that, it does achieve a linear order $\mathcal{O}(K)$ same as the optimal order of a binary logistic setting with $Kd$ unknown parameters. That said, it remains an open question to determine a lower bound that formally justifies the optimal dependence is $\mathcal{O}(K)$ or determine an improved algorithm/analysis that lowers the dependency. Thank you for your question; it makes for a great discussion, which will be added in the revision and we hope that it motivates further investigation.

---

> > ### Comment · Reviewer_Shft · 2021-08-20
> > **review round 2**
> >
> > Thanks for your clarification. I have some more detailed questions:
> >
> > 1. Please define $\delta_{ij}$ in line 163.
> > 2. In line 502: $\|x_t\|_2$ should be changed to $\|x_t\|_2^2$.
> > 3. Lemma 7: For all $\|\xi\|\in \frac{1}{\sqrt{K}}B_2(Kd)$; it is clearer to state:  $\|\xi\|\in B_2(K^2 d)$.
> > 4. In line 513, please define the normalization constant N.
> > 5. What is [N_t] in line 641? A more formal way to define the set would be $\{\theta'_{t,i}:\theta'_{t,i}\in C_t(\delta)\}$. You may want to change the notation to differentiate from item 4.
> > 6. What is the definition of $\theta'$ in line 677? Is it from the definition in line 641 (for any element in that set)?

---

> > > ### Author Response · Authors · 2021-08-21
> > > **Response round 2**
> > >
> > >
> > > Thank you for your questions. Please find our responses below. We appreciate your careful reading of the paper and are available to answer any other questions that you might have.
> > >
> > > **1.**  $\delta_{ij}$ is the Kronecker delta, i.e. $\delta_{ij}=1$ if $i\neq j$ and $\delta_{ij}=0$ if $i=j$. We will formally define this in the revision. Good catch.
> > >
> > >
> > > **2.** In line 502, we have $||\mathbf{x}_t||_2$ because in the third and fourth lines, we used the fact that
> > > $\mathbf{x}_t^T(\sum_i{\bar\xi}_i{\bar\xi}_i^T) \mathbf{x}_t \leq ||\mathbf{x}_t||_2^2 \cdot\lambda_1$, where $\lambda_1$ denotes the maximum eigenvalue of the matrix $\sum_i \bar\xi_i\bar\xi_i^T$. The LHS is in the square-root, so this gives rise to $||\mathbf{x}_t||_2$ as written.
> > >
> > > **3.** According to the definition in line 497, $\mathcal{B}_2(Kd)$ is the unit ball in $Kd$-dimensional space. Therefore, $\frac{1}{\sqrt{K}}\mathcal{B}_2(Kd)$ is a ball with radius $\frac{1}{\sqrt{K}}$ in the $Kd$-dimensional space. In particular, this is different from $\mathcal{B}_2(K^2d)$, a unit ball in $K^2d$-dimensional space in our notation.
> > >
> > > **4.** As stated in line 513, $N(h)$ and $N(g)$ are the normalization constants of $h$ and $g$. The explicit definitions are as follows:
> > > $$N(h):=\int_{S}{\text{exp}}(-\lambda||\xi||^2_2), \quad\text{where}~ S:=\\{ \xi\in\mathbb{R}^{Kd} : ||\xi||_2\leq  \sqrt\frac{1}{K} \\}$$
> > >
> > > $$N(g)=\int_{S}{\text{exp}}(-\frac{1}{2}||\xi||^2_{2\mathbf{H}_t({\boldsymbol{\theta}}_\ast)}), \quad\text{where}~ S:=\\{ \xi\in\mathbb{R}^{Kd} : ||\xi||_2\leq  \frac{1}{2}\sqrt\frac{1}{K} \\}$$. We will add these explicit definitions in the revision for clarity.
> > >
> > >
> > > **5.** The notation $[n]$ for integer $n$ is introduced in line 65. Specifically, $[N_t]$ in line 641 denotes the set $\\{1,2,\ldots,N_t\\}$. Thank you for the suggestion regarding defining the set of $\theta^\prime_{t,i}$ in line 641. We will adopt it in the revision. We will also change the notations for normalization constants $N(h)$ and $N(g)$ to $Z(h)$ and $Z(g)$ to differentiate them from $N_t$.
> > >
> > > **6.** You are correct: the definition of ${\boldsymbol{\theta}}^\prime_{t,h}$ comes from line 677. We will add a note in the revision to remind the reader of the definition in Lines 641-643.

---

### Official Review · Reviewer_yP7L · 2021-07-25

**Rating:** 5
**Confidence:** 5

**Summary:**

The paper studies multinomial logistic (MNL) regression bandits, a generalization of binary logistic bandits. It differs from previous works on MNL bandits by considering a different decision-making process. That is, the authors do not address the combinatorial action selection of previous MNL bandits, but rather study the problem setting where the agent offers a single item, but with K different outcomes.

The paper introduces a UCB-based algorithm for this problem, for which the authors establish a sub-linear regret bound. By building on recent advances in logistic bandits, they improve the prohibitive impact of the constant \kappa from both the design of their algorithm and its theoretical guarantees.

**Main Review:**

The paper is well-written and easy to understand. The problem setting is stylistic but well-motivated. I am convinced that this variant of the MNL problem is worth attention from the research community. My biggest concern, however, is the technical contribution to the community.

What is the technical novelty that the authors have shown on top of the existing techniques in the logistic bandits shown in Faury et al. (2020)? Generalization to the multinomial setting from logistic setting using Faury et al. (2020) appears to be fairly straightforward. Can you elaborate on what is technically difficult in such a generalization?

Also, the paper does not mention a recent work on MNL contextual bandits Agrawal et al. (2020) which studies the combinatorial setting, hence a different problem, but still leverages the work of Faury et al. (2020) to establish improved regret bounds.

- Faury et al. (2020), Improved Optimistic Algorithms for Logistic Bandits
- Agrawal et al. (2020), A Tractable Online Learning Algorithm for the Multinomial Logit Contextual Bandit,

**Time Spent Reviewing:**

4

---

> ### Author Response · Authors · 2021-08-10
> **Response to Reviewer yP7L**
>
> We thank the reviewer for their valuable time and detailed feedback. We are encouraged by the reviewer’s comments: “the paper is well-written and easy to understand”, “the problem setting is well-motivated”, and “this variant of the MNL problem is worth attention from the research community”. We make the following clarifications below.
>
> **Novelty:** We acknowledge that our analysis is inspired by and builds upon Faury et al. (2020), which is pointed out multiple times in the paper (e.g. Lines 43-45). However, we are confident that the extension to the MNL setting studied here involves several non-trivial contributions: (1) coming up with an appropriate and analytically tractable formulation of the necessary assumptions on problem constants (Assumption 2); (2) computations of $\kappa$ and $L$, which need their own analysis (detailed in Section 3) that is comparatively more challenging and delicate than their binary logistic model's counterparts; (3) making careful use of Kronecker product properties in all the proofs' steps adds extra intricacies compared to analogous steps in the binary case (e.g., see Lines 218-237 and 575-576); (4) coming up with a carefully designed algorithm (Improved MNL-UCB in Appendix D) that further improves the regret bound in terms of $\kappa$ is especially challenging because of the following: A close inspection of the proof presented in the previous subsections reveals that the "extra" $\sqrt{\kappa}$  factor enters the regret when we replace the matrix $\mathbf{G}_t(\boldsymbol\theta_\ast,\boldsymbol\theta)$ with the simpler matrix $\tilde{\mathbf{V}}_t$ (cf. Lemma 12). In turn, this is possible by replacing the key matrix $\mathbf{A}(\mathbf{x},\boldsymbol\theta)$ ---the Jacobian of the MNL model--- by the smaller ---in the sense of the Loewner order--- matrix $\frac{1}{\kappa}I_K$ (cf. Assumption1). In the binary case, the idea of Faury et al. (2020) to circumvent this step is to introduce a refined ``local" lower bound to the derivative of the logistic model (corresponding to our Jacobian above). In the binary case, such a lower bound always exists. However, this is not at all clear in the multinomial case because the Loewner order is only a partial order. This is yet another demonstration that extensions to the multinomial case are challenging in their own right and might require careful treatment. A detailed discussion on how to carefully propose new definitions of the estimator $\boldsymbol\theta_t$ chosen from a newly defined set $
> \Theta\cap\mathcal{M}_t(\delta)$ (instead of $\Theta$ in equation 21) based on the minimal elements of partially Loewner-ordered set $\mathcal{C}_t(\delta)\cap \Theta$ is presented in Appendix D.
>
> We thank the reviewer for bringing Agrawal et al. (2020) into our attention. As pointed out by the reviewer, this paper studies a different problem setup than ours, however, a comparison with their technical contributions is definitely worth including and we will do so in the revision. Please see the first paragraph in our response to Reviewer Shft for some details.

---

### Author Response · Authors · 2021-09-03
**Thank you & happy to discuss more as needed**

Dear Reviewers and AC,

As the discussion period reaches its end, we wanted to thank you all for your time spent on our submission.

We appreciate your feedback. If there are more questions, we are always happy to discuss and clarify.

Best regards,
Authors

---

### Decision · Program_Chairs · 2021-09-27

**Decision:**

Accept (Poster)

**Comment:**

This paper proposes a UCB-based algorithm for multinomial logit bandits. Its initial scores were 6, 6, 5, and 5; and they did not change during the discussion. The reviewers liked the rebuttal of the authors and agreed that this paper would a good contribution to NeurIPS. The main factor in this decision was the importance of the application. On the other hand, the paper builds heavily on Faury et al. (2020) and contains no experiments. In my opinion, this paper is borderline plus and I support its acceptance.